# Extreme Risk Measures: Estimation and Optimization via Stochastic Approximation

## Abstract

Risk measures such as Value at Risk (VaR) and Conditional Value at Risk (CVaR) are critical to evaluating performance in high-risk scenarios such as high-frequency trading, healthcare, risk-sensitive control and insurance. VaR quantifies the maximum potential return over a specified time horizon at a given confidence level, while CVaR extends this by estimating the expected return exceeding the VaR threshold. Estimating the extreme version of these risk measures is inherently sensitive and volatile due to the limited data available at the tail end of the return distribution. This paper introduces an incremental, single-pass, and adaptive variance reduction technique to estimate extreme VaR and CVaR for cases where the underlying distribution is either known or unknown. Additionally, we present a multi-time scale method to optimize CVaR within a parameterized distribution space in an online fashion. We provide both theoretical and empirical analyses to demonstrate the effectiveness and competitiveness of our proposed approaches.

## 1 Introduction

Quantiles (Bahadur, 1966; Jorion, 2007), also known as Value at Risk (VaR), form a class of downside/upside probabilistic measures that provide instrumental statistical information on the risk of a system for effective risk-sensitive decision-making with significant applications in engineering, management, economics and finance. Given a continuously-valued random vector $\mathbf{X} \in \mathbb{R}^d$ defined over a measurable space $(S, \mathcal{F})$, where $S$ and $\mathcal{F}$ is a $\sigma$-field on $S$. Let $\mathbb{P}_x$ be the probability measure of $\mathbf{X}$ which is absolutely continuous *w.r.t.* Lebesgue measure $\nu$. Also, consider a bounded, continuous return function $\phi : \mathbb{R}^d \to \mathbb{R}$ with $\phi(x) \in [\phi_l, \phi_u]$ and $-\infty < \phi_l, \phi_u < \infty$. Given a degree of certainty $\rho \in [0, 1]$, we define the upside $\rho$-quantile of $\phi(\mathbf{X})$ *w.r.t.* $\mathbb{P}_x$ (denoted as $\text{VaR}_\rho(\mathbb{P}_x)$) as follows:

$$\text{VaR}_\rho(\mathbb{P}_x) = \sup_{\gamma \in [\phi_l, \phi_u]} \{\mathbb{P}_x\big(\phi(\mathbf{X}) \geqslant \gamma\big) \geqslant \rho\}, \tag{1}$$

The closed, analytic form of the performance function $\phi$ may be unknown, however, for each input $x$, the uncorrupted performance value $\phi(x)$ is available.

Thus, the quantity $\text{VaR}_\rho(\mathbb{P}_x)$ can be construed as the threshold in the range of the performance values of the performance function $\phi$ beyond which the probability measure with respect to $\mathbb{P}_x$ is at least $\rho$. Quantiles are more generalized quantities than the median (Hodges & Lehmann, 1983), a more popular performance measure that is simply the 0.5-quantile. The quantiles can also be interpreted as synonymous with statistical ordering of the performance values of $\phi$ with respect to the probability measure $\mathbb{P}_x$, since for each $\rho$, the quantile provides a bifurcation of the entire range of performance values into two regions (not necessarily disjoint) of probability mass (*w.r.t.* $\mathbb{P}_x$) of at least $\rho$ and at most $1 - \rho$, respectively. It is easy to verify that in the considered setting (both $\phi$ and $\mathbb{P}_x$ are continuous), we have $\text{VaR}_\rho(\mathbb{P}_x) = F_\phi^{-1}(1 - \rho)$, where $F_\phi$ is the cumulative distribution function (CDF) of $\phi(\mathbf{X})$, *i.e.*, $F_\phi(\gamma) = \mathbb{P}_x(\phi(\mathbf{X}) \leqslant \gamma)$, $\gamma \in [\phi_l, \phi_u]$. For lucidity, the range of $F_\phi^{-1}$ is restricted to the closed interval $[\phi_l, \phi_u]$. However, in most cases, a closed analytic form for $F_\phi^{-1}$ is not available. Therefore, it may not always be possible to develop a tractable deterministic procedure to calculate $\text{VaR}_\rho(\mathbb{P}_x)$ and one has to resort to efficient estimation approaches. $\text{VaR}$s are often of interest in

the analysis of data for outlier detection, extreme value theory, control risk (Bienstock et al., 2014) and are often adopted by banking regulators for risk management.

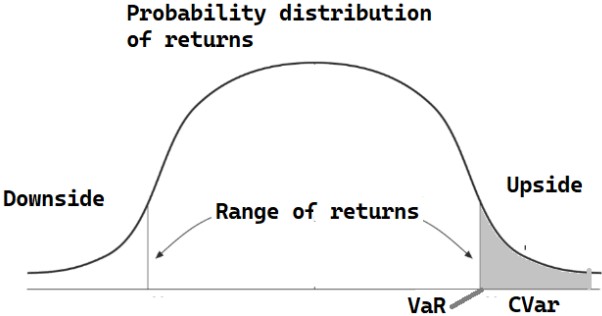

Figure 1: Illustration of `VaR` and `CVaR` over a return distribution.

Another structurally appealing risk measure is the Conditional Value at Risk (`CVaR`) (Rockafellar & Uryasev, 2002; Norton et al., 2021) also called the superquantile or expected shortfall or average `VaR` , is the conditional expectation of returns beyond `VaR`, which is defined as follows:

$$\text{CVaR}_\rho(\mathbb{P}_x) = \mathbb{E}\left[\phi(\mathbf{X})|\phi(\mathbf{X}) \geqslant \text{VaR}_\rho(\mathbb{P}_x)\right] \tag{2}$$

$$= \frac{\int_{\text{VaR}_\rho(\mathbb{P}_x)}^{\infty} \phi(\mathbf{X})d\mathbb{P}_x}{\mathbb{P}_x(\phi(\mathbf{X}) \geqslant \text{VaR}_\rho(\mathbb{P}_x))} \tag{3}$$

Similar to `VaR`, the superquantile `CVaR` can be used to assess the tail of the distribution. Putting emphasis on the tail of the distribution where extreme losses occur, `CVaR` addresses the risk of severe but rare adverse events. This is crucial to understand the impact of worst-case scenarios, which might not be fully captured by `VaR` alone. `VaR` only indicates the worst loss at a certain confidence level but does not account for the magnitude of losses beyond this level. `CVaR` provides a more comprehensive measure of risk than `VaR` by considering the severity of losses beyond the `VaR` level, which is illustrated in Figure(1). Therefore, in situations where a distribution may have a heavy tail, the superquantile accounts for magnitudes of low-probability large-loss tail events, while the quantile does not account for this information. The superquantile is also more tractable in optimization contexts due to its desirable properties such as coherency (Artzner et al., 1999) with respect to the return distribution, convexity, continuity, positively homogeneous and monotonic *w.r.t.* stochastic dominance of order 1 (Rockafellar et al., 2000). `VaR` loses coherence with non-Gaussian return distributions, while `CVaR`'s effectiveness becomes more apparent with respect to these non-Gaussian copulas. Ultimately, the choice between `VaR` and `CVaR` depends on their effectiveness and the relative strengths and weaknesses they exhibit in specific scenarios.

In this paper, we consider the estimation and optimization of extreme risk measures (Glynn, 1996) which are nothing more than $\text{VaR}_\rho$ and $\text{CVaR}_\rho$ with $\rho$ being extremely small, say less than $10^{-6}$. The utility value of the extreme risk measures is significant (Pickands III, 1975) as they assess and manage risks associated with very rare but highly impactful events. For example, consider high-frequency trading (Philippe, 2001), which is an algorithmic trading method in which large volumes of financial commodities are transacted at very high speeds, where fortunes and tail risks, which are rare events, can be characterized as extreme risk measures. Similarly, in government policy making (Wolters, 2012), extreme quantiles can be an effective tool, since the $(1 - \rho)$ quantile can be interpreted as the threshold below which the probability is $(1 - \rho)$ and which thus provides a statistical upper bound (confidence interval) for the health of society and economy in terms of poverty, spending power, labour, prices and so on. Another relevant example is the insurance sector (Dowd & Blake, 2006), where extreme risk measures can provide statistical information about the probable or mean fraction of the customers claiming reimbursement. Other applications include supply chain management, real estate, portfolio management, healthcare sector, and many more.

Although crude Monte Carlo methods can theoretically converge to the true quantile $\forall \rho \in [0, 1]$, the convergence is only realized after considering an infinite number of samples (Pfanzagl, 1976; Ghosh, 1971; Thomas

& Learned-Miller, 2019). For extreme quantiles, the performance of these algorithms over a finite time interval is considerably poor due to huge variance, as the probability of the event $\{\phi(\mathbf{X}) \geqslant \mathtt{VaR}_\rho(\mathbb{P}_x)\}$ is at most $\rho$, an extremely small quantity. Thus, the extreme tail of the distribution is underrepresented, and extreme events are rarely observed in finite samples, leading to high variance in the estimates. Hence, the estimation of the extreme risk measures incurs precision issues especially in the case of $\mathtt{CVaR}$ which requires integration over the tail. This could have significant consequences for risk management, particularly in finance and insurance, where precise estimation of extreme risk measures is crucial for understanding and preparing for rare but severe risks. If risk is underestimated, it can harm the firm's profits and stability, while overestimating risk can lead to holding excess capital.

## 1.1 Related Work

The estimation of extreme risk measures has gained significant attention due to its crucial role in the evaluation of rare but highly impactful events (Glynn, 1996; Pickands III, 1975). These measures have applications in various domains including high-frequency trading, government policy making, insurance, supply chain management (Xie et al., 2016), risk sensitive sequential decision making (Le Tallec, 2007), and healthcare care (Philippe, 2001; Wolters, 2012; Dowd & Blake, 2006). Variance reduction techniques, particularly importance sampling (IS), have been utilized to manage the large variance in extreme risk estimation. (Glynn, 1996) applied large deviation theory and tail approximation to approximate sampling ratios and identify distributions where extreme events are less rare. (Morio, 2012) proposed a non-parametric approach using Gaussian kernel density, while (Egloff & Leippold, 2010) developed a consistent quantile estimator using stochastic approximation (SA) for importance sampling parameter updates. (Pan et al., 2020) established consistency and showed a reduction in variance for adaptive IS in a two-layer model. (Wächter et al., 2017) demonstrated the convergence of the sample average approximation (SAA) for non-IID samples with adaptive IS. (Bardou et al., 2009) presented a method to estimate $\mathtt{VaR}$ and $\mathtt{CVaR}$ via incremental, adaptive unconstrained IS, achieving the smallest asymptotic variance among the chosen IS class under the Gaussian and inverse Gaussian settings. Building on this, recent research has generalized and extended adaptive IS techniques to handle more complex parametrizations, solve general stochastic root-finding problems, and embed adaptive IS in both SA and SAA frameworks.

Other variance reduction techniques in quantile estimation include controlled stratification (Cannamela et al., 2008), bootstrap quantile estimation through importance sampling (Hu & Su, 2008), and efficient simulation of large deviation events (Botev & Lloyd, 2015). The field of rare-event simulation has significantly contributed to extreme quantile estimation. Approaches include IS schemes for deterministic oracles based on the theory of large deviations (Budhiraja & Dupuis, 2019), the dominating point method (Sadowsky & Bucklew, 1990; Dieker & Mandjes, 2005; Owen & Zhou, 2019; Bai et al., 2022), subsolution approaches (Dupuis & Wang, 2009), twisting of the risk rate (Juneja & Shahabuddin, 2002), and mixture-based schemes (Blanchet & Glynn, 2008). These techniques have been applied to various domains including queueing (Kroese & Nicola, 1999; Blanchet & Lam, 2009), communication networks (Kesidis & Walrand, 1993), finance (Glasserman & Li, 2005), insurance (Asmussen, 1985), reliability (Nicola et al., 1993; Heidelberger, 1995; Rubino & Tuffin, 2009), biological processes (Grassberger, 2002; Sandmann, 2009), and dynamic systems (Dupuis et al., 2012; Vanden-Eijnden & Weare, 2012).

The cross-entropy method (Rubinstein & Kroese, 2004; 2016; De Boer et al., 2005) offers another approach for designing IS estimator. While similar to adaptive IS in using sequential updates and SAA, it differs in its formulation and the guarantees it provides. Other relevant algorithms include the Monte-Carlo method (Cannamela et al., 2008), stochastic approximation method (Joseph & Bhatnagar, 2015), quantile regression method (Koenker, 2005; Chakraborty, 2003; Takeuchi et al., 2006), adaptable buffer algorithm (Arandjelović et al., 2015), $P^2$ algorithm (Jain & Chlamtac, 1985), single-pass low-storage algorithm (Liechty et al., 2003), and the quadratic approximation procedure for computing VaR (Glasserman et al., 2000).

(He et al., 2023) addresses circular problem which is inherent in solutions based on importance sampling, where an effective importance sampler requires knowing the solution. To address this, adaptive importance sampling is being introduced, which sequentially updates the sampler to simultaneously find the optimal sampler and solution. Despite these advancements, the challenge of efficiently estimating extreme quantiles

with high precision remains an active area of research, particularly for applications in finance and insurance where accurate risk assessment is crucial.

## 1.2 Our contribution

Our work extends the estimation of risk measures (quantile and superquantile) to extreme scenarios by introducing a single-pass, incremental, adaptive, stable algorithm that could efficiently process large datasets in the order of quadratic complexity per iteration with respect to the dimension of the input space. Our algorithm ensures an almost sure convergence to the risk measures with minimal assumptions, offering robustness against extreme values of $\rho$ without incurring heavy computational cost. We also extend our approach to superquantile optimization using a zero order multi-timescale approach. These contributions address critical challenges underlying extreme risk measures such as large variance and latent probability distribution, providing a more efficient and robust solution for applications in finance, insurance, risk-sensitive sequential decision making and other domains that require accurate estimation of extreme events.

## 1.3 Paper Outline

In Section 2, we introduce the theoretical foundation of quantiles and reformulate them as an optimization problem, setting up our stochastic approximation approach. Section 3 presents our core algorithm, using importance sampling with a truncated normal surrogate to address high variance associated with extreme quantiles. We derive the optimal surrogate, discuss its approximation via the Natural Exponential Family (NEF), and analyze the approximation error, providing a concrete Gaussian surrogate implementation along with convergence analysis for `VaR` and `CVaR`. Section 4 extends this to optimize `CVaR` using a multi-timescale stochastic approximation and randomized finite-difference gradients. In Section 5, we tackle cases where the true return distribution is unknown, using moment projection for distribution approximation. Finally, Section 6 empirically validates our algorithms in risk-adjusted portfolio optimization, risk-sensitive reinforcement learning for robotic control, and glycemic control.

## 1.4 Summary of Notation

$\mathbb{I}_{\{\cdot\}}$ is the indicator function, *i.e.*, for an arbitrary set $A$, we have $\mathbb{I}_A(x) = 1$, if $x \in A$ and 0 otherwise. Let $\mathbb{I}_{d \times d}$ represent the identity matrix of order $d$. Also, $\mathbb{E}_{\mathbb{P}}[\cdot]$ and $\mathbb{V}_{\mathbb{P}}[\cdot]$ are the expectation and the variance *w.r.t.* the probability measure $\mathbb{P}$ respectively. Let $S_{++}^d$ represent the space of real-valued, symmetric, positive-definite matrices of order $d$. And `Bernoulli`$(\{a, b\}, \lambda)$ represents the Bernoulli distribution with $\mathbb{P}(a) = \lambda$ and $\mathbb{P}(b) = 1 - \lambda$. We let $\nu$ to denote the Lebesgue measure. The KL-divergence between two probability measures $P$ and $Q$ is defined as follows: $\text{KL}(P, Q) := \int \log \frac{dP}{dQ} dP$, where $dP/dQ$ is the Radon-Nikodym derivative of $P$ *w.r.t.* $Q$. Note that $dP/dQ$ is defined only if $P$ is absolutely continuous *w.r.t.* $Q$ denoted as $P \ll Q$, *i.e.*, $Q(A) = 0 \Rightarrow P(A) = 0$, for every Borel set $A$.

$$\text{We define, for } x, y \in \mathbb{R}, \quad \mathbb{I}(x, y)^+ = \begin{cases} 1 & \text{if } x \geqslant y \\ 0 & \text{otherwise} \end{cases} \quad \text{and} \quad \mathbb{I}(x, y)^- = \begin{cases} 1 & \text{if } x < y \\ 0 & \text{otherwise} \end{cases} \quad (4)$$

We quantify the same quantity by $\mathbb{I}^+(\cdot, \cdot)$ and $\mathbb{I}(\cdot, \cdot)^+$ interchangeably using abuse of notation. Similarly, $\mathbb{I}^-(\cdot, \cdot)$ and $\mathbb{I}(\cdot, \cdot)^-$. We define $\text{CONV}(A)$ as the convex hull of the set $A$, which is the intersection of all convex sets that contain $A$. Formally, $\text{CONV}(A) = \{\sum_{i=1}^n \lambda_i x_i : x_i \in A, \lambda_i \geqslant 0, \sum_{i=1}^n \lambda_i = 1\}$, where $A \subseteq C$ and $C$ is a vector space. We denote the closure of the convex hull of $A$ as $\overline{\text{CONV}}(A)$. As shorthand, we write the region $\{x \in \mathbb{R}^d : \phi(x) \geqslant \gamma\}$ as $\{\phi(x) \geqslant \gamma\}$ and similarly use $\{\phi(x) \leqslant \gamma\}$ to denote $\{x \in \mathbb{R}^d : \phi(x) \leqslant \gamma\}$. For a random vector $\mathbf{X} \in \mathbb{R}^d$, the event $\{\phi(\mathbf{X}) \geqslant \gamma\}$ represents the set $\{\mathbf{X} \in \mathbb{R}^d : \phi(\mathbf{X}) \geqslant \gamma\}$ and similarly, the event, $\{\phi(\mathbf{X} \leqslant)\gamma\}$ represents $\{\mathbf{X} \in \mathbb{R}^d : \phi(\mathbf{X}) \leqslant \gamma\}$.

## 2 Background

The quantile problem is reformulated as an optimization problem in Lemma 1 of (Homem-de Mello, 2007). The lemma provides a characterization of the $(1 - \rho)$-quantile of a function with real value $\phi$ *w.r.t.* in a

given probability measure $\mathbb{P}_x$. This reformulation improves the ability to compute and analyze quantiles in practical applications. For better comprehension, we restate the lemma here:

**Lemma 1.** *(Lemma 1 of (Homem-de Mello, 2007)) The upside $\rho$-quantile of a bounded, real-valued function $\phi$ $\left(\text{with } \phi(x) \in [\phi_l, \phi_u], \forall x\right)$ w.r.t. the probability measure $\mathbb{P}_x$ is reformulated as the optimization problem*

$$VaR_\rho(\mathbb{P}_x) = \arg\min_{\gamma \in [\phi_l, \phi_u]} \int \psi(\phi(\mathbf{X}), \gamma) d\mathbb{P}_x \tag{5}$$

*where the residual function $\psi(\phi(x), \gamma) := (1 - \rho)(\phi(x) - \gamma)\mathbb{I}(\phi(x), \gamma)^+ + \rho(\gamma - \phi(x))\mathbb{I}(\phi(x), \gamma)^-$.*

The above characterization can be interpreted as the solution to the weighted mean of linear residues in two disjoint regions $\{\phi(x) \geq \gamma\}$ and $\{\phi(x) \leq \gamma\}$ with each region weighted asymmetrically, *i.e.*, $(1 - \rho)$ for the region $\{\phi(x) \geq \gamma\}$ and $\rho$ for the region $\{\phi(x) \leq \gamma\}$. This is synonymous to the fact that the mean can be interpreted as the solution to the mean quadratic residues. Note that the asymmetry of the residues deepens as $\rho$ approaches 0 or 1. In a naive manner, one can verify the above lemma by assigning the subdifferential of $\mathbb{E}_{\mathbb{P}_x}[\psi(H(\mathbf{X}), \gamma)]$ to 0. For now, assume that the sub-differential operator can be taken inside the expectation (nuanced details provided in Eq. (7)). Then, we have

$$\partial_\gamma \int \psi(\phi(\mathbf{X}), \gamma) d\mathbb{P}_x = \int \partial_\gamma \psi(\phi(\mathbf{X}), \gamma) d\mathbb{P}_x = \int -(1 - \rho)\mathbb{I}(\phi(\mathbf{X}), \gamma)^+ + \rho\mathbb{I}(\phi(\mathbf{X}), \gamma)^- d\mathbb{P}_x$$
$$= -(1 - \rho)\mathbb{P}_x(\phi(\mathbf{X}) \geq \gamma) + \rho\mathbb{P}_x(\phi(\mathbf{X}) \leq \gamma)$$
$$= \rho(\mathbb{P}_x(\phi(\mathbf{X}) \geq \gamma) + \mathbb{P}_x(\phi(\mathbf{X}) \leq \gamma)) - \mathbb{P}_x(\phi(\mathbf{X}) \geq \gamma)$$
$$= \rho - 1 + F_\phi(\gamma),$$

where $F_\phi$ is the cumulative distribution function (CDF) of $\phi(\mathbf{X})$, *i.e.*, $F_\phi(\gamma) = \mathbb{P}_x(\phi(\mathbf{X}) \leq \gamma)$.

Now equating the sub-differential to 0, we obtain

$$\partial_\gamma \int \psi(\phi(\mathbf{X}), \gamma) d\mathbb{P}_x = 0 \quad \Rightarrow \rho - 1 + F_\phi(\gamma) = 0 \quad \Rightarrow F_\phi(\gamma) = 1 - \rho.$$

A pertinent observation about the objective function in Lemma 1 is the following:

**Proposition 1.** *The function $\mathbb{E}_{\mathbb{P}_x}[\psi(\phi(\mathbf{X}), \gamma)]$ is convex in $\gamma$.*

*Proof.* Please refer to Appendix. $\square$

To comprehend our algorithm, it is imperative to explore a few more structural properties of the objective function $\int \psi(\phi(\mathbf{X}), \gamma) d\mathbb{P}_x$. For a given performance function $\phi$, it is easy to verify that $\psi(\phi(x), \gamma)$ is continuous for a fixed $\gamma \in \mathbb{R}$. This follows directly from the definition of $\psi$. Also, for a fixed $\gamma \in \mathbb{R}$, one can easily verify that $\psi(\phi(x), \gamma)$ is differentiable at all points except in the set $\{\phi(x) = \gamma\}$. However, at all the points belonging to the set $\{\phi(x) = \gamma\}$ sub-differential exists (follows from the convexity). By simple analysis, we obtain the following closed form expression for the sub-differential of $\psi$:

$$\partial_\gamma \psi(\phi(x), \gamma) = \begin{cases} \{-(1 - \rho)\mathbb{I}(\phi(x), \gamma)^+ + \rho\mathbb{I}(\phi(x), \gamma)^-, \text{ for } \gamma \neq \phi(x), \\ [-(1 - \rho), \rho], \text{ for } \gamma = \phi(x). \end{cases} \tag{6}$$

Additionally, note that $\partial_\gamma \psi(\phi(\cdot), \cdot)$ is bounded. Hence, by appealing to the Dominated Convergence Theorem (Rubinstein & Shapiro, 1993), one can indeed interchange the operators $\partial_\gamma$ and $\mathbb{E}_{\mathbb{P}_x}[\cdot]$ in the expression of $\partial_\gamma \mathbb{E}_{\mathbb{P}_x}[\psi(\phi(\mathbf{X}), \gamma)]$, *i.e.*,

$$\int \partial_\gamma \psi(\phi(\mathbf{X}), \gamma) d\mathbb{P}_x = \partial_\gamma \int \psi(\phi(\mathbf{X}), \gamma) d\mathbb{P}_x. \tag{7}$$

This above reformulation provides a more tractable method for finding quantiles by leveraging optimization techniques. One can solve the quantile estimation problem, by finding the solution to the optimization

problem (5) using stochastic approximation techniques (Robbins & Monro, 1951; Ljung, 1978; Kushner & Clark, 2012) as follows:

$$\gamma_{t+1} = \gamma_t - \alpha_{t+1}\Delta_t^{\gamma}(\mathbf{X}_{t+1}), \text{ where } \mathbf{X}_{t+1} \sim \mathbb{P}_x, \tag{8}$$

with the decrement term $\Delta_t^{\gamma}$ given by

$$\Delta_t^{\gamma}(x) := -(1-\rho)\mathbb{I}(\phi(x), \gamma_t)^+ + \rho\mathbb{I}(\phi(x), \gamma_t)^-. \tag{9}$$

The algorithm is specifically a stochastic subgradient descent, where a time-indexed random variable $\gamma_t$ is maintained to track the true quantile, where at each time instant $t$, the random variable $\gamma_t$ is calibrated in the direction antipodal to the sub-gradient of the residual function $\psi$ in congruence with the perceived randomness characterized by the probability measure $\mathbb{P}_x$. The decrement term in the stochastic recursion is the sub-gradient contained in the sub-differential $\partial_\gamma \psi$.

## 3   Quantile and Superquantile Estimation Algorithm

One can indeed show that the stochastic recursion (8) asymptotically tracks the true quantile *i.e.*, $\gamma_t \to \mathtt{VaR}_\rho(\mathbb{P}_x)$ as $t \to \infty$ almost surely. However, this is a theoretical convergence result that is only realized in the limiting sense, *i.e.*, as the number of samples tends to infinity. Practically, stochastic recursion (8) alone does not produce quality estimates of the true quantile in finite time, specifically for situations where the quantile parameter $\rho$ is extremely small. We elaborate on this situation more vividly here. Consider the visual setting provided in Figure 2. There, the red canvas represents a graphic representation of the distribution of the probability measure $\mathbb{P}_x$ over $\phi(\mathbf{X})$ (with darker red shades representing higher probability compared to lighter ones), while the dotted horizontal line is the real line along which the iterates $\{\gamma_t\}$ generated by recursion (8) drift. At the initial stage of the recursion, say at $t = 1$, since the probability measure is strongly concentrated beyond $\gamma_1$, the performance value of the sample $\mathbf{X}_2$ is more likely to lie beyond $\gamma_1$ and hence the drift of $\gamma_1$ is more likely towards the true quantile. However, after the initial transient stage, say at $t = 100$, note that the iterate $\gamma_{100}$ has leaped beyond the heavy probability region, and in this case, the performance value of the sample $\mathbf{X}_{101}$ is more likely to lie behind $\gamma_{100}$, and hence the subsequent iterate drifts away from the true quantile (the magnitude of the drift is small since it is weighted by $\rho$, however, it prevents the positive drift towards the true quantile). This results in huge variance resulting in poor precision while estimating extreme quantiles.

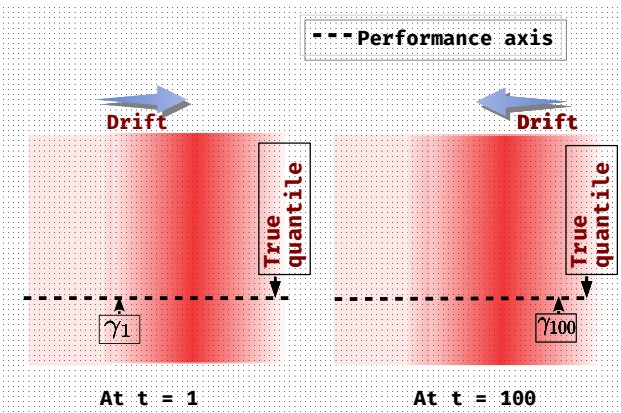

Figure 2: We illustrate the drift of $\gamma_t$ generated by the stochastic recursion (8) at two points: $t = 1$ and $t = 100$. At $t = 1$, $\gamma_t$ drifts toward the true quantile with high probability due to the sampling distribution's strong presence near and beyond the target region. However, by $t = 100$, as $\gamma_t$ approaches the true quantile, the drift direction flips. This occurs because the sampling distribution has lower likelihood beyond $\gamma_{100}$ and higher probability behind it, causing a reverse drift. This reversal increases the variance of the estimates, as they are more likely to deviate from the neighborhood of the true quantile.

To address these issues, we use the importance sampling technique (Geweke, 1989; Glynn, 1996; Asmussen & Glynn, 2007). In our approach, at each iteration $t$, the sample $\mathbf{X}_{t+1}$ is drawn using a surrogate measure $Q_t$, which is absolutely continuous with respect to the Lebesgue measure $\nu$ and may differ from the given measure $\mathbb{P}_x$. The discrepancy in the sampling distribution is then corrected by re-weighting the samples using the Radon–Nikodym derivative of the original measure with respect to the surrogate measure. However, the choice of the surrogate measure $Q_t$ cannot be arbitrary, but possesses the conforming characteristic that the event $\{\phi(\mathbf{X}) \geqslant \gamma_t\}$ is more likely with respect to $Q_t$ than with respect to $\mathbb{P}_x$. This introduces a level of adaptability to the sampling process, potentially enhancing the precision of estimates, particularly in scenarios involving a small quantile parameter. To identify an appropriate surrogate measure, it suffices to restrict the search to the subspace of measures for which $\mathbb{P}_x$, when truncated to the region $\{\phi(\mathbf{X}) \geqslant \gamma_t\}$, is absolutely continuous with respect to the surrogate measure. In other words, any Borel set within the region $\{\phi(\mathbf{X}) \geqslant \gamma_t\}$ that has zero measure under $Q_t$ must also have zero measure under $\mathbb{P}_x$. Formally, this requirement can be stated as follows:

$$Q_t(\{\phi(\mathbf{X}) \geqslant \gamma_t\} \cap B) = 0 \Rightarrow \mathbb{P}_x(\{\phi(\mathbf{X}) \geqslant \gamma_t\} \cap B) = 0, \forall B : \texttt{Borel set}. \tag{10}$$

This implies that

$$\frac{dQ_t}{d\nu}(x) = 0 \Rightarrow \frac{d\mathbb{P}_x}{d\nu}(x) = 0, \forall x \in \{\phi(x) \geqslant \gamma_t\} \texttt{ almost everywhere}$$

$$\Rightarrow \left\{ \frac{dQ_t}{d\nu}(x)\mathbb{I}(\phi(x), \gamma_t)^+ \neq 0 \right\} \supseteq \left\{ \frac{d\mathbb{P}}{d\nu}(x)\mathbb{I}(\phi(x), \gamma_t)^+ \neq 0 \right\} \texttt{ almost everywhere}. \tag{11}$$

This means that the support of the Radon-Nikodym derivative $dQ_t/d\nu$ contains the support of $d\mathbb{P}_x/d\nu$ almost everywhere in the region $\{\phi(x) \geqslant \gamma_t\}$.

Now, by sampling $\mathbf{X}_{t+1}$ using the surrogate PDF $Q_t$, we rewrite the stochastic recursion (8) as follows:

$$\gamma_{t+1} := \gamma_t - \alpha_t \frac{d\mathbb{P}_x}{dQ_t}(\mathbf{X}_{t+1})\Delta_t^\gamma(\mathbf{X}_{t+1}), \text{ with } \mathbf{X}_{t+1} \sim Q_t$$

$$= \gamma_t - \alpha_t \frac{d\mathbb{P}_x}{dQ_t}(\mathbf{X}_{t+1})\Big( -(1-\rho)\mathbb{I}(\phi(\mathbf{X}_{t+1}), \gamma_t)^+ + \rho\mathbb{I}(\phi(\mathbf{X}_{t+1}), \gamma_t)^- \Big), \tag{12}$$

where $\frac{d\mathbb{P}_x}{dQ_t}$ is the Radon-Nikodym derivative of $\mathbb{P}_x$ $w.r.t.$ the surrogate distribution $Q_t$. Let $\zeta_{t+1}(x) := \frac{d\mathbb{P}_x}{dQ_t}(x)$.

As mentioned earlier, our goal is to find the surrogate measure $Q_t$ such that the event $\{\phi(\mathbf{X}_{t+1}) \geqslant \gamma_t\}$ is more likely. Hence, we seek the optimum surrogate measure that minimizes the variance of the random variable $\zeta_t(\mathbf{X}_{t+1})\mathbb{I}(\phi(\mathbf{X}_{t+1}), \gamma_t)^+$, $i.e.$,

$$\text{Find the probability measure sequence } \{Q_t\}, \text{ where } Q_t \text{ satisfies } \mathbb{V}_{Q_t}\big[\zeta_t(\mathbf{X})\mathbb{I}(\phi(\mathbf{X}), \gamma_t)^+\big] = 0 \tag{13}$$

For brevity, let

$$\ell_{t+1} := \int \mathbb{I}(\phi(\mathbf{X}), \gamma_t)^+ d\mathbb{P}_x \text{ and } \widehat{\ell}_{t+1} := \zeta_t(\mathbf{X})\mathbb{I}(\phi(\mathbf{X}), \gamma_t)^+, \text{ where } \mathbf{X} \sim Q_t. \tag{14}$$

Note that assuming Eq. (10), we see that $\widehat{\ell}_{t+1}$ is an unbiased estimate of $\ell_{t+1}$. Indeed, for $G := \{x \in \mathbb{R}^d : \frac{dQ_t}{d\nu}(x) \neq 0\}$, we have

$$\mathbb{E}_{Q_t}\big[\widehat{\ell}_{t+1}\big] = \int_G \zeta_t(\mathbf{X})\mathbb{I}(\phi(\mathbf{X}), \gamma_t)^+ dQ_t + \int_{G^c} \zeta_t(\mathbf{X})\mathbb{I}(\phi(\mathbf{X}), \gamma_t)^+ dQ_t$$

$$= \int_G \frac{d\mathbb{P}_x}{dQ_t}\mathbb{I}(\phi(\mathbf{X}), \gamma_t)^+ dQ_t = \int_G \mathbb{I}(\phi(\mathbf{X}), \gamma_t)^+ d\mathbb{P}_x$$

$$= \int \mathbb{I}(\phi(\mathbf{X}), \gamma_t)^+ d\mathbb{P}_x = \ell_{t+1} \tag{15}$$

We dropped the integral over $G^c$ from the first equality since $Q_t(G^c) = 0$. Additionally, we have the following result on the variance of $\widehat{\ell}_{t+1}$:

**Proposition 2.** *Let the surrogate probability measure $Q_t$ satisfy Eq. (10). Then*

$$\mathbb{V}_{Q_t}\left[\widehat{\ell}_{t+1}\right] = \int \frac{((d\mathbb{P}_x/d\nu)\mathbb{I}(\phi(\mathbf{X}),\gamma_t)^+ - \ell_{t+1}(dQ_t/d\nu))^2}{(dQ_t/d\nu)}d\nu.$$

*Proof.* Please refer to Appendix. □

Now, the optimal surrogate probability measure is the one that achieves zero variance. Therefore, the Radon-Nikodym derivative of the optimal surrogate measure $Q_t$ with respect to the product Lebesque measure is obtained as follows:

$$\mathbb{V}_{Q_t}\left[\widehat{\ell}_{t+1}\right] = 0 \Leftrightarrow \int \frac{((d\mathbb{P}_x/d\nu)\mathbb{I}(\phi(\mathbf{X}),\gamma_t)^+ - \ell_{t+1}(dQ_t/d\nu))^2}{(dQ_t/d\nu)}d\nu = 0$$

$$\Leftrightarrow \frac{dQ_t}{d\nu}(x) = \frac{(d\mathbb{P}_x/d\nu)(x)\mathbb{I}(\phi(x),\gamma_t)^+}{\ell_{t+1}} = \frac{(d\mathbb{P}_x/d\nu)(x)\mathbb{I}(\phi(x),\gamma_t)^+}{\int \mathbb{I}(\phi(\mathbf{X}),\gamma_t)^+ d\mathbb{P}_x}. \quad (16)$$

Note that the Radon-Nikodym derivative $\frac{dQ_t}{d\nu}$ of the surrogate measure $Q_t$ w.r.t. Lebegue measure $\nu$ is indeed a valid probability density function (since $\frac{dQ_t}{d\nu}(x) \geq 0$ and $\int \frac{dQ_t}{d\nu}d\nu = 1$) and has its entire support in the region $\{\phi(x) \geq \gamma_t\}$. Also, it satisfies Eq. (10) as required. However, computing the values of $dQ_t/d\nu(x)$ for different values of $x$ is intractable since its expression contains $\ell_{t+1}$ which is hard to compute. This makes sampling using the optimal surrogate probability measure infeasible. To overcome this, one must resort to approximation techniques, where we attempt to find a tractable, albeit close approximation to the surrogate measure $Q_t$. We approximate the optimal surrogate measure $Q_t$ by projecting it onto a parametrized family of probability measures that is rich enough (over $\mathbb{R}^d$) $\mathcal{Q}_\Theta := \{Q_\theta | \theta \in \Theta \subseteq \mathbb{R}^m\}$ (the specifics regarding the choice of $\mathcal{Q}$ are detailed later) with respect to the Kullback-Leibler (KL) divergence (Kullback, 1959) (moment projection). This is illustrated in Figure (3). In other words, $Q_t$ is approximated by the probability measure $Q_{\theta_t} \in \mathcal{F}_\Theta$, where

$$\theta_{t+1} := \arg\min_{\theta \in \Theta} \text{KL}(Q_t, Q_\theta). \quad (17)$$

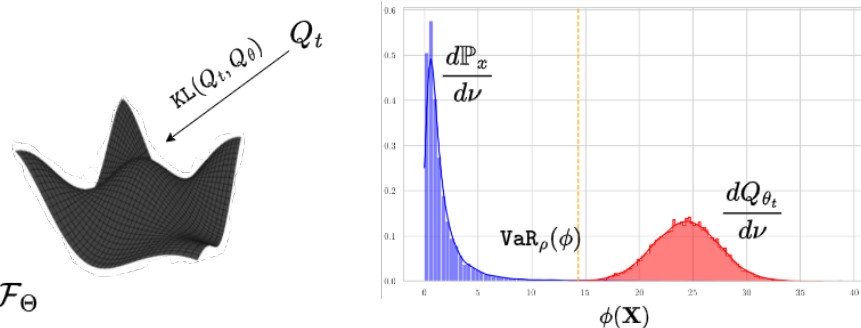

Figure 3: The optimal surrogate PDF which is obtained is the one which minimizes the KL distance.

For brevity, we denote $\mathbb{E}_\theta[\mathbf{X}] = \int X dQ_\theta$ and $Q_\theta(A) = \int_A dQ_\theta$. Now note that

$$
\begin{aligned}
\arg\min_{\theta\in\Theta} \mathsf{KL}(Q_t, Q_\theta) &= \arg\min_{\theta\in\Theta} \int dQ_t \log\left(\frac{dQ_t}{dQ_\theta}\right) \\
&= \arg\min_{\theta\in\Theta} \underbrace{\int (dQ_t/d\nu)\log(dQ_t/d\nu)d\nu}_{\text{Does not contain } \theta. \text{ So we drop it.}} - \int (dQ_t/d\nu)\log(dQ_\theta/d\nu)d\nu \\
&= \arg\min_{\theta\in\Theta} -\int (dQ_t/d\nu)\log(dQ_\theta/d\nu)d\nu \\
&= \arg\max_{\theta\in\Theta} \int (dQ_t/d\nu)\log(dQ_\theta/d\nu)d\nu \\
&= \arg\max_{\theta\in\Theta} \frac{\int (d\mathbb{P}_x/d\nu)\mathbb{I}(\phi(\mathbf{X}),\gamma_t)^+ \log(dQ_\theta/d\nu)d\nu}{\int \mathbb{I}(\phi(\mathbf{X}),\gamma_t)^+ d\mathbb{P}_x} \\
&= \arg\max_{\theta\in\Theta} \int \frac{(d\mathbb{P}_x/d\nu)}{(dQ_{\theta_t}/d\nu)}(dQ_{\theta_t}/d\nu)\mathbb{I}(\phi(\mathbf{X}),\gamma_t)^+ \log(dQ_\theta/d\nu)d\nu.
\end{aligned}
$$

Therefore,

$$
\begin{aligned}
\theta_{t+1} &= \arg\max_{\theta\in\Theta} \int \frac{(d\mathbb{P}_x/d\nu)}{(dQ_{\theta_t}/d\nu)}\mathbb{I}(\phi(\mathbf{X}),\gamma_t)^+ \log(dQ_\theta/d\nu)dQ_{\theta_t} \\
&= \arg\max_{\theta\in\Theta} \int \zeta_t(\mathbf{X})\mathbb{I}(\phi(\mathbf{X}),\gamma_t)^+ \log(dQ_\theta/d\nu)dQ_{\theta_t}, \text{ where } \zeta_t(x) = \frac{(d\mathbb{P}_x/d\nu)}{(dQ_{\theta_t}/d\nu)}(x).
\end{aligned} \tag{18}
$$

The natural exponential family (NEF) is a class of parametrized probability measures $\mathcal{Q}_\Theta = \{Q_\theta | \theta \in \Theta \subseteq \mathbb{R}^m\}$ which satisfies the following form:

$$
(dQ_\theta/d\nu)(x) = h(x)\exp\left(\theta^\top \Gamma(x) - K(\theta)\right), \tag{19}
$$

where $\theta$ is the parameter of the measure, $\Gamma(x)$ is the sufficient statistic, that captures all necessary information for inference about $\theta$ from data, $K(\theta) = \ln \int \exp\left(\theta^\top \Gamma(x)\right)dx$ is the cumulant function, which ensures that the distribution is normalized and $h(x)$ is a base measure that depends only on the data and not on the parameter $\theta$. Here $\Theta = \{\theta \in \mathbb{R}^m | \quad |K(\theta)| < \infty\}$. Let $m(\theta) = \int \Gamma(\mathbf{X})dQ_\theta$.

In the case of NEF as the choice of the distribution space, by the first-order optimization conditions, the solution $\theta_{t+1}$ to the optimization problem (18) satisfies the following:

$$
\begin{aligned}
&\nabla_\theta \int \zeta_t(\mathbf{X})\mathbb{I}(\phi(\mathbf{X}),\gamma_t)^+ \log(dQ_\theta/d\nu)dQ_{\theta_t} = 0 \\
&\Rightarrow \nabla_\theta \int \zeta_t(\mathbf{X})\mathbb{I}(\phi(\mathbf{X}),\gamma_t)^+ \left(\log h(\mathbf{X}) + \theta^\top \Gamma(\mathbf{X}) - K(\theta)\right)dQ_{\theta_t} = 0 \\
&\Rightarrow \int \zeta_t(\mathbf{X})\mathbb{I}(\phi(\mathbf{X}),\gamma_t)^+ \left(\Gamma(\mathbf{X}) - \nabla_\theta K(\theta)\right)dQ_{\theta_t} = 0 \\
&\Rightarrow \int \zeta_t(\mathbf{X})\mathbb{I}(\phi(\mathbf{X}),\gamma_t)^+ \Gamma(\mathbf{X})dQ_{\theta_t} = \int \zeta_t(\mathbf{X})\mathbb{I}(\phi(\mathbf{X}),\gamma_t)^+ \nabla_\theta K(\theta)dQ_{\theta_t} \\
&\Rightarrow m(\theta)\int \zeta_t(\mathbf{X})\mathbb{I}(\phi(\mathbf{X}),\gamma_t)^+ dQ_{\theta_t} = \int \zeta_t(\mathbf{X})\mathbb{I}(\phi(\mathbf{X}),\gamma_t)^+ \Gamma(\mathbf{X})dQ_{\theta_t} \\
&\Rightarrow m(\theta_{t+1}) = \frac{\int \zeta_t(\mathbf{X})\mathbb{I}(\phi(\mathbf{X}),\gamma_t)^+ \Gamma(\mathbf{X})dQ_{\theta_t}}{\int \zeta_t(\mathbf{X})\mathbb{I}(\phi(\mathbf{X}),\gamma_t)^+ dQ_{\theta_t}} \\
&\Rightarrow \theta_{t+1} = m^{-1}\left(\frac{\int \zeta_t(\mathbf{X})\mathbb{I}(\phi(\mathbf{X}),\gamma_t)^+ \Gamma(\mathbf{X})dQ_{\theta_t}}{\int \zeta_t(\mathbf{X})\mathbb{I}(\phi(\mathbf{X}),\gamma_t)^+ dQ_{\theta_t}}\right)
\end{aligned} \tag{20}
$$
$$\tag{21}$$

The function $K(\theta)$ is strictly convex in $\Theta^o$ (interior of $\Theta$) with $\nabla K(\theta) = \int \Gamma(\mathbf{X})dQ_\theta = m(\theta)$ and $\nabla^2 K(\theta) = \mathbb{E}_\theta\left[(\Gamma(\mathbf{X}) - \mathbb{E}_\theta[\Gamma(\mathbf{X})])^2\right]$ (Morris, 1982). Therefore, the Jacobian of $m(\theta)$ is positive definite. From the inverse function theorem, it follows that $m$ is also invertible (Spivak, 2018).

Thus, the recursive $\mathtt{VaR}_\rho(\mathbb{P}_x)$ estimate in eq. (12) can be adapted by using samples drawn from the approxiate surrogate measure $Q_{\theta_t}$. Specifically, the recursion formula for $\mathtt{VaR}_\rho(\mathbb{P}_x)$ can be modified to incorporate these surrogate samples, which will help in refining the $\mathtt{VaR}$ estimate as follows:

$$\gamma_{t+1} = \gamma_t - \alpha_t \frac{d\mathbb{P}_x}{dQ_{\theta_t}}(\mathbf{X}_{t+1})\Big( - (1-\rho)\mathbb{I}(\phi(\mathbf{X}_{t+1}),\gamma_t)^+ + \rho\mathbb{I}(\phi(\mathbf{X}_{t+1}),\gamma_t)\Big)^-, \text{ with } \mathbf{X}_{t+1} \sim Q_{\theta_t} \qquad (22)$$

where $\frac{d\mathbb{P}_x}{dQ_{\theta_t}}$ is the Radon-Nikodym derivative of $\mathbb{P}_x$ $w.r.t.$ the surrogate distribution $Q_t$.

### 3.1 Approximation Error Bounds

The crucial question to address is whether the application of the aforementioned update rule (21) will produce a new surrogate measure $Q_{\theta_{t+1}}$ that demonstrates a higher probability for the event $\{\phi(\mathbf{X}) \geqslant \gamma_t\}$ compared to the original measure $\mathbb{P}_x$. In other words, we need to determine if the updated parameters $\theta_{t+1}$ derived from the update rule, will lead to an improvement in the likelihood of observing the event $\{\phi(\mathbf{X}) \geqslant \gamma_t\}$ relative to the likelihood provided by $\mathbb{P}_x$. In this section, we attempt to quantify the likelihood of the event $\{\phi(\mathbf{X}) \geqslant \gamma_t\}$ with respect to the surrogate measure $Q_{\theta_t}$. To achieve this, observe that

$$0 \leqslant \mathtt{KL}(Q_{t+1}, Q_{\theta_t}) - \mathtt{KL}(Q_{t+1}, Q_{\theta_{t+1}})$$

$$\leqslant \mathtt{KL}(Q_{t+1}, Q_{\theta_t}) - \mathtt{KL}(Q_{t+1}, Q_{\theta_{t+1}}) + \mathtt{KL}(Q_{t+1}, \bar{Q}_{\theta_{t+1}}), \text{ where } \frac{d\bar{Q}_{\theta_{t+1}}}{d\nu}(x) := \frac{\mathbb{I}^+(\phi(x),\gamma_t)(dQ_{\theta_{t+1}}/d\nu)(x)}{\int \mathbb{I}(\phi(\mathbf{X}),\gamma_t)^+ dQ_{\theta_{t+1}}}$$

$$= \int \log\left(\frac{dQ_{\theta_{t+1}}}{dQ_{\theta_t}}\right) dQ_{t+1} + \int \log\left(\frac{d\mathbb{P}_x}{dQ_{\theta_{t+1}}}\right) dQ_{t+1} + \log \frac{\int \mathbb{I}(\phi(\mathbf{X}),\gamma_t)^+ dQ_{\theta_{t+1}}}{\int \mathbb{I}(\phi(\mathbf{X}),\gamma_t)^+ d\mathbb{P}_x}$$

$$= \int \log\left(\frac{d\mathbb{P}_x}{dQ_{\theta_t}}\right) dQ_{t+1} + \log \frac{\int \mathbb{I}(\phi(\mathbf{X}),\gamma_t)^+ dQ_{\theta_{t+1}}}{\int \mathbb{I}(\phi(\mathbf{X}),\gamma_t)^+ d\mathbb{P}_x}$$

$$\leqslant \log \int \frac{d\mathbb{P}_x}{dQ_{\theta_t}} dQ_{t+1} + \log \frac{\int \mathbb{I}(\phi(\mathbf{X}),\gamma_t)^+ dQ_{\theta_{t+1}}}{\int \mathbb{I}(\phi(\mathbf{X}),\gamma_t)^+ d\mathbb{P}_x} \quad \text{(Jensen's inequality)}$$

$$= \log \int \frac{\zeta_t(\mathbf{X})}{\mathbb{P}_x(\phi(\mathbf{X}) \geqslant \gamma_t)} dQ_{t+1} + \log \frac{\int \mathbb{I}(\phi(\mathbf{X}),\gamma_t)^+ dQ_{\theta_{t+1}}}{\int \mathbb{I}(\phi(\mathbf{X}),\gamma_t)^+ d\mathbb{P}_x}.$$

The first inequality above follows since $\theta_{t+1}$ is the solution to the optimization problem (17). The second inequality follows since $\mathtt{KL}(\cdot,\cdot) \geqslant 0$.

If $\log \int \frac{\zeta_t(\mathbf{X})}{\mathbb{P}_x(\phi(\mathbf{X}) \geqslant \gamma_t)} dQ_{t+1} \leqslant 0$, then,

$$\log \frac{\int \mathbb{I}(\phi(\mathbf{X}),\gamma_t)^+ dQ_{\theta_{t+1}}}{\int \mathbb{I}(\phi(\mathbf{X}),\gamma_t)^+ d\mathbb{P}_x} \geqslant 0 \quad \Rightarrow \frac{\int \mathbb{I}(\phi(\mathbf{X}),\gamma_t)^+ dQ_{\theta_{t+1}}}{\int \mathbb{I}(\phi(\mathbf{X}),\gamma_t)^+ d\mathbb{P}_x} \geqslant 1$$

$$\Rightarrow \int \mathbb{I}(\phi(\mathbf{X}),\gamma_t)^+ dQ_{\theta_{t+1}} \geqslant \int \mathbb{I}(\phi(\mathbf{X}),\gamma_t)^+ d\mathbb{P}_x$$

$$\Rightarrow Q_{\theta_{t+1}}(\phi(\mathbf{X}) \geqslant \gamma_t) \geqslant \mathbb{P}_x(\phi(\mathbf{X}) \geqslant \gamma_t). \qquad (23)$$

The above inequality implies that the update rule (18) will monotonically improve the concentration of the probability measure of the surrogate measure on the event $\{\phi(\mathbf{X}) \geqslant \gamma_t\}$. This result is quite promising, since our primary goal is to find a surrogate measure that has a more likely presence in the extreme region relative to the original measure $\mathbb{P}_x$. A temporal improvement in the probability of the extreme region $w.r.t.$ to the surrogate distribution at each time step will effectively reduce the variance of the quantile estimates in the long run and thus positively tighten the entire estimation process. Although inequality guarantees an improvement in likelihood, it does not provide a precise measure of the extent of this improvement. However,

a naive estimate of the improvement can be obtained using the Bretagnolle-Huber inequality. In fact,

$$\left\|Q_{\theta_{t+1}} - Q_{t+1}\right\|_{\mathrm{TV}} \leqslant \sqrt{1 - \exp\left(-\mathtt{KL}(Q_{t+1}, Q_{\theta_{t+1}})\right)} \quad \text{(Bretagnolle–Huber inequality)}$$

$$\Rightarrow |Q_{\theta_{t+1}}\left(\phi(\mathbf{X}) \geqslant \gamma_t\right) - Q_{t+1}\left(\phi(\mathbf{X}) \geqslant \gamma_t\right)| \leqslant \left\|Q_{\theta_{t+1}} - Q_{t+1}\right\|_{\mathrm{TV}} \leqslant \sqrt{1 - \exp\left(-\mathtt{KL}(Q_{t+1}, Q_{\theta_{t+1}})\right)}$$

$$\Rightarrow Q_{\theta_{t+1}}\left(\phi(\mathbf{X}) \geqslant \gamma_t\right) \geqslant Q_{t+1}\left(\phi(\mathbf{X}) \geqslant \gamma_t\right) - \sqrt{1 - \exp\left(-\mathtt{KL}(Q_{t+1}, Q_{\theta_{t+1}})\right)}$$

$$\Rightarrow Q_{\theta_{t+1}}\left(\phi(\mathbf{X}) \geqslant \gamma_t\right) \geqslant 1 - \sqrt{1 - \exp\left(-\mathtt{KL}(Q_{t+1}, Q_{\theta_{t+1}})\right)} \tag{24}$$

The total variation distance is defined as $\|P - Q\|_{TV} := \sup_{A \in \mathcal{F}} |P(A) - Q(A)|$ with $P$ and $Q$ be probability measures on $(S, \mathcal{F})$. Here, $\mathtt{KL}(Q_{t+1}, Q_{\theta_{t+1}})$ is the approximation error incurred while projecting the zero-variance surrogate measure $Q_{t+1}$ onto the probability space $\mathcal{F}$.

**Lemma 2.** *Let $P$ and $Q$ be two probability measures with $P \ll \nu$ and $Q \ll \nu$ and $(dQ/d\nu)(x) \geqslant \frac{1}{v_Q^2}$, $\forall x$ for some $v_Q > 0$. Then we have*

$$KL(P, Q) \leqslant v_Q^2 \int \left(\frac{dP}{d\nu} - \frac{dQ}{d\nu}\right)^2 d\nu$$

*Proof.*

$$\mathtt{KL}(f, g) = \int \log \frac{dP/d\nu}{dQ/d\nu} dP \leqslant \log \int \frac{dP/d\nu}{dQ/d\nu} dP \quad \text{(Jensen's inequality)}$$

$$= \log \int \frac{(dP/d\nu)^2}{dQ/d\nu} d\nu \leqslant \int \frac{(dP/d\nu)^2}{dQ/d\nu} d\nu - 1 \quad (\because \log x \leqslant x - 1, \forall x > 0)$$

$$= \int \frac{(dP/d\nu)^2}{dQ/d\nu} d\nu - \int \frac{(dQ/d\nu)^2}{dQ/d\nu} d\nu = \int \frac{(dP/d\nu)^2 - (dQ/d\nu)^2}{dQ/d\nu} d\nu$$

$$= \int \frac{(dP/d\nu)^2 + (dQ/d\nu)^2 - 2(dQ/d\nu)^2 - 2(dP/d\nu)(dQ/d\nu) + 2(dP/d\nu)(dQ/d\nu)}{dQ/d\nu} d\nu$$

$$= \int \frac{(dP/d\nu - dQ/d\nu)^2}{dQ/d\nu} d\nu - 2 \int \frac{(dQ/d\nu)^2}{dQ/d\nu} d\nu + 2 \int \frac{(dP/d\nu)(dQ/d\nu)}{dQ/d\nu} d\nu = \int \frac{(dP/d\nu - dQ/d\nu)^2}{dQ/d\nu} d\nu$$

$$\leqslant v_Q^2 \int (dP/d\nu - dQ/d\nu)^2 d\nu$$

$\square$

We need the following result from (Zeevi & Meir, 1997),(Petersen, 1983) and (Rana, 2002) regarding the density of the continuously differentiable functions in the space of square integrable functions.

**Lemma 3.** *(Theorem 8.7.10 of (Rana, 2002)) Let $q \in C\left(\mathbb{R}^d\right)$ with $q \geqslant 0$, and $\int q(x)dx = 1$. We define mollifier $q_\sigma(x) = \sigma^{-d}q(\sigma^{-1}x)$ where $\sigma > 0$. Then for any $f \in C\left(\mathbb{R}^d\right)$ with $\int f^2(x)dx < \infty$, we have the following*

$$\int \left(\int q_\sigma(x - y)f(y)dy - f(x)\right)^2 dx \to 0 \text{ as } \sigma \downarrow 0 \tag{25}$$

**Lemma 4.** *Let $Q_\theta$ be an NEF measure. Then*

$$\int (dQ_\theta/d\nu)^2 d\nu \leqslant \mathcal{C}_\theta < \infty.$$

*Proof.* Let $(dQ_\theta/d\nu)(x) = h(x)\exp(\theta^\top\Gamma(x) - K(\theta))$ then

$$
\begin{aligned}
\int (dQ_\theta/d\nu)^2 d\nu &= \int h^2(x)\exp(2\theta^\top\Gamma(x) - 2K(\theta))dx \\
&= \int h(x)\exp(\theta^\top\Gamma(x) - K(\theta))h(x)\exp(\theta^\top\Gamma(x) - K(\theta))dx \\
&\leqslant \mathcal{C}_\theta \int h(x)\exp(\theta^\top\Gamma(x) - K(\theta))dx \quad \left[\text{where } \mathcal{C}_\theta = \sup_x \left(h(x)\exp(\theta^\top\Gamma(x) - K(\theta))\right)\right] \\
&= \mathcal{C}_\theta < \infty.
\end{aligned}
$$

$\square$

**Lemma 5.** *Assume $\bar{\mathcal{C}}_\Theta = \sup_\theta C_\theta < \infty$. Let $P$ be a probability measure with $dP/d\nu$ continuous and bounded. Then for a given arbitrary $\epsilon_1 > 0$, there exists an $L \in \mathcal{Q}_\Theta$ and $\sigma > 0$ such that ,*

$$
\int \left( \int q_\sigma(x-y)\frac{dP}{d\nu}(y)dy - \frac{dL}{d\nu}(x) \right)^2 dx \leqslant \epsilon_1 + \bar{\mathcal{C}}_\Theta - 1, \text{ where } q \in \mathcal{Q}_\Theta \text{ and } q_\sigma(x) = \sigma^{-d}\frac{dq}{d\nu}(\sigma^{-1}x)
$$

*Proof.* Let $f = dP/d\nu$ and $\bar{f}(x) = \int q_\sigma(x-y)f(y)dy$. Since $f, q_\sigma$ are continuous and Riemann integrable, it follows that $\bar{f} \in \overline{\text{CONV}}(\mathcal{Q}_\Theta)$, where $\overline{\text{CONV}}$ represents closure of convex hull of $\mathcal{Q}_\Theta$. Hence, for $\epsilon_1 > 0$, we have

$$
\int (\bar{f}(x) - f_c(x))^2 dx \leqslant \epsilon_1 \tag{26}
$$

where $f_c(x) = \sum_{k=1}^m c_k \bar{g}_k(x)$ with $\bar{g}_k \in \mathcal{Q}_\Theta, c_k \geqslant 0, \sum_{k=1}^m c_k = 1$, for some sufficiently large $m$. Let $\bar{g}$ be random function drawn from the set $\{\bar{g}_1, \cdots, \bar{g}_m\}$ with $\mathbb{P}_{\bar{g}}(\bar{g} = \bar{g}_k) = c_k$. Then $\mathbb{E}_{\bar{g}}[\bar{g}] = f_c$, and

$$
\mathbb{E}_{\bar{g}} \left[ \int (\bar{g}(x) - f_c(x))^2 dx \right] = \mathbb{E}_{\bar{g}} \left[ \int \bar{g}^2(x)dx \right] - \int f_c^2(x)dx \leqslant \bar{\mathcal{C}}_\Theta - \int f_c^2(x)dx \tag{27}
$$

Hence, there exists a $g \in \{\bar{g}_1, \cdots, \bar{g}_m\}$ such that

$$
\int (g(x) - f_c(x))^2 dx \leqslant \bar{\mathcal{C}}_\Theta - \int f_c^2(x)dx \tag{28}
$$

Furthermore,

$$
\begin{aligned}
\int (\bar{f}(x) - g(x))^2 dx &\leqslant \int (\bar{f}(x) - f_c(x))^2 dx + \int (g(x) - f_c(x))^2 dx \quad \text{(by Triangle Inequality)} \\
&\leqslant \epsilon_1 + \bar{\mathcal{C}}_\Theta - \int f_c^2(x)dx \quad \text{(From Eqs. (26) and (28))} \\
&\leqslant \epsilon_1 + \bar{\mathcal{C}}_\Theta - \left( \int f_c(x)dx \right)^2 \quad \text{( by Jensen's Inequality)} \\
&= \epsilon_1 + \bar{\mathcal{C}}_\Theta - 1
\end{aligned}
$$

$\square$

The following result offers a lower bound for the probability of the event $\{\phi(\mathbf{X}) \geqslant \gamma_t\}$ with respect to the surrogate measure $Q_{\theta_t}$.

**Theorem 1.** *Let $\upsilon_\Theta = \inf_{Q_\theta \in \mathcal{Q}_\Theta} \sup_x (dQ_\theta/d\nu)(x) > 0$. Assume $dQ_{t+1}/d\nu$ is continuous everywhere except at countable number of points. Then, for $t \geqslant 0$,*

$$
Q_{\theta_{t+1}}(\phi(\mathbf{X}) \geqslant \gamma_t) \geqslant 1 - \sqrt{1 - \exp(\upsilon_\Theta^2(1 - \epsilon - \bar{\mathcal{C}}_\Theta))}.
$$

*Proof.* From 24 we have

$$Q_{\theta_{t+1}}\left(\phi(\mathbf{X}) \geqslant \gamma_t\right) \geqslant 1 - \sqrt{1 - \exp\left(-\mathtt{KL}(Q_{t+1}, Q_{\theta_{t+1}})\right)} \tag{29}$$

Now, we upper bound $\mathtt{KL}(Q_{t+1}, Q_{\theta_{t+1}})$ as follows - From Lemma 3, we know that, for $\epsilon > 0$, there exists a $\sigma_{\frac{\epsilon}{2}} > 0$ such that

$$\int \left(\int q_{\sigma_{\frac{\epsilon}{2}}}(x-y)\frac{dQ_{t+1}}{d\nu}(y)dy - \frac{dQ_{t+1}}{d\nu}(x)\right)^2 dx < \frac{\epsilon}{2} \tag{30}$$

Also from Lemma 5, there exists a $L \in \mathcal{Q}_\Theta$ such that ,

$$\int \left(\int q_{\sigma_{\frac{\epsilon}{2}}}(x-y)\frac{Q_{t+1}}{d\nu}(y)dy - \frac{dL}{d\nu}(x)\right)^2 dx \leqslant \frac{\epsilon}{2} + \bar{\mathcal{C}}_\Theta - 1, \tag{31}$$

Now, from Lemma 2, we have the following.

$$\mathtt{KL}(Q_{t+1}, Q_{\theta_{t+1}}) \leqslant \mathtt{KL}\left(Q_{t+1}, L\right) \leqslant \upsilon_L^2 \int ((dQ_{t+1}/d\nu)(x) - (dL/d\nu)(x))^2 dx$$

$$\leqslant \upsilon_L^2 \left(\int \left(\int q_{\sigma_{\frac{\epsilon}{2}}}(x-y)\frac{dQ_{t+1}}{d\nu}(y)dy - \frac{dQ_{t+1}}{d\nu}(x)\right)^2 dx + \int \left(\int q_{\sigma_{\frac{\epsilon}{2}}}(x-y)\frac{dQ_{t+1}}{d\nu}(y)dy - \frac{dL}{d\nu}(x)\right)^2 dx\right)$$

$$\text{(Using Triangle Inequality)}$$

$$\leqslant \upsilon_\Theta^2 \left(\frac{\epsilon}{2} + \frac{\epsilon}{2} + \bar{\mathcal{C}}_\Theta - 1\right) \quad \text{(From Eqs. (30 and 31))}$$

$$= \upsilon_\Theta^2(\epsilon + \bar{\mathcal{C}}_\Theta - 1). \tag{32}$$

Finally, substituting eq.(32) in eq.(29) we obtain

$$Q_{\theta_{t+1}}\left(\phi(\mathbf{X}) \geqslant \gamma_t\right) \geqslant 1 - \sqrt{1 - \exp\left(\upsilon_\Theta^2(1 - \epsilon - \bar{\mathcal{C}}_\Theta)\right)} \tag{33}$$

$\square$

The above result provides a lower bound for the probability of the event $\{\phi(\mathbf{X}) \geqslant \gamma_t\}$ with respect to the surrogate measure $Q_{\theta_t}$, expressed in terms of the space of parameterized NEF measures $\mathcal{Q}_\theta$. It is important to note that $\bar{\mathcal{C}}_\Theta$ and $\vartheta_\Theta$ represent the upper and lower bounds of the Radon-Nikodyn derivatives for the NEF measures in $\mathcal{Q}_\Theta$. Note that as $\bar{\mathcal{C}}_\Theta$ becomes larger and $\vartheta_\Theta$ is closer to 0, then $Q_{\theta_{t+1}}\left(\phi(\mathbf{X}) \geqslant \gamma_t\right)$ becomes closer to 0 indicating that as the Radon-Nikodyn derivates are narrow, the approximation is poor. As $\bar{\mathcal{C}}_\Theta$ increases and $\vartheta_\Theta$ approaches zero, the probability $Q_{\theta_{t+1}}(\phi(\mathbf{X}) \geqslant \gamma_t)$ tends toward 0. This outcome suggests that when the Radon-Nikodyn derivatives become increasingly narrow—meaning the surrogate parametrized space is more concentrated—the approximation of the true probability becomes poorer. In other words, as the surrogate measure becomes more "localized" or "precise" ,it may fail to adequately approximate the true distribution, leading to a less reliable estimate of the probability of the event $\{\phi(\mathbf{X}) \geqslant \gamma_t\}$. This highlights the importance of ensuring that the Radon-Nikodyn derivatives maintain a sufficiently wide range for the approximation to remain stable and accurate.

An improved bound on the probability of extreme events $Q_{\theta_{t+1}}(\phi(\mathbf{X}) \geqslant \gamma_t)$ can be obtained by bounding the total variation distance between the probability measures $Q_{\theta_{t+1}}$ and $Q_{t+1}$, respectively. From (Zhang, 2007; Sason, 2015) (specifically, Eq. (7) of (Sason, 2015)), we have the following expression for the total variation distance between the probability measures induced by $Q_{\theta_{t+1}}$ and $Q_{t+1}$

$$\left\|Q_{\theta_{t+1}} - Q_{t+1}\right\|_{\mathrm{TV}} = \mathbb{E}_{\theta_{t+1}}\left[\left|1 - \exp\left(-\log\frac{dQ_{\theta_{t+1}}}{dQ_{t+1}}(\mathbf{X})\right)\right|\right],$$

where $\frac{dQ_{\theta_{t+1}}}{dQ_{t+1}}$ is the Radon-Nikodym derivative of $Q_{\theta_{t+1}}$ *w.r.t.* $Q_{t+1}$ (note that the Radon-Nikodym derivative is well defined since $Q_{\theta_{t+1}}$ is absolutely continuous *w.r.t.* $Q_{t+1}$, *i.e.*, $Q_{t+1}(A) = 0 \Rightarrow Q_{\theta_{t+1}}(A) = 0, \forall A \in \mathcal{F}$). Then

$$|Q_{\theta_{t+1}}(\phi(\mathbf{X}) \geq \gamma_t) - Q_{t+1}(\phi(\mathbf{X}) \geq \gamma_t)| \leq \left\| Q_{\theta_{t+1}} - Q_{t+1} \right\|_{\mathsf{TV}} = \mathbb{E}_{\theta_{t+1}} \left[ \left| 1 - \exp\left( -\log \frac{dQ_{\theta_{t+1}}}{dQ_{t+1}}(\mathbf{X}) \right) \right| \right]$$

$$\Rightarrow |Q_{\theta_{t+1}}(\phi(\mathbf{X}) \geq \gamma_t) - Q_{t+1}(\phi(\mathbf{X}) \geq \gamma_t)| \leq \mathbb{E}_{\theta_{t+1}} \left[ \left| 1 - \exp\left( -\log \frac{dQ_{\theta_{t+1}}}{dQ_{t+1}}(\mathbf{X}) \right) \right| \right]$$

$$\Rightarrow |Q_{\theta_{t+1}}(\phi(\mathbf{X}) \geq \gamma_t) - Q_{t+1}(\phi(\mathbf{X}) \geq \gamma_t)| \leq \mathbb{E}_{\theta_{t+1}} \left[ \left| 1 - \exp\left( -\log \frac{(dQ_{\theta_{t+1}}/d\nu)(\mathbf{X})}{\frac{(d\mathbb{P}_x/d\nu)(\mathbf{X})\mathbb{I}(\phi(\mathbf{X}),\gamma_{t+1})^+}{\mathbb{E}_{\mathbb{P}_x}[\mathbb{I}(\phi(\mathbf{X}),\gamma_{t+1})^+]}} \right) \right| \right]$$

$$\text{(Follows from Eq. (16))}$$

$$= \mathbb{E}_{\theta_{t+1}} \left[ \left| 1 - \exp\left( -\log \frac{\mathbb{E}_{\mathbb{P}_x}[\mathbb{I}(\phi(\mathbf{X}),\gamma_{t+1})^+] (dQ_{\theta_{t+1}}/d\nu)(\mathbf{X})}{(d\mathbb{P}_x/d\nu)(\mathbf{X})} \right) \right| \right]. \tag{34}$$

**Proposition 3.** *For $t \geq 0$, and $\delta \in [0,1]$, if $1 - \delta \leq \frac{\zeta_t(x)\mathbb{I}(\phi(x),\gamma_t)^+}{\mathbb{P}_x(\phi(\mathbf{X}) \geq \gamma_t)} \leq 1 + \delta, \forall x$, then*

$$Q_{\theta_{t+1}}(\phi(\mathbf{X}) \geq \gamma_t) \geq 1 - \delta$$

*Proof.* Given that for $\delta \in [0,1]$,

$$1 - \delta \leq \frac{\zeta_t(x)\mathbb{I}(\phi(x),\gamma_t)^+}{\mathbb{P}_x(\phi(\mathbf{X}) \geq \gamma_t)} \leq 1 + \delta, \forall x$$

$$\Rightarrow \log(1 - \delta) \leq -\log\left( \frac{\mathbb{P}_x(\{\phi(\mathbf{X}) \geq \gamma_t\})}{\zeta_t(x)\mathbb{I}(\phi(x),\gamma_t)^+} \right) \leq \log(1 + \delta), \forall x$$

$$\Rightarrow (1 - \delta) \leq \exp\left( -\log \frac{\mathbb{P}_x(\{\phi(\mathbf{X}) \geq \gamma_t\})}{\zeta_t(x)\mathbb{I}(\phi(x),\gamma_t)^+} \right) \leq (1 + \delta), \forall x$$

$$\Rightarrow -\delta \leq 1 - \exp\left( -\log \frac{\mathbb{P}_x(\{\phi(\mathbf{X}) \geq \gamma_t\})}{\zeta_t(x)\mathbb{I}(\phi(x),\gamma_t)^+} \right) \leq \delta, \forall x$$

$$\Rightarrow \left| 1 - \exp\left( -\log \frac{\mathbb{P}_x(\{\phi(\mathbf{X}) \geq \gamma_t\})}{\zeta_t(x)\mathbb{I}(\phi(x),\gamma_t)^+} \right) \right| \leq \delta, \forall x. \tag{35}$$

Substituting Eq.(35) in E.q.(34), we get,

$$|Q_{\theta_{t+1}}(\phi(\mathbf{X}) \geq \gamma_t) - Q_{t+1}(\phi(\mathbf{X}) \geq \gamma_t)| \leq \mathbb{E}_{\theta_{t+1}} \left[ \left| 1 - \exp\left( -\log \frac{\mathbb{P}_x(\{\phi(\mathbf{X}) \geq \gamma_t\})}{\zeta_t(\mathbf{X})\mathbb{I}(\phi(\mathbf{X}),\gamma_t)^+} \right) \right| \right] \leq \delta$$

$$\Rightarrow Q_{\theta_{t+1}}(\phi(\mathbf{X}) \geq \gamma_t) \geq 1 - \delta \text{ since } Q_{t+1}(\phi(\mathbf{X}) \geq \gamma_t) = 1. \tag{36}$$

$\square$

### 3.2 Algorithm (Stochastic Approximation Version)

The key challenge is to estimate the measure parameter $\theta_t$ efficiently. This question is relevant since the computation of the true values of this parameter (Eq. (21) is intractable, *i.e.*, hard to compute (specifically, due to the implicit hardness involved in computing $\int dQ_{\theta_t}$). Additionally, maintaining an incremental, online, single-pass approach is highly desirable from a computational complexity standpoint, especially for real-time applications where data arrives sequentially. Therefore, to estimate them, we employ an additional stochastic approximation recursion to track the tunable parameter $\theta_t$ as follows:

$$\eta_{t+1} = \eta_t + \beta_t \zeta_t(\mathbf{Y}_{t+1})(\mathbb{I}(\phi(\mathbf{Y}_{t+1}),\gamma_t)^+ \Gamma(\mathbf{Y}_{t+1}) - \eta_t \mathbb{I}^-(\phi(\mathbf{Y}_{t+1}),\gamma_t)), \text{ where } \mathbf{Y}_{t+1} \sim Q_{\theta_t} \tag{37}$$

Then $\theta_t$ is estimated as $\theta_t = m^{-1}(\eta_t)$. We prove in Lemma 6 that iterates $\eta_t$ indeed track the ideal $m(\theta_t)$ and $m^{-1}$ can typically be computed in time that is proportional to a polynomial in $d$. For a given $\gamma_t$, note that the

ideal $Q_{\theta_t}$ identifies the surrogate measure that has strong probabilistic support in the region $\{\phi(\mathbf{X}) \geqslant \gamma_t\}$. This requires that the $\theta_t$ have to be estimated with sufficient accuracy before $\gamma_t$ can drift significantly. This requirement regarding the asynchronicity of the convergence rates of the stochastic recursions can be achieved by following a multi-timescale stochastic approximation framework (Borkar, 1997; 2008). In this framework, we maintain the stochastic recursion of $\gamma_t$ along a slower time-scale (lower convergence rate) relative to $\eta_t$ which are maintained along a faster time-scale (faster convergence rate). This setup can be interpreted as $\gamma_t$ being quasi-static, while $\eta_t$ converges close to $m(\theta_t)$ with respect to the static value of $\gamma_t$. The continuous nature of this coupled updating process prevents large discrete changes, thus reducing variance and contributing to the stability of the estimates (Konda & Tsitsiklis, 2003).

Observe that at each iteration $t$, if the sample $\mathbf{X}_{t+1}$ is drawn using the surrogate measure $Q_{\theta_t}$, then one might fall prey to over-compliance *i.e.*, scenarios where a substantial fraction of the samples belong to the region $\{\phi(\mathbf{X}) \geqslant \gamma_t\}$. This is because the measure $Q_{\theta_t}$ is pursued to maintain strong probabilistic support in the region $\{\phi(\mathbf{X}) \geqslant \gamma_t\}$. This is quite synonymous with the earlier scenario, where we had a considerable number of samples originating from the region $\{\phi(\mathbf{X}) \leqslant \gamma_t\}$ when samples were sampled using $\mathbb{P}_x$. Therefore, to achieve a balance, we follow a randomized approach, where at each iteration $t$, we obtain the sampling measure $\widehat{Q}_t$ by choosing between the original measure $\mathbb{P}_x$ and the surrogate measure $Q_{\theta_t}$ based on an independent Bernoulli trial with parameter $\lambda \in [0,1]$ (fixed a priori), *i.e.*, $\mathbb{P}(\texttt{choosing } \mathbb{P}_x) = \lambda$ and $\mathbb{P}(\texttt{choosing } Q_{\theta_t}) = 1 - \lambda$. One can indeed interpret $\widehat{Q}_t$ as a mixture measure, i.e., $\widehat{Q}_t = \lambda \mathbb{P}_x + (1 - \lambda)Q_{\theta_t}$. In fact, for an arbitrary Borel set $A$ and $\widehat{X} \sim \widehat{Q}_t$, we have

$$\widehat{Q}_t(\widehat{X} \in A) = \int_A d\widehat{Q}_t = \int_A \lambda d\mathbb{P}_x + (1 - \lambda) \int_A dQ_{\theta_t}$$
$$= \mathbb{P}(\texttt{choosing } \mathbb{P}_x)\mathbb{P}_x(\widehat{X} \in A) + \mathbb{P}(\texttt{choosing } Q_{\theta_t})Q_{\theta_t}(\widehat{X} \in A).$$

Now to esitmate $\texttt{CVaR}$, we consider the $\texttt{CVaR}$ definition from Eq. (2),

$$\texttt{CVaR}_\rho(\mathbb{P}_x) = \frac{\int_{\texttt{VaR}_\rho(\mathbb{P}_x)}^\infty \phi(\mathbf{X})d\mathbb{P}_x}{\mathbb{P}_x(\phi(\mathbf{X}) \geqslant \texttt{VaR}_\rho(\mathbb{P}_x))}$$
$$= \frac{\int \phi(\mathbf{X})\mathbf{I}(\phi(\mathbf{X}), \texttt{VaR}_\rho(\mathbb{P}_x))^+ d\mathbb{P}_x}{\mathbb{P}_x(\phi(\mathbf{X}) \geqslant \texttt{VaR}_\rho(\mathbb{P}_x))}$$
$$= \int \phi(\mathbf{X})d\mathbb{P}_x^{\texttt{CVaR}}, \text{ where } \frac{d\mathbb{P}_x^{\texttt{CVaR}}}{d\nu}(x) = \frac{\mathbf{I}(\phi(x), \texttt{VaR}_\rho(\mathbb{P}_x))^+(d\mathbb{P}_x/d\nu)(x)}{\mathbb{P}_x(\phi(\mathbf{X}) \geqslant \texttt{VaR}_\rho(\mathbb{P}_x))} \tag{38}$$

By Theorem 2, the iterates $\gamma_t$ converges to $\texttt{VaR}_\rho(\mathbb{P}_x)$ as $t \to \infty$. Hence, by continuity of probability measures, we get,

$$Q_t = \frac{(d\mathbb{P}_x/d\nu)(x)\mathbf{I}^+(\phi(x), \gamma_t)}{\mathbb{P}_x(\phi(\mathbf{X}) \geqslant \gamma_t)} \to \frac{\mathbf{I}(\phi(x), \texttt{VaR}_\rho(\mathbb{P}_x))^+(d\mathbb{P}_x/d\nu)(x)}{\mathbb{P}_x(\phi(\mathbf{X}) \geqslant \texttt{VaR}_\rho(\mathbb{P}_x))} = \frac{d\mathbb{P}_x^{\texttt{CVaR}}}{d\nu}(x) \text{ as } t \to \infty. \tag{39}$$

This implies that if $Q_{\theta_t} = \min_\theta \texttt{KL}(Q_t, Q_\theta)$ which approximates $Q_t$ as closely as possible converges as $t \to \infty$, then $\lim_{t\to\infty} Q_{\theta_t}$ is a good approximation of $\mathbb{P}_x^{\texttt{CVaR}}$. Hence,

$$\lim_{t\to\infty} \int \phi(\mathbf{X})dQ_{\theta_t} \approx \texttt{CVaR}_\rho(\mathbb{P}_x). \tag{40}$$

Therefore, one can estimate $\texttt{CVaR}_\rho(\mathbb{P}_x)$ as follows:

$$\bar{\eta}_{t+1} = \frac{1}{(t+1)^c}(t^c\bar{\eta}_t + \phi(\mathbf{Y}_{t+1})) \text{ with } \mathbf{Y}_{t+1} \sim Q_{\theta_t} \text{ and } c \geqslant 1. \tag{41}$$

This recursive formula involves iteratively updating the estimate by incorporating new samples from the distribution $Q_{\theta_t}$ and averaging them. The parameter $c$ controls the weight given to new samples versus previous estimates. The recursion described above involves repeatedly aggregating the samples $\phi(\mathbf{Y}_t)$ followed by their averaging, which incrementally leads to a more accurate estimate of the $\texttt{CVaR}$ as $t \to \infty$.

### 3.3 Gaussian Version

The Gaussian (or normal) distribution is in fact one of the most common among the NEF family. For the Gaussian case with the parameterization given by $\theta = (\mu, \Sigma)^\top$ with $\mu \in \mathbb{R}^d$, $\Sigma \in S_{++}^d$, we have the following closed-form expression for the surrogate Gaussian:

$$\partial_\mu \mathbb{E}_{\theta_t} \left[ \zeta_t(\mathbf{X}) \mathbb{I}(\phi(\mathbf{X}), \gamma_t)^+ \left( \frac{1}{2}(\mathbf{X} - \mu)^\top \Sigma^{-1}(\mathbf{X} - \mu) + \frac{1}{2} \log\left(2\pi|\Sigma|\right) \right) \right] = 0$$

$$\Rightarrow \Sigma^{-1} \mathbb{E}_{\theta_t} \left[ \zeta_t(\mathbf{X}) \mathbb{I}(\phi(\mathbf{X}), \gamma_t)^+ (\mathbf{X} - \mu) \right] = 0$$

$$\Rightarrow \mathbb{E}_{\theta_t} \left[ \zeta_t(\mathbf{X}) \mathbb{I}(\phi(\mathbf{X}), \gamma_t)^+ (\mathbf{X} - \mu) \right] = 0 \quad \text{(since } \Sigma^{-1} \text{ is full rank)}$$

$$\Rightarrow \mu_{t+1} = \frac{\mathbb{E}_{\theta_t} \left[ \zeta_t(\mathbf{X}) \mathbb{I}(\phi(\mathbf{X}), \gamma_t)^+ \mathbf{X} \right]}{\mathbb{E}_{\theta_t} \left[ \zeta_t(\mathbf{X}) \mathbb{I}(\phi(\mathbf{X}), \gamma_t)^+ \right]}. \tag{42}$$

Similarly, $\partial_\Sigma \mathbb{E}_{\theta_t} \left[ \zeta_t(\mathbf{X}) \mathbb{I}(\phi(\mathbf{X}), \gamma_t)^+ \left( \frac{1}{2}(\mathbf{X} - \mu)^\top \Sigma^{-1}(\mathbf{X} - \mu) + \frac{1}{2} \log\left(2\pi|\Sigma|\right) \right) \right] = 0$

$$\Rightarrow \mathbb{E}_{\theta_t} \left[ \zeta_t(\mathbf{X}) \mathbb{I}(\phi(\mathbf{X}), \gamma_t)^+ \left( -\frac{1}{2}\Sigma^{-T}(\mathbf{X} - \mu)(\mathbf{X} - \mu)^\top \Sigma^{-T} + \frac{1}{2}\Sigma^{-T} \right) \right] = 0$$

$$\Rightarrow \mathbb{E}_{\theta_t} \left[ \zeta_t(\mathbf{X}) \mathbb{I}(\phi(\mathbf{X}), \gamma_t)^+ \left( -(\mathbf{X} - \mu)(\mathbf{X} - \mu)^\top \Sigma^{-T} + I \right) \right] = 0 \quad \text{(since } \Sigma^{-T} \text{ is full rank)}$$

$$\Rightarrow \Sigma_{t+1} = \frac{\mathbb{E}_{\theta_t} \left[ \zeta_t(\mathbf{X}) \mathbb{I}(\phi(\mathbf{X}), \gamma_t)^+ \left( (\mathbf{X} - \mu)(\mathbf{X} - \mu)^\top \right) \right]}{\mathbb{E}_{\theta_t} \left[ \zeta_t(\mathbf{X}) \mathbb{I}(\phi(\mathbf{X}), \gamma_t)^+ \right]}. \tag{43}$$

In Figure(4), we illustrate the evolution of surrogate Gaussian distributions by following the update rules given by Eqs.(42) and (43).

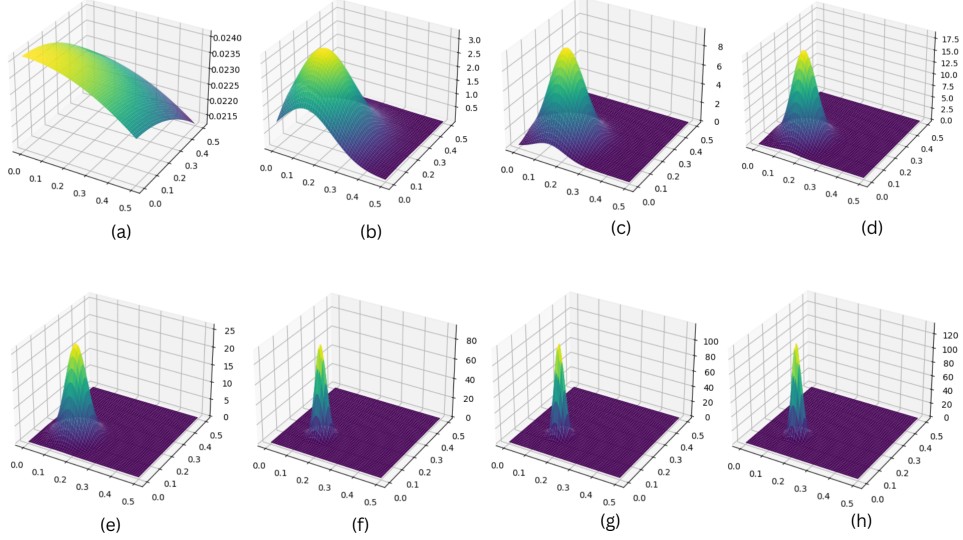

Figure 4: Illustration of the approximate optimal surrogate distribution obtained after projecting the optimal surrogate distribution to the space of parametrized Gaussian distributions collected over 100 iterations each.

Here, we state some necessary assumptions:

**Assumption (A1):** The step-size schedules $\{\alpha_t\}_{t\in\mathbb{N}}$ and $\{\beta_t\}_{t\in\mathbb{N}}$ are real-valued, positive, deterministic and pre-determined sequences and they satisfy

$$\sum_{t\in\mathbb{N}} \left( \alpha_t^2 + \beta_t^2 \right) < \infty, \quad \sum_{t\in\mathbb{N}} \alpha_t = \sum_{t\in\mathbb{N}} \beta_t = \infty, \quad \lim_{t\to\infty} \frac{\alpha_t}{\beta_t} = 0.$$

---

**Algorithm 1** [Extreme quantile and superquantile estimation algorithm]

---
1: **Input parameters:** $\alpha_t, \beta_t > 0, \lambda, \rho \in [0, 1], T \in \mathbb{N}$;
2: **Initialize** $\gamma_0 = -\infty, t = 0, \eta_0$;
3: **while** $t < T$ **do**
4: $\quad \hat{Q}_t \sim \texttt{Bernoulli}(\{\mathbb{P}_x, Q_{\theta_t}\}, \lambda)$, where $\theta_t = m^{-1}(\eta_t)$;
5: $\quad \gamma_{t+1} = \gamma_t + \alpha_t \zeta_t(\hat{\mathbf{X}}_{t+1})((1-\rho)\mathbb{I}(\phi(\hat{\mathbf{X}}_{t+1}), \gamma_t)^+ - \rho\mathbb{I}(\phi(\hat{\mathbf{X}}_{t+1}), \gamma_t)^-)$, where $\hat{\mathbf{X}}_{t+1} \sim \hat{Q}_t$;
6: $\quad \eta_{t+1} = \eta_t + \beta_t \zeta_t(\mathbf{Y}_{t+1})(\mathbb{I}(\phi(\mathbf{Y}_{t+1}), \gamma_t)^+ \Gamma(\mathbf{Y}_{t+1}) - \eta_t \mathbb{I}(\phi(\mathbf{Y}_{t+1}), \gamma_t)^-)$, where $\mathbf{Y}_{t+1} \sim Q_{\theta_t}$;
7: $\quad \bar{\eta}_{t+1} = \frac{1}{(t+1)^c}(t^c \bar{\eta}_t + \phi(\mathbf{Y}_{t+1}))$;
8: $\quad t = t + 1$;
9: **end while**
10: **Return** $\widehat{\texttt{VaR}}_\rho(\phi) = \gamma_T, \widehat{\texttt{CVaR}}_\rho(\phi) = \bar{\eta}_T$;

---

Examples of such step sizes are $\alpha_t = \frac{1}{t}, \beta_t = \frac{1}{t^c}, c \in (1/2, 1); \alpha_t = \frac{1}{1+t\log t}, \beta_t = \frac{1}{t}$ and so on.

**Assumption (A2):** The given measure $\mathbb{P}_x$ satisfies $\mathbb{E}_{\mathbb{P}_x}[|\Gamma(\mathbf{X})|] < \infty$.

### 3.4 Convergence Analysis

The proposed algorithm is a two-timescale stochastic approximation algorithm where there exists a bilateral coupling between the stochastic recursions defined in steps 5,6 and 7 of Algorithm 1. It is a known fact that the asymptotic behaviour of the multi-timescale approximation algorithm is dependent on the relationship between the step-sizes of the individual recursions (assuming all the regularity conditions are satisfied). Note that the step-size schedules $\{\alpha_t\}_{t\in\mathbb{N}}$ and $\{\beta_t\}_{t\in\mathbb{N}}$ satisfy $\frac{\alpha_t}{\beta_t} \to 0$, which implies that the step-size sequence $\{\alpha_t\}_{t\in\mathbb{N}}$ decays to 0 relatively faster than the sequence $\{\beta_t\}_{t\in\mathbb{N}}$. This disparity in terms of the decay rate of the step sizes results in the emergence of an asynchronous and coherent convergence behavior asymptotically (Borkar, 1997), with the threshold sequence $\{\gamma_t\}$ converging slower relative to the sequence $\{\eta_t\}_{t\in\mathbb{N}}$. The rationale being that the decrement term $\zeta_t(\hat{\mathbf{X}}_{t+1})\Delta_t^\gamma(\hat{\mathbf{X}}_{t+1})$ of the $\gamma_t$ recursion (step 5) is weighted with $\alpha_t$, which is order of magnitude smaller compared to that of $\beta_t$ asymptotically, *i.e.* $\{\alpha_t\} \in o(\{\beta_t\})$. This unique pseudo heterogeneity induces multiple perspectives, *i.e.*, when viewed from the faster timescale recursion (recursion controlled by $\beta_t$), the slower timescale recursion (recursion controlled by $\alpha_t$) seems quasi-static ('almost a constant') and while viewed from the slower timescale, the faster timescale recursion seems equilibrated. This intuition is indeed theoretically corroborated in (Borkar, 1997) (or Chapter 6 of (Borkar, 2008)) where the multi-timescale stochastic approximation algorithms are analyzed and shown that the asymptotic dynamics of a two-timescale recursion is equivalent to that of its slowest timescale component with the faster timescale variable replaced by the limit point of the faster timescale recursion obtained by keeping the slower timescale variable quasi-static (also assuming that the faster timescale recursion has a single limit point). So, in this paper, we follow this line of analysis and hence we initially analyze the faster timescale recursion (step 6) assuming that the slower timescale variable $\gamma_t$ is quasi-static. The results are provided in Lemma 6. Further, we analyze the slower timescale recursion (step 5) after replacing $\eta_t$ by their quasistatic limit points (which happen to be unique and finite for each quasi-static variable). The results are provided in Theorem 2. Note that $\bar{\eta}_t$, is independent of the other recursions as it has unidirectional coupling with the other recursions and hence it is analyzed independently in part 2 of following lemma.

Define the filtration $\{\mathcal{F}_t\}_{t\in\mathbb{N}}$, where the $\sigma$-field $\mathcal{F}_t := \sigma(\gamma_i, \eta_i, \bar{\eta}_i, \theta_i, 0 \leqslant i \leqslant t, \hat{\mathbf{X}}_i, \mathbf{Y}_i, 1 \leqslant i \leqslant t)$.

**Lemma 6.** *Let $\eta_0$ be integrable, i.e., $\mathbb{E}[|\eta_0|] < \infty$. Assume $\theta_t \equiv \theta, \forall t$ and $\gamma_t \equiv \gamma, \forall t$ (i.e., quasi-static). Let Assumptions (A1) and (A2) hold. Then, almost surely,*

$$\lim_{t\to\infty} \eta_t = \eta_{|\gamma,\theta} = \frac{\int \mathbb{I}(\phi(\mathbf{Y}), \gamma)^+ \Gamma(\mathbf{Y}) d\mathbb{P}_x}{\int \mathbb{I}(\phi(\mathbf{Y}), \gamma)^+ d\mathbb{P}_x}, \text{ and } \lim_{t\to\infty} \bar{\eta}_t = \int \Gamma(\mathbf{Y}) dQ_\theta.$$

*Proof.* Please refer to Appendix. $\qquad\square$

**Theorem 2.** *Let the learning rates $\{\alpha_t\}_{t \in \mathbb{N}}$ and $\{\beta_t\}_{t \in \mathbb{N}}$ satisfy Assumption (A1). Also, let Assumption (A2) hold. Also let $\alpha_t \in \Omega(1/(t+1)^c)$. Then the stochastic sequences $\{\gamma_t\}_{t \in \mathbb{N}}$, $\{\eta_t\}_{t \in \mathbb{N}}$ and $\{\bar{\eta}_t\}_{t \in \mathbb{N}}$ generated by the Algorithm 1 satisfy*

$$\lim_{t \to \infty} \gamma_t = \textit{VaR}_\rho(\mathbb{P}_x) \ \ \textit{a.s.}, \qquad \lim_{t \to \infty} \eta_t = \frac{\int \mathbb{I}(\phi(\mathbf{Y}), \textit{VaR}_\rho(\mathbb{P}_x))^+ \Gamma(\mathbf{Y}) d\mathbb{P}_x}{\int \mathbb{I}(\phi(\mathbf{Y}), \textit{VaR}_\rho(\mathbb{P}_x))^+ d\mathbb{P}_x} \ \ \textit{a.s.}, and$$

$$\lim_{t \to \infty} \bar{\eta}_t = \int \Gamma(\mathbf{Y}) dQ_{\theta^*} \ \ \textit{a.s.}, \ \ \textit{where } \theta^* = m^{-1}(\lim_{t \to \infty} \eta_t).$$

*Proof.* Please refer to Appendix. $\qquad\qquad\qquad\qquad\qquad\qquad\qquad\qquad\qquad\qquad\qquad\qquad$ $\square$

## 4 Superquantile Optimization Algorithm

**Superquantile Optimization:** Consider the collection of parametrized probability measures given by $\{\mathbb{P}_\omega | \omega \in \mathbb{R}^p\}$. Let $\texttt{CVaR}_\rho(\omega) = \texttt{CVaR}_\rho(\mathbb{P}_\omega)$. In the superquantile optimization problem (Rockafellar et al., 2000), one seeks to obtain the optimal distribution parameter that maximizes the superquantile. This is expressed as follows:

$$\text{Find } \omega^* \in \arg\max_\omega \texttt{CVaR}_\rho(\omega) \tag{44}$$

CVaR optimization focuses on adjusting the parameters of a probability distribution to maximize expected returns in extreme scenarios. This approach ensures that risk management strategies do not only consider the most likely returns, but also the potential gains from extreme events. Unlike traditional methods that prioritize optimizing mean returns, CVaR optimization specifically captures the decision maker's sensitivity to risk. This is particularly beneficial in industries such as finance, insurance, and operations, where the rewards from rare, high-impact events can be significant. The optimization process aids in developing policies and strategies that are robust and capable of capitalizing on low-probability, high-reward situations. Additionally, CVaR optimization enhances scenario planning by offering insights into how portfolios or strategies perform under extreme conditions, ensuring a more thorough evaluation of risk by focusing on the potential upside in rare events.

### 4.1 Related Work

Conditional Value-at-Risk (CVaR) optimization has been a subject of extensive research in recent years, with various approaches proposed to improve its estimation and application. Building upon the foundational work in (Rockafellar et al., 2000), several techniques for CVaR estimation and optimization has come to surface. (Zhang et al., 2020) introduced an adaptive importance sampling method to enhance CVaR optimization efficiency. In (Li & Zhou, 2021) a robust CVaR optimization framework for portfolio selection under distributional uncertainty was introduced. (Chen et al., 2022) leveraged deep learning techniques for CVaR estimation in high-dimensional problems. (Kalogerias & Powell, 2023) presented a model-free reinforcement learning approach for CVaR optimization in Markov decision processes. Wang and Liu (Wang & Liu, 2024) explored non-parametric CVaR estimation using quantile regression forests. In addition to these works, (Gao & Kleywegt, 2022) proposed a distributionally robust CVaR optimization model to address ambiguity in probability distributions. (Takeda & Kanamori, 2021) introduced a mixture CVaR model for portfolio optimization, combining multiple risk measures. (Singh & Maddison, 2023) developed a gradient-based CVaR optimization method for large-scale machine learning problems. (Cardoso & Palomar, 2022) presented an online CVaR optimization algorithm for streaming data applications. Lastly, (Nemirovski & Shapiro, 2022) proposed an efficient simulation-based approach for CVaR estimation in complex systems.

In our approach, we aim to preserve the incremental, online, and adaptive characteristics of our algorithm. This means that the algorithm should continuously update its estimates as new data arrives, adapt to changes in the data or environment, and operate efficiently with minimal computational overhead. Hence we optimize the superquantile by calibrating the measure parameter $\omega$ in the direction of the gradient of

the $\texttt{CVaR}_\rho$ which is estimated using the randomly perturbed finite difference method, as follows:

$$\widehat{\nabla C}_\rho(\omega_t) = \frac{C_\rho(\omega_t + c_t\Delta_t) - C_\rho(\omega_t - c_t\Delta_t)}{2c_t\Delta_t}, \tag{45}$$

where $C_\rho(\cdot) = \texttt{CVaR}_\rho(\cdot)$, $c_t > 0$ with $\lim_{t\to\infty} c_t \downarrow 0$ and $\Delta_t \in \mathbb{R}^p$ with each of the components are $\Delta_{t_i} \overset{\text{iid}}{\sim} \texttt{Bernoulli}(\{-1, 1\}, 0.5)$. Also, $\Delta_t^{-1} = [\Delta_{t_1}^{-1}, \Delta_{t_2}^{-1} \ldots \Delta_{t_p}^{-1}]^\top$. The random perturbations (with zero mean) can introduce variance in gradient estimates which gets asymptotically averaged over multiple random perturbations during the stochastic gradient recursion. As the number of perturbations increases, the average gradient estimate approaches the true gradient asymptotically, meaning that with enough samples, the variance in the gradient estimate becomes negligible and the estimate becomes increasingly accurate.

---

**Algorithm 2** [Extreme Superquantile Optimization Algorithm]

---

1: **Require:** Learning rates $a_t, c_t, \alpha_t, \beta_t > 0$

2: **Input parameters:** $\lambda, \rho \in [0, 1]$

3: **Initialize** $t = 0$, $\gamma_{0,0}^+ = \gamma_{0,0}^- - \infty$, $\eta_{0,0}^+ = \eta_{0,0}^- = 0$

4: **while** Stopping criteria is not satisfied **do**

5: $\quad \Delta_{t_i} \overset{\text{iid}}{\sim} \texttt{Bernoulli}(\{-1, 1\}, 0.5), \quad \forall i = 1 \ldots p$

6: $\quad$ Let $\omega_t^+ = \omega_t + c_t\Delta_t$ and $\omega_t^- = \omega_t - c_t\Delta_t$, where $\Delta_t = [\Delta_{t_1} \ldots \Delta_{t_p}]^\top$

7: $\quad$ **for** $k = 0$ to $N_t$ **do**

8: $\quad\quad \widehat{Q}_{t,k}^+ \sim \texttt{Bernoulli}(\{\mathbb{P}_{\omega_t^+}, Q_{\theta_{t,k}^+}\}, \lambda)$, where $\theta_{t,k}^+ = m^{-1}(\eta_{t,k}^+)$;

9: $\quad\quad$ Update $\texttt{VaR}$ estimate: For $\widehat{\mathbf{X}}_{t,k+1}^+ \sim \widehat{Q}_{t,k}^+$,

$$\gamma_{t,k+1}^+ = \gamma_{t,k}^+ + \alpha_k \zeta_{t,k}^+(\widehat{\mathbf{X}}_{t,k+1}^+)((1 - \rho)\mathbb{I}(\phi(\widehat{\mathbf{X}}_{t,k+1}^+), \gamma_{t,k}^+)^+ - \rho\mathbb{I}(\phi(\widehat{\mathbf{X}}_{t,k+1}^+), \gamma_{t,k}^+)^-);$$

10: $\quad\quad$ Update surrogate parameters: For $\mathbf{Y}_{t,k+1}^+ \sim Q_{\theta_{t,k}^+}$,

$$\eta_{t,k+1}^+ = \eta_{t,k}^+ + \beta_k \zeta_{t,k}^+(\mathbf{Y}_{t,k+1}^+)(\mathbb{I}(\phi(\mathbf{Y}_{t,k+1}^+), \gamma_t^+)^+ \Gamma(\mathbf{Y}_{t,k+1}^+) - \eta_{t,k}^+ \mathbb{I}(\phi(\mathbf{Y}_{t,k+1}^+), \gamma_{t,k}^+)^-);$$

11: $\quad\quad$ Update $\texttt{CVaR}$ gradient estimate: $\bar{\eta}_{t,k+1}^+ = \frac{1}{(k+1)^c}\left(t^c\bar{\eta}_{t,k}^+ + \phi\left(\mathbf{Y}_{t,k+1}^+\right)\right)$;

12: $\quad$ **end for**

13: $\quad$ **Repeat** steps 8 - 11 for $\gamma_{t,k}^-, \eta_{t,k}^-$ and $\bar{\eta}_{t,k}^-$;

14: $\quad$ Update distribution parameters

$$\omega_{t+1} = \omega_t + \frac{a_t}{2c_t\Delta_t}\left\{\bar{\eta}_{t,N_t}^+ - \bar{\eta}_{t,N_t}^-\right\}$$

15: $\quad t = t + 1$

16: **end while**

---

Define the filtration $\{\mathcal{F}_t\}_{t\in\mathbb{N}}$, where the $\sigma$-field $\mathcal{F}_t := \sigma\left(\omega_i, \Delta_{i-1}, \gamma_i, \eta_i, \bar{\eta}_i, \theta_i, 0 \leqslant i \leqslant t, \widehat{\mathbf{X}}_i, \mathbf{Y}_i, 1 \leqslant i \leqslant t\right)$.

**Lemma 7.** *Let $C_\rho^{(3)}(\omega) \equiv \partial^3 C_\rho/\partial\omega^T\partial\omega^T\partial\omega^T$ exists and $\max_{i_1,i_2,i_3} \sup_\omega \|C_{\rho i_1 i_2 i_3}^{(3)}(\omega)\|_\infty \leqslant \alpha$. Then for all $\omega \in \Omega$*

$$b_t = \mathbb{E}\left[\widehat{\nabla C}_\rho(\omega_t) - \nabla C_\rho(\omega_t) \mid \mathcal{F}_t\right] = \mathcal{O}(c_t^2).$$

*Proof.* Please refer to Appendix. $\qquad\qquad\qquad\qquad\qquad\qquad\qquad\qquad\qquad\qquad\qquad\qquad\square$

Now we state our main result.

**Theorem 3.** *Assume that $N_t = O(\lceil (t+1)^{c/2} \rceil)$ and $a_t = o(1/(t+1)^c)$. Then the iterates $\{\omega_t\}$ generated by Algorithm satisfy the following:*

$$\omega_t \to H = \{\omega | \nabla CVaR_\rho(\omega) = 0\} \text{ on the event } \{\sup_t \|\omega_t\| < \infty\} \text{ as } t \to \infty.$$

*Further, if $H$ is a discrete set, then we have the following.*

$$\omega_t \to \{\omega | \nabla CVaR_\rho(\omega) = 0 \text{ and } \nabla^2 CVaR(\omega) \ll 0\} \text{ on the event } \{\sup_t \|\omega_t\| < \infty\} \text{ as } t \to \infty.$$

*Proof.* Please refer to Appendix. □

**Remark.** *The aforementioned result indicates that the distribution parameters $\omega_t$ converge to the local maxima of the objective $CVaR_\rho$, provided that the iterates $\omega_t$ remains bounded which is denoted by the condition $\sup_t \|\omega_t\| < \infty$. This condition is necessary because noise can cause the iterates to gradually drift outward, potentially leading to divergence. This can be achieved by constraining the iterates to remain within a convex compact set, and if they drift beyond its boundary, they can be projected back onto the set. The projected version of the recursion is as follows:*

$$\omega_{t+1} = \Pi^\Omega \left( \omega_t + \frac{a_t}{2c_t \Delta_t} \left\{ \bar{\eta}_t^+ - \bar{\eta}_t^- \right\} \right), \text{ where } \Pi^\Omega(v) = \arg\min_{\omega \in \Omega} \|v - \omega\|_2^2. \tag{46}$$

*Further, one can show that the projected iterates behave asymptotically similar to the long term behaviour of the ODE*

$$\frac{d\omega(t)}{dt} = \bar{\Pi}^\Omega_{\omega(t)}(\nabla C_\rho(\omega_t)), t \geq 0, \tag{47}$$

*where $\bar{\Pi}^\Omega$ is the Frechet derivative which is defined as $\bar{\Pi}^\omega_x(y) = \lim_{\delta \to 0^-} \frac{\Pi^\Omega(x + \delta y) - x}{\delta}$ exists.*

**Remark.** *To improve the quality of the solution, one can inject a decaying Gaussian noise (Maryak & Chin, 2008) to the iterates $\omega_t$ as follows:*

$$\omega_{t+1} = \omega_t + \frac{a_t}{2c_t \Delta_t} \left\{ \bar{\eta}_t^+ - \bar{\eta}_t^- \right\} + q_t \varepsilon_t,$$

*where $q_t > 0$ is the step schedule and $\varepsilon_t \overset{iid}{\sim} \mathcal{N}(0, \mathcal{I})$. The noise term $q_t \epsilon_t$ introduces randomness into the update process, but this randomness is controlled by $q_t$ which typically decreases over time to ensure that the influence of noise diminishes. When the noise is suitably behaved and certain other conditions are satisfied, the iterates $\omega_t$ so generated converge to the global maxima of $C_\rho$ in the following sense:*

$$\lim_{t \to 0} \mathbb{E}[C_\rho(\omega_t)] = \int C_\rho d\pi \tag{48}$$

*where $\pi$ is the Dirac measure that concentrates on the global maxima of $C_\rho$ (assuming that the global maxima are unique), which means that the measure places all its mass on the global maximum. Please refer to (Maryak & Chin, 2008) for the conditions required to ensure convergence.*

## 5 Extreme Risk Measure Estimation with Latent Measure

In reinforcement learning (RL) and many real-world scenarios, one often deals with uncertainty and unknowns. One of the challenges is that the true underlying probability distribution $\mathbb{P}_x$ that governs the rewards is not explicitly known. Instead, one has access to an oracle which can provide samples or realizations from this distribution $\mathbb{P}_x$. These schemes are applicable in cases where the exact dynamics are unknown or complex, but simulation is easy to perform. Here, we propose an extension of our algorithm that aims to perform extreme risk measurement estimation and optimization in those settings as well. The samples we obtain provide empirical data that we use to approximate the behavior of $\mathbb{P}_x$ by performing moment projection into the parameterized probability measure space $\mathcal{U}_\mathcal{K} = \{U_\kappa | \kappa \in \mathcal{K} \subseteq \mathbb{R}^{d_2}\}$. The moment

projection technique maps empirical moments from the sample data to the specific measure space, allowing us to approximate the characteristics of the underlying distribution despite not knowing its explicit form. The moment projection is obtained by the following optimization problem:

$$\kappa_{t+1} := \arg\min_{\kappa \in \mathcal{K}} \mathrm{KL}(\mathbb{P}_x, U_\kappa)$$

Further,

$$
\begin{aligned}
\arg\min_{\kappa \in \mathcal{K}} \mathrm{KL}(\mathbb{P}_x, U_\kappa) &= \arg\min_{\kappa \in \mathcal{K}} \int d\mathbb{P}_x \log \frac{d\mathbb{P}_x}{dU_\kappa} \\
&= \arg\min_{\kappa \in \mathcal{K}} \underbrace{\int (d\mathbb{P}_x/d\nu) \log (d\mathbb{P}_x/d\nu) d\nu}_{\text{Does not contain } \kappa. \text{ So we drop it.}} - \int (d\mathbb{P}_x/d\nu) \log (dU_\kappa/d\nu) d\nu \\
&= \arg\min_{\kappa \in \mathcal{K}} - \int (d\mathbb{P}_x/d\nu) \log (dU_\kappa/d\nu) d\nu \\
&= \arg\max_{\kappa \in \mathcal{K}} \int (d\mathbb{P}_x/d\nu) \log (dU_\kappa/d\nu) d\nu \\
&= \arg\max_{\kappa \in \mathcal{K}} \int \log (dU_\kappa/d\nu) d\mathbb{P}_x
\end{aligned}
$$

The above optimization problem can be solved using a stochastic recursion as follows

$$\kappa_{t+1} = \kappa_t + \alpha_t \nabla \log \frac{dU_\kappa}{d\nu}(\mathbf{Z}_{t+1}) \quad \text{where } \mathbf{Z}_{t+1} \sim \mathbb{P}_x \tag{49}$$

Note that we follow the same timescale as that of $\gamma_t$ since there exists a unilateral coupling between $\kappa_t$ and other iterates. The existence of a unilateral coupling means that while $\kappa_t$ might affect other variables or iterates, the reverse is not necessarily true: These other variables do not directly influence $\kappa_t$. Also note that $\zeta_t(x) = (dU_{\kappa_t}/dQ_{\theta_t})(x)$.

The algorithm for the latent case is provided in Algorithm 3.

---

**Algorithm 3** [Extreme quantile and superquantile estimation algorithm with latent $\mathbb{P}_x$]

---

1: **Require:** $\alpha_t, \beta_t > 0$ and $T \in \mathbb{N}$

2: **Input parameters:** $\lambda, \rho \in [0, 1]$

3: **Initialize** $\gamma_0 = -\infty$, $t = 0$, $\eta_0$

4: **while** $t < T$ **do**

5: $\quad \kappa_{t+1} = \kappa_t + \alpha_t \nabla \log \left( (dU_\kappa/d\nu)(\mathbf{Z}_{t+1}) \right)$, where $\mathbf{Z}_{t+1} \sim \mathbb{P}_x$;

6: $\quad \widehat{Q}_t \sim \mathtt{Bernoulli}(\{dU_{\kappa_t}/d\nu, Q_{\theta_t}\}, \lambda)$, where $\theta_t = m^{-1}(\eta_t)$;

7: $\quad \gamma_{t+1} = \gamma_t + \alpha_t \zeta_t(\widehat{\mathbf{X}}_{t+1})((1-\rho)\mathbb{I}^+(\phi(\widehat{\mathbf{X}}_{t+1}), \gamma_t) - \rho\mathbb{I}^-(\phi(\widehat{\mathbf{X}}_{t+1}), \gamma_t))$, where $\widehat{\mathbf{X}}_{t+1} \sim \widehat{Q}_t$;

8: $\quad \eta_{t+1} = \eta_t + \beta_t \zeta_t(\mathbf{Y}_{t+1})(\mathbb{I}^+(\phi(\mathbf{Y}_{t+1}), \gamma_t)\Gamma(\mathbf{Y}_{t+1}) - \eta_t\mathbb{I}^-(\phi(\mathbf{Y}_{t+1}), \gamma_t))$, where $\mathbf{Y}_{t+1} \sim Q_{\theta_t}$;

9: $\quad \bar{\eta}_{t+1} = \frac{1}{(t+1)^c} \left( t^c \bar{\eta}_t + \phi(\mathbf{Y}_{t+1}) \right)$;

10: $\quad t = t + 1$;

11: **end while**

12: **Return** $\widehat{\mathrm{VaR}}_\rho(\phi) = \gamma_T$, $\widehat{\mathrm{CVaR}}_\rho(\phi) = \bar{\eta}_T$;

---

**Theorem 4.** *Let the learning rates $\{\alpha_t\}_{t \in \mathbb{N}}$ and $\{\beta_t\}_{t \in \mathbb{N}}$ satisfy Assumption (A1). Also, let Assumption (A2) hold. Also let $\alpha_t \in \Omega(1/(t+1)^c)$. Then the iterates $\{\kappa_t\}$ generated by Algorithm satisfy the following:*

$$\kappa_t \to G_\kappa = \left\{ \kappa \,\Big|\, \nabla \mathbb{E}\left[ \log \frac{dU_\kappa}{d\nu}(\mathbf{Z}) \right] = 0 \right\} \quad \textbf{a.s.} \text{ on the set } \{\sup_t \|\kappa_t\| < \infty\} \text{ as } t \to \infty.$$

*Further, if $G$ is a discrete set, then we have the following:*

$$\kappa_t \rightarrow \left\{ \kappa \left| \nabla \mathbb{E} \left[ \log \frac{dU_\kappa}{d\nu}(\mathbf{Z}) \right] = 0 \; and \; \nabla^2 \mathbb{E} \left[ \log \frac{dU_\kappa}{d\nu}(\mathbf{Z}) \right] \preccurlyeq 0 \right\} \quad \textbf{\textit{a.s.}} \; on \; the \; set \; \{ \sup_t \| \kappa_t \| < \infty \} \; as \; t \rightarrow \infty.$$

$$\lim_{t\rightarrow\infty} \gamma_t = \textit{VaR}_\rho(U_{\kappa*}) \;\; \textbf{\textit{a.s.}}, \qquad \lim_{t\rightarrow\infty} \eta_t = \frac{\int \mathbb{I}(\phi(\mathbf{Y}), \textit{VaR}_\rho(U_{\kappa*}))^+ \Gamma(\mathbf{Y}) d\mathbb{P}_x}{\int \mathbb{I}(\phi(\mathbf{Y}), \textit{VaR}_\rho(U_{\kappa*}))^+ d\mathbb{P}_x} \;\; \textbf{\textit{a.s.}}, \; where \; \kappa^* \in G_\kappa, and$$

$$\lim_{t\rightarrow\infty} \bar{\eta}_t = \int \Gamma(\mathbf{Y}) dQ_{\theta*} \;\; \textbf{\textit{a.s.}}, \; where \; \theta^* = m^{-1}(\lim_{t\rightarrow\infty} \eta_t).$$

*Proof.* Please refer to Appendix. $\qquad\qquad\qquad\qquad\qquad\qquad\qquad\qquad\qquad\qquad\qquad\qquad\qquad\qquad\qquad$ $\square$

## 6 Experimental Results

### 6.1 Risk adjusted Portfolio Optimization

The importance of VaR and CVaR is monumental in the domain of finance, in various problems that consider the inherent risk in executing trades or optimal portfolio allocation using the risk-adjusted returns which is a direct consequence of Modern Portfolio Theory (Markowitz, 1991). In our effort to prove the resilience of our procedure in real world scenarios we conduct experiments in two parts, primarily we see how our algorithm fares with actual returns from a single stock then we extend this to compute the optimal risk adjusted portfolio.

### 6.1.1 Single Stock

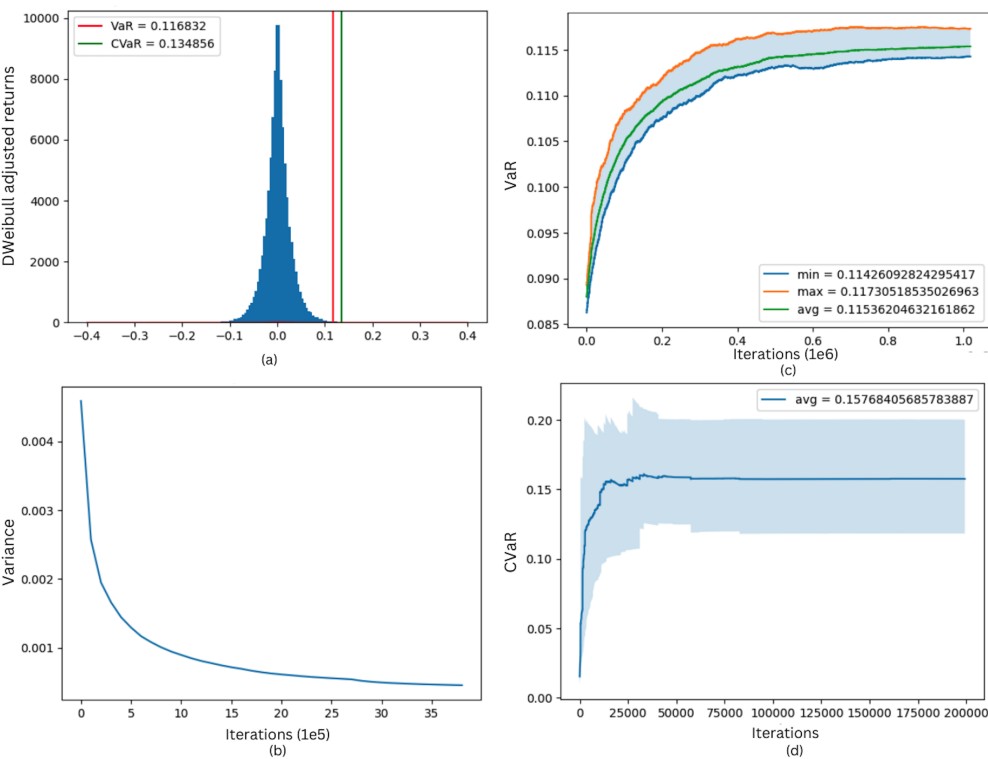

Figure 5: Estimation of `VaR` and `CVaR` at $\rho = 0.001$ of `$AAPL` returns data from 1980-2023 using the per day granularity on closing price shown in (c) and (d). The convergence to the true approximate values stated in (a) is shown by the decrease in the adaptive variance of the surrogate PDF in (b).

We collect the split adjusted per day closing price for the stock $AAPL over four decades precisely, 1980 to 2023, from which we generate the empirical CDF for the sorted returns, given by

$$\widehat{F}_n(t) = \frac{1}{n} \sum_{i=1}^{n} \mathbb{I}_{X_i \leqslant t}$$

where $n$ is the number of samples and $X_i$ is the sample from the sorted returns. The VaR and CVaR is then computed using Monte-Carlo and verified by fitting a double Weibull distribution to the returns using MLE shown in Figure 5(a). We estimate the quantities precisely at $\rho = 10^{-3}$ using Algorithm 1 with the consistently depleting error bound using our method illustrated in 5(c) and 5(d) - which is also verified by the decay of the variance to 0 of the surrogate measure obtained from the Gaussian surrogate. Along with the hyperparameters given Table 1 we additionally use a threshold of value $10^{-4}$ as the stopping criterion.

| Hyperparameter | Value |
|----------------|-------|
| `num_samples` | 10000 |
| `bootstrap_samples` | 100 |
| $\lambda$ | 0.4 |
| $\alpha_0$ | 0.015 |
| $\alpha_{decay}$ | 0.9 |
| $\beta_0$ | 0.1 |
| $\beta_{decay}$ | 0.55 |
| $\mu_0$ | 35 |
| $\Sigma_0$ | 1 |
| $\eta_0$ | 10 |

Table 1: Hyperparameters for $AAPL stock quantile estimation.

### 6.1.2 Portfolio Allocation

For this paper, we try to find the optimal CVaR portfolio, which formulated as a constrained optimization problem. Let weight vector, $\mathbf{w} = (w_1, \ldots, w_n)^\top$ denote the weights of each asset in the portfolio for $n$ assets, and $\mathbf{r} = (r_1, \ldots, r_n)^\top$ represent the random vector of asset returns. For a portfolio, $P$ the total return is given by $R_P = \mathbf{w}^\top \mathbf{r}$ and the CVaR portfolio return for a specified confidence level $\rho \in (0,1)$, typically 0.05 or 0.001, is defined as

$$\begin{aligned} \min_{\mathbf{w}} \quad & \text{CVaR}_\rho(-\mathbf{w}^\top \mathbf{r}) \\ \text{subject to} \quad & \mathbf{w}^\top \mathbf{1} = 1 \\ \text{s.t.} \quad & w_i \in [0,1) \quad \forall i \in [1,n] \end{aligned} \tag{50}$$

this optimization problem aims to minimize the CVaR of the negative portfolio return subject to constraints that the sum of the weigthts equal to 1 and the weights are normalized. This formulation provides a robust framework for portfolio optimization that accounts for tail risk, aligning with contemporary risk management practices in quantitative finance and addressing the limitations of traditional mean-variance optimization approaches.

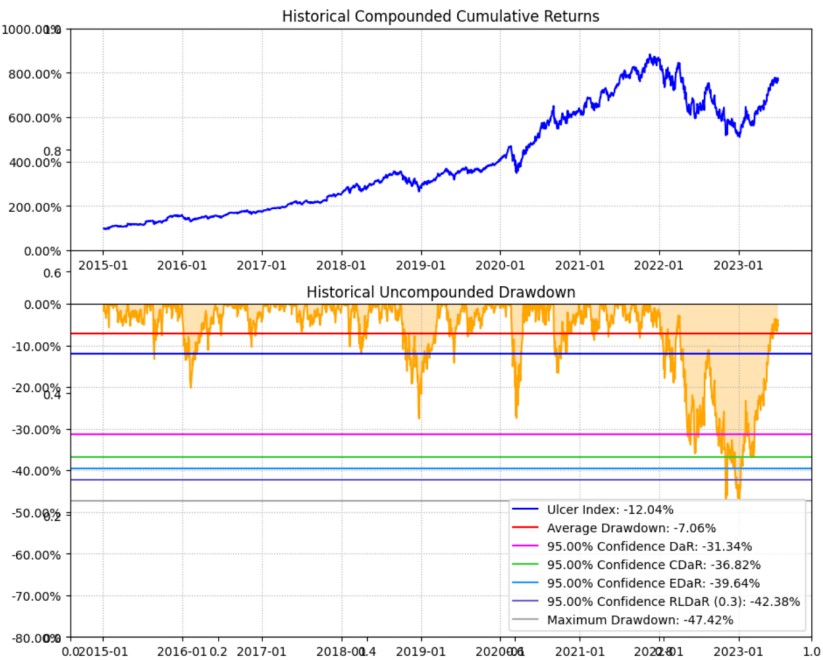

Figure 6: Cumulative risk-adjusted returns (top) and the draw-down risk (bottom) represented in the depth graph for the portfolio under consideration.

The negative returns $(-\mathbf{w}^\top \mathbf{r})$, represents the loss associated with a given portfolio allocation which in the `CVaR` portfolio stems from the focus on downside risk (Klebaner et al., 2017), as we are primarily concerned with potential losses rather than gains. By using negative returns, we align our optimization problem with the conventional definition of `VaR` and `CVaR` in the spectrum of finance, which are typically expressed in terms of losses. This approach aligns with the primary concern of many investors and regulators - minimizing the potential for significant losses in adverse market conditions.

The matrix $S_t \in \mathbb{R}^{h \times n}$, represents the day close values over the selected period, where $h$ is a predefined window size and $n$ is the number of assets. Each element $s_{i,j}$ of the matrix represents the return of the asset $j$ at the time step (per day) $t - h + i$, we refer to this as the state matrix. Formally:

$$S_t = \{r_{i,j} | i \in [t - h + 1, t], j \in [1, n]\} \tag{51}$$

where $r_{i,j}$ is the return of asset $j$ at time $i$. The observation space is bounded, with $S_t \in [0, 1]^{h \times n}$. At each time step $t$, the state $S_t$ is represented by a matrix $S_t \in \mathbb{R}^{h \times n}$, where, $h$ is the window size (number of historical time steps and $n$ is the number of assets in the portfolio. We define the state matrix as follows:

$$S_t = \begin{bmatrix} r_{t-h+1,1} & r_{t-h+1,2} & \cdots & r_{t-h+1,n} \\ r_{t-h+2,1} & r_{t-h+2,2} & \cdots & r_{t-h+2,n} \\ \vdots & \vdots & \ddots & \vdots \\ r_{t,1} & r_{t,2} & \cdots & r_{t,n} \end{bmatrix} \tag{52}$$

where $r_{i,j}$ represents the return of asset $j$ at time step $i$. Each element $s_{i,j}$ of the matrix $S_t$ corresponds to the return of a specific asset at a specific time:

$$s_{i,j} = r_{t-h+i,j} = \frac{P_{t-h+i,j} - P_{t-h+i-1,j}}{P_{t-h+i-1,j}} \tag{53}$$

where $P_{t,j}$ is the price of asset $j$ at time $t$. To ensure numerical stability and consistent scale across different assets, we bound the elements of the state matrix $S_t \in [0, 1]^{h \times n}$. The rows of the matrix $S_t$ represent

different time steps, with the most recent returns in the bottom row and the oldest returns in the top row. The rows of the matrix $S_t$ represent different time steps, with the most recent returns in the bottom row and the oldest returns in the top row. This structure allows the agent to identify temporal patterns or trends in asset returns potentially.

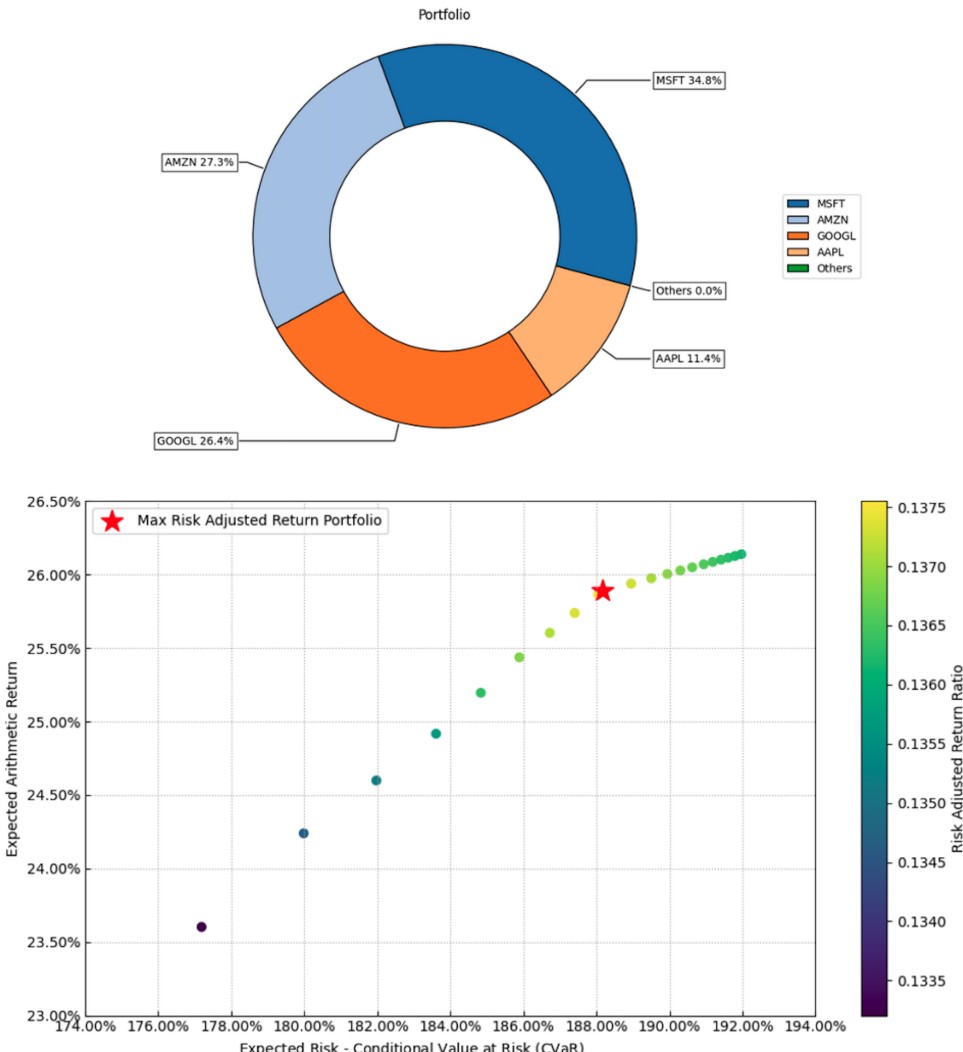

Figure 7: Final optimal allocation (top) and the expected efficient frontier analysis (bottom).

In our experiments, we construct our portfolio consisting of the assets [ $MSFT, $AAPL, $META, $GOOGL ] with $\rho = 0.001$ and starting with equal proportions of each of them. We choose a period of about a decade from 2015 - 2023 and solve the optimization problem in Eq. 50 using *CVXPY* (Diamond & Boyd, 2016) and *MOSEK* (ApS, 2024) solver. The drawdown risk along with the cumulative returns is visualized in Figure6, where it is evident that in periods of volatility, namely, during COVID (2020-2021) where see rise in the tech stocks while there persists quantifiable amount of risk and also during the tech recession (2022-2023) the returns are sustained even though there exists a higher draw-down risk.

The final portfolio allocation is given by the pie-chart in Figure 7(top) which is obtained under the strategy followed by the CVAR adjusted portfolio allocation and we also verify by the efficient frontier analysis (Maiti, 2021; Banihashemi & Navidi, 2017) which suggest that to an investor no other portfolio exists that maximizes their returns while minimizing the associated risk. For this we use the *CLARABEL* (Goulart & Chen, 2024) solver with *RiskFolio* (Cajas, 2024), given in Figure 7(bottom).

**Drawdown risk:** Drawdowns measure the decline from a historical peak in an investment's value, capturing both the depth (percentage drop from peak to trough) and duration (time to recover). The top graph in Figure 6 displays historical compounded cumulative returns, showing the portfolio's overall growth while bottom graph highlights drawdowns, illustrating periods of decline, such as during 2019 and 2022-2023, with a maximum drawdown of around -47%. While the portfolio shows long-term positive returns, significant drawdowns underline the importance of balancing upside potential with downside risk.

**Efficient frontier analysis:** The efficient frontier analysis evaluates the trade-off between risk and return for various portfolios. Portfolios on the frontier are optimal, offering the highest return for a given risk or the lowest risk for a given return. The curvature reflects diversification benefits, with steeper curves indicating greater risk reduction through diversification. We identify portfolios with superior risk-adjusted returns (e.g., maximizing return per unit of risk) using a color gradient, which highlights areas with optimal balance. This analysis confirms the stability and efficiency of portfolios generated by the `CVaR` optimization method. In essence, the efficient frontier provides a visual representation of the best possible risk-return combinations an investor can achieve with a given set of assets. The specific portfolio chosen on the frontier depends on the investor's risk appetite.

## 6.2 Risk Sensitive Reinforcement Learning

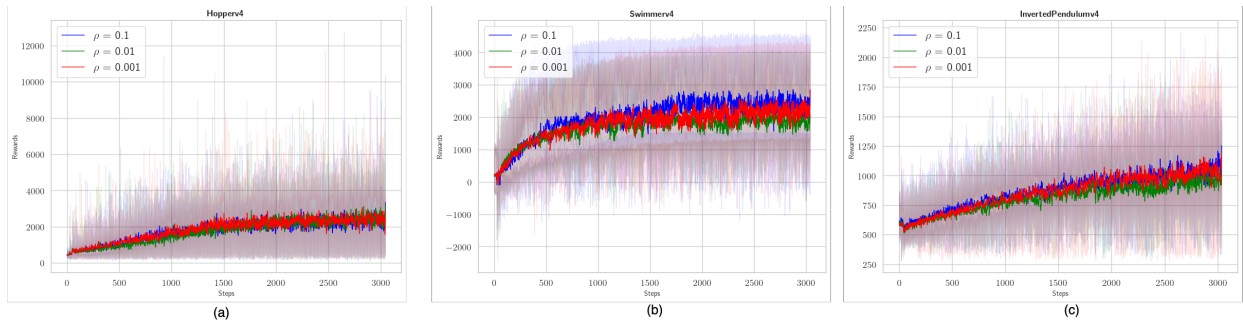

Figure 8: Rewards obtained under CVaR as risk measure at varied levels of confidence for the locomotion tasks *Hopper-v4, Swimmer-v4* and *InvertedPendulum-v4* (left to right).

We assume that in the context of this paper our problem is modeled as a MDP (Puterman, 2014), defined by the tuple $(\mathcal{S}, \mathcal{A}, R, P, S_0, \gamma)$, where $\mathcal{S} = 1 : S$ and $\mathcal{A} = 1 : A$ are the state space and action space which is a set of states and actions, respectively. The given expression $a : b$ denotes a sequence or a set $a, a+1, \ldots, b$. The reward function $R : \mathcal{S} \times \mathcal{A} \times \mathcal{S} \to \mathbb{R}$ represents the reward received for each state after taking an action. Here we consider our rewards to be bounded such that $r(s, a, s') \in [0, r_{max}]$ and $s \in \mathcal{S}$ and $a \in \mathcal{A}$. The transition probabilities are $\mathbb{P} : \mathcal{S} \times \mathcal{A} \to \Delta^S$, where $\Delta^S$ is the probability simplex in $\mathbb{R}^S$ and for a particular state-action pair, $P(\cdot|s, a)$ is the transition probability, and $P_0(\cdot)$ is the initial state distribution which is defined as $P_0 = \mathbb{I}_{\{s=s_0\}}$ for some given initial state. Finally, $S_0 \in \mathcal{S}$ is the initial state and $\delta \in (0, 1]$ is the discount factor. For each state $x$, the set $\mathcal{A}(s)$ gives all available actions.

A stationary policy $\pi(\cdot|s)$ is a probability distribution over actions that depends on current state $s$. Here we consider it parameterized by a $p$-dimensional vector $\omega$, which means the policy space can be written as $\Pi_\omega = \{\pi(\cdot|s, \omega), s \in \mathcal{S}, \omega \in \Omega \subseteq \mathbb{R}^p\}$ and the parameter space $\Omega$ is assumed to be a convex compact set. In contrast to the risk neutral case, the objective function is augmented with the risk measure, $\phi[.]$ to obtain the risk sensitive objective as $\max_{\pi_\omega \in \Pi} \phi[J_\omega^\pi]$. The $\pi_\omega$ we receive is the risk-sensitive policy induced by the risk measure $\phi[\cdot]$. Here, we consider CVaR as the risk measure and the primary objective is to maximize the CVaR of the discounted cumulative rewards received by following policy given as

$$\max_{\pi_\omega \in \Pi} \mathtt{CVaR}_\rho[J_\omega^\pi] \text{ where } J_\omega^\pi := \sum_{k=0}^{\tau-1} \delta^k R(s_k, a_k, s_{k+1}) \text{ with } s_0 \sim \mathbb{P}_0, \tag{54}$$

$a_k \sim \pi_\omega(\cdot|s_k), S' \sim P(\cdot|s_k, a_k)$ , $\forall k \in [0 : \tau]$ and $\tau \in \mathbb{N}^+ \cup \{\infty\}$ which is the stopping time.

---

**Algorithm 4** Superquantile Policy Optimization

---

1: **Require:** Learning rates $a_t, c_t, \alpha_t, \beta_t > 0$

2: **Input parameters:** $\rho, \delta, \lambda \in [0, 1)$, batch size $M$, sample size $N$ and discount factor $\delta$

3: **Initialize** policy network $(\pi_\omega)$ parameters, $\omega \in \Omega$, $\gamma_{0,0}^+ = \gamma_{0,0}^- = -\infty$, $t = 0$, $\eta_{0,0}^+ = \eta_{0,0}^- = 0$

4: **while** Stopping criteria is not satisfied **do**

5:     $\Delta_{t_i} \overset{\text{iid}}{\sim} \texttt{Bernoulli}(\{-1, 1\}, 0.5), \quad \forall i = 1 \ldots p$

6:     Let $\omega_t^+ = \omega_t + c_t \Delta_t$ and $\omega_t^- = \omega_t - c_t \Delta_t$, where $\Delta_t = [\Delta_{t_1} \ldots \Delta_{t_p}]^\top$

7:     **for** $k = 0$ to $N_t$ **do**

8:         $\hat{Q}_{t,k}^+ \sim \texttt{Bernoulli}(\{\mathbb{P}_{\pi_{\omega_t^+}}, Q_{\theta_{t,k}^+}\}, \lambda)$, where $\theta_{t,k}^+ = m^{-1}(\eta_{t,k}^+)$;

9:         Compute $\bar{J}_{t,k+1}^+ = \sum_{j=0}^{M-1} \delta^j R\left(s_{t,j}^+, a_{t,j}^+, s_{t,j+1}^+\right)$ from $\left\{s_{t,0}^+, a_{t,0}^+, s_{t,1}^+, \ldots s_{t,M-1}^+, a_{t,M-1}^+, s_{t,M}^+\right\} \sim \pi_{\omega_t^+}$

10:         Update $\texttt{VaR}$ estimate: For $\hat{\mathbf{X}}_{t,k+1}^+ \sim \hat{Q}_{t,k}^+$,

$$\gamma_{t,k+1}^+ = \gamma_{t,k}^+ + \alpha_k \zeta_{t,k}^+(\hat{\mathbf{X}}_{t,k+1}^+)((1-\rho)\mathbb{I}(\phi(\hat{\mathbf{X}}_{t,k+1}^+), \gamma_{t,k}^+)^+ - \rho\mathbb{I}(\phi(\hat{\mathbf{X}}_{t,k+1}^+), \gamma_{t,k}^+)^-);$$

11:         Update surrogate parameters: For $\mathbf{Y}_{t,k+1}^+ \sim Q_{\theta_{t,k}^+}$,

$$\eta_{t,k+1}^+ = \eta_{t,k}^+ + \beta_k \zeta_{t,k}^+(\mathbf{Y}_{t,k+1}^+)(\mathbb{I}(\phi(\mathbf{Y}_{t+1}^+), \gamma_t^+)^+ \Gamma(\mathbf{Y}_{t,k+1}^+) - \eta_{t,k}^+ \mathbb{I}(\phi(\mathbf{Y}_{t,k+1}^+), \gamma_{t,k}^+)^-);$$

12:         Update $\texttt{CVaR}$ gradient estimate: $\bar{\eta}_{t+1,k+1}^+ = \frac{1}{(k+1)^c}\left(t^c \bar{\eta}_{t,k}^+ + \phi\left(\mathbf{Y}_{t,k+1}^+\right)\right)$;

13:     **end for**

14:     **Repeat** steps 8 - 12 for $\gamma_{t,k}^-, \eta_{t,k}^-$ and $\bar{\eta}_{t,k}^-$;

15:     Update distribution parameters

$$\omega_{t+1} = \omega_t + \frac{a_t}{2c_t\Delta_t}\left\{\bar{\eta}_{t,N_t}^+ - \bar{\eta}_{t,N_t}^-\right\}$$

16:     $t = t + 1$

17: **end while**

---

When the reward is of the form $R = f_\omega(X)$, and $X$ is a random vector, CVaR optimization can be formulated as a stochastic program and solved using classical approaches. This structure is often found in portfolio optimization, where the investment strategy generally does not influence asset prices. However, in the RL context, the policy parameters affect the probability distribution of the trajectory space. Consequently, the CVaR values are sensitive to the policy parameters. In optimizing CVaR policies, the objective is to find the policy that achieves the optimal CVaR (Tamar et al., 2015; Chow et al., 2015).

In our experimental setting, we consider parameterization of our policy using the Gaussian distribution which is given as, $\pi_\omega(a \mid s) = \mathcal{N}\left(a \mid \Phi(s)^\top x, e^y\right)$, with parameters $\omega = [x, y]^\top$. Here the mean is given by $\Phi(s)^\top \mathbf{x}$ and variance as $e^y$, with $\inf_y e^y > 0$. Here, $\Phi(s)$ represents a normalized feature map with which maps a state $s$ to a finite dimensional Euclidean space, i.e., $\Phi : \mathcal{S} \to \mathbb{R}^d$. We carry out locomotion tasks on the various control environments *InvertedDoublePendulum-v4, Swimmer-v4* and *Hopper-v4* which are part of the MuJoCo framework (Tassa et al., 2018) to test our algorithm.

To further amplify the uncertainty in the decision-making scenario, we introduce a zero-mean white noise component to both the observation space and the action space. This augmentation reinforces the agent's exposure to uncertainty. The selected control environments are specifically designed with penalty terms embedded within their reward functions, which penalize the agent for unsuccessful task completion. As a

result, these environments generate reward distributions characterized by higher variance, thereby increasing the probability of encountering worst-case scenarios, independent of the expected rewards.

| Hyperparameter | Value |
|---|---|
| `seeding schedule` | [42, 30, 8, 67, 52] |
| $\lambda$ | 0.4 |
| $\alpha_0$ | 0.015 |
| $\alpha_{decay}$ | 0.9 |
| $\beta_0$ | 0.1 |
| $\beta_{decay}$ | 0.55 |
| `learning_rate`($a_t$) | 0.01 |
| `discount_factor` | 0.9 |
| `num_episodes` | 200 |
| `lr_decay` | $10^{-3}$ |
| `lr_power` | 0.5 |
| `px` | 2.0 |
| `px_decay` | $10^{-2}$ |
| `px_power` | 0.161 |

Table 2: Hyperparameter setup for `CVaR` policy.

In our experiments, we evaluated the performance of a `CVaR`-based policy under varying levels of confidence, specifically $\rho = [0.1, 0.01, 0.001]$. The results demonstrate that the agent maintains stability despite increasing levels of uncertainty associated with decreasing $\rho$, which is illustrated in Figure 8. From Table 2, the seeding schedule refers to the random seed used for ensuring reproducibility of experiments. The `lr_decay` parameter specifies the rate at which the learning rate decreases over time, with `lr_power` defining the exponent governing this decay. Similarly, `px` represents the base perturbation size used for gradient approximation, while `px_decay` controls its decay rate, and `px_power` sets the exponent for its decay schedule.

Notably, the agent continues to accumulate rewards effectively, and the variance in its performance remains well-controlled, avoiding any significant escalation. These findings suggest that the `CVaR` policy exhibits robust risk management capabilities, ensuring consistent performance even in highly uncertain environments. The ability to prevent variance from exploding, even under heightened uncertainty, highlights the policy's efficacy in mitigating risk and maintaining stability.

### 6.3 Sensitivity Analysis

We study the sensitivity of varying learning rates for the gradient estimate step in our `CVaR` optimization problem. Our sensitivity analysis encompasses a range of learning rates $a_t = [0.01, 0.1, 0.2, 0.5]$, systematically evaluated across multiple random seeds to account for stochastic variations inherent in the environment and the algorithm. To enhance the algorithm's adaptability to the optimization landscape, we incorporate the Adaptive Moment Estimation (ADAM) (Kingma, 2014) within our optimization algorithm. This combination allows for parameter-specific adaptive learning rates, potentially mitigating the sensitivity to initial learning rate selection. We maintain separate moment estimates for each parameter, updating them based on the gradient approximations. In this context of our gradient estimator with ADAM, our learning rate parameter update rule is:

$$a_{t+1} = a_t - \psi \cdot \frac{\hat{m}_t}{\sqrt{\hat{v}_t} + \epsilon}$$

where: $\hat{m}_t$ is the bias-corrected first moment estimate and $\hat{v}_t$ is the bias-corrected second moment estimate. To show that under the ADAM condition the convergence is guaranteed, we need to show that $\mathbb{E}[\|a_t - a^*\|^2] \to 0$ as $t \to \infty$, where $a^*$ is the optimal learning rate parameter. We give a proof sketch as follows

*Proof.* We define

$$\mathcal{L}_t = \mathbb{E}[\|a_t - a^*\|^2]$$

Now, the expected change $\mathbb{E}[\mathcal{L}_{t+1} - \mathcal{L}_t] = \mathbb{E}[\|a_{t+1} - a^*\|^2 - \|a_t - a^*\|^2]$, which is upper bounded as

$$\mathbb{E}[L_{t+1} - L_t] = \mathbb{E}[\|a_{t+1} - a^*\|^2 - \|a_t - a^*\|^2]$$

$$= \mathbb{E}[\|a_t - \psi\frac{\hat{m}_t}{\sqrt{\hat{v}_t} + \epsilon} - a^*\|^2 - \|a_t - a^*\|^2]$$

$$\leqslant -2\psi\mathbb{E}[(a_t - a^*)\frac{\hat{m}_t}{\sqrt{\hat{v}_t} + \epsilon}] + \psi^2\mathbb{E}[\|\frac{\hat{m}_t}{\sqrt{\hat{v}_t} + \epsilon}\|^2]$$

Choosing a decreasing learning rate schedule $\psi_t = \frac{\psi_0}{\sqrt{t}}$ and under conditions $\sum_{t=1}^{\infty} \psi_t = \infty, \sum_{t=1}^{\infty} \psi_t^2 < \infty$, we can conclude that under appropriate choice of constants and applying the above conditions, we can show that $\mathbb{E}[\mathcal{L}_{t+1} - \mathcal{L}_t] \leqslant 0$ for sufficiently large t, which implies convergence. $\qquad\square$

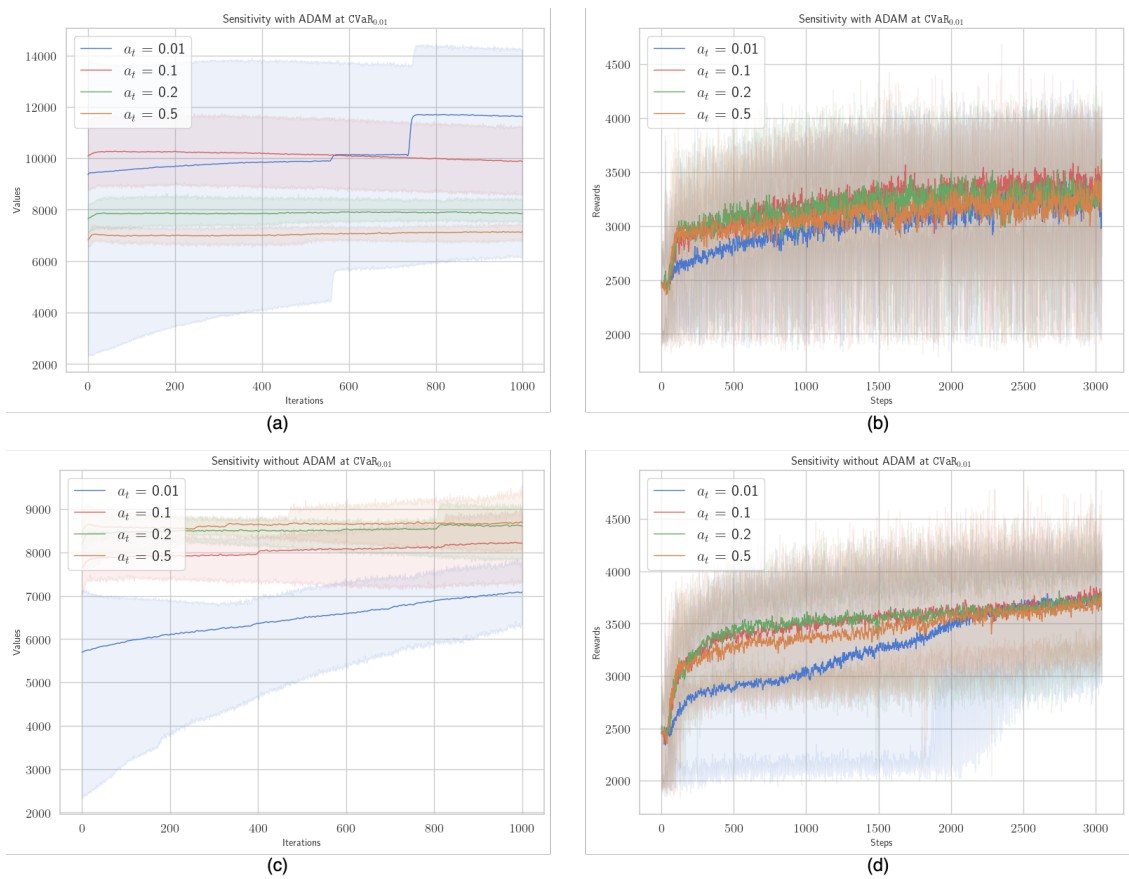

Figure 9: Sensitivity analysis of the learning rates.

For bounding the variance $\mathbb{V}[a_t]$, we proceed as follows

$$\mathbb{V}[a_{t+1}] = \mathbb{V}[a_t - \psi_t\frac{\hat{m}_t}{\sqrt{\hat{v}_t} + \epsilon}]$$

$$\leqslant (1 - \psi_t\lambda)^2\mathbb{V}[a_t] + \psi_t^2\sigma^2$$

where $\lambda$ is related to the smallest eigenvalue of the Hessian of the objective function, and $\sigma^2$ bounds the variance of the gradient estimate. Now unrolling the recursion, we get

$$\mathbb{V}[a_t] \leqslant \prod_{i=1}^{t}(1 - \psi_i\lambda)^2\mathbb{V}[a_0] + \sum_{i=1}^{t}\psi_i^2\sigma^2\prod_{j=i+1}^{t}(1 - \psi_j\lambda)^2$$

Using the decreasing learning rate schedule with $\psi_t = \frac{\psi_0}{\sqrt{t}}$, we show that $\mathbb{V}[a_t] = O\left(\frac{1}{\sqrt{t}}\right)$.

As learning rate sensitivity affects both convergence and variance a higher sensitivity can lead to faster initial convergence but may impact long-term stability. Very high sensitivity can increase the upper bound on variance, potentially leading to less stable convergence. The adaptive nature of ADAM helps mitigate these effects by adjusting the effective learning rate based on the moments of the gradients.

For this experiment, the environment *InvertedDoublePendulum-v4* is considered with added noise and using Algorithm 4, we generate 5 batches for each learning rate, with each batch running for 1000 iterations and the risk sensitivity level is kept fixed at $\rho = 0.001$. Here, the objective function is further augmented with a weighted average return component, balancing risk-sensitivity with overall performance optimization.

| Hyperparameter | Value |
|---|---|
| $\texttt{momentum}(\hat{m}_t)$ | 0.9 |
| $\psi$ | 0.999 |
| $\epsilon$ | $10^{-7}$ |

Table 3: Additional hyperparameters for ADAM

From Figure 9(a) which depicts the movement of the iterates and Figure9(b) the total reward obtained, it is evident that the introduction of an adaptive learning schedule for gradient estimator of $\texttt{CVaR}$ controls rapid movement of the iterates and is resilient against environment dynamics. When compared against the non-adaptive case Figure 9(c) and (d), we clearly see increased movement as the initial learning rate decreases depicting high susceptibility to the initial choice of the learning rate. The values explicitly refers to $\texttt{CVaR}$ objective function under consideration and the rewards depict rewards obtained policy the $\texttt{CVaR}$ based policy.

### 6.4 Glycemic Control

Type 1 diabetes mellitus (DM1) is a chronic disease characterized by the body's inability to produce insulin, a hormone essential for regulating blood glucose levels. Patients with DM1 must carefully monitor their glucose levels and administer insulin exogenously to maintain homeostasis. Effective risk management is critical for these patients, as they face the constant threat of hypo and hyperglycemic episodes that can have severe consequences if not properly controlled. We demonstrate the capability of our algorithm to manage high-risk situations in the administration of insulin to T1DM patients. We employ an artificial pancreatic simulator (Man et al., 2014) that exogenously administers insulin via a controller, allowing us to test various control algorithms in a controlled environment before deployment in clinical settings. To accomplish this we use the Simglucose framework (Xie, 2018), which is designed to mimic real-world scenarios encountered by patients with diabetes. In the this environment, a proportional-integral-derivative (PID) controller is used to regulate insulin administration, adjusting the dosage based on the patient's blood glucose levels to maintain them within a safe range (Clarke & Kovatchev, 2009).

The environment is modeled as a Markov Decision Process (MDP), with the state space consisting of multiple noisy glucose measurements at various time points in the past, carbohydrate intakes, and other relevant patient information. The action is the amount of insulin to be administered, a scalar value. The reward function takes the current and previous Continous Glucose Monitor (CGM) values along with the current insulin intake value. The reward function has a total of three components. First component is the negative exponent of the difference of risk index values of two CGM inputs which ensures to have decreasing risk index. Second component is the negative square of the risk index of the current CGM value indicative that patient has healthy glucose value and the third component is a conditional value, it is added to the whole reward when CGM is not in normal state. By minimizing the $\texttt{CVaR}$ of the glucose levels, we ensure that not only is the average risk low, but the probability of extreme high-risk events is also minimized.

We carry out experiments on two patient profiles - one adolescent 10 and two adults Figure 11 - with the objective of minimizing the risk of the patient entering hypo- or hyperglycemic levels with risk sensitivity level at $\rho = 0.01$. The patient profiles are given below where, CF: Carbohydrate Factor, CR: Carb Ratio (often referred to as Carb-to-Insulin Ratio) and TDI: Total Daily Insulin

| Name | CR | CF | Age | TDI |
|------|-----|-----|-----|-----|
| adolescent#001 | 12 | 15.0360 | 18 | 36.7339 |
| adult#001 | 10 | 8.77310657487 | 61 | 50.416652 |
| adult#002 | 8 | 9.2128 | 65 | 57.8688 |

Table 4: Patient profiles

We observe from our experiments that our algorithm is able to keep the patient at admissible levels of risk. There are instances where the glucose breaches into the hypo- or hyperglycemic regions, but the policy is able to course-correct and maintain a stable condition. This is crucial, as even brief excursions outside the target glucose range can have serious consequences for patients with DM1.

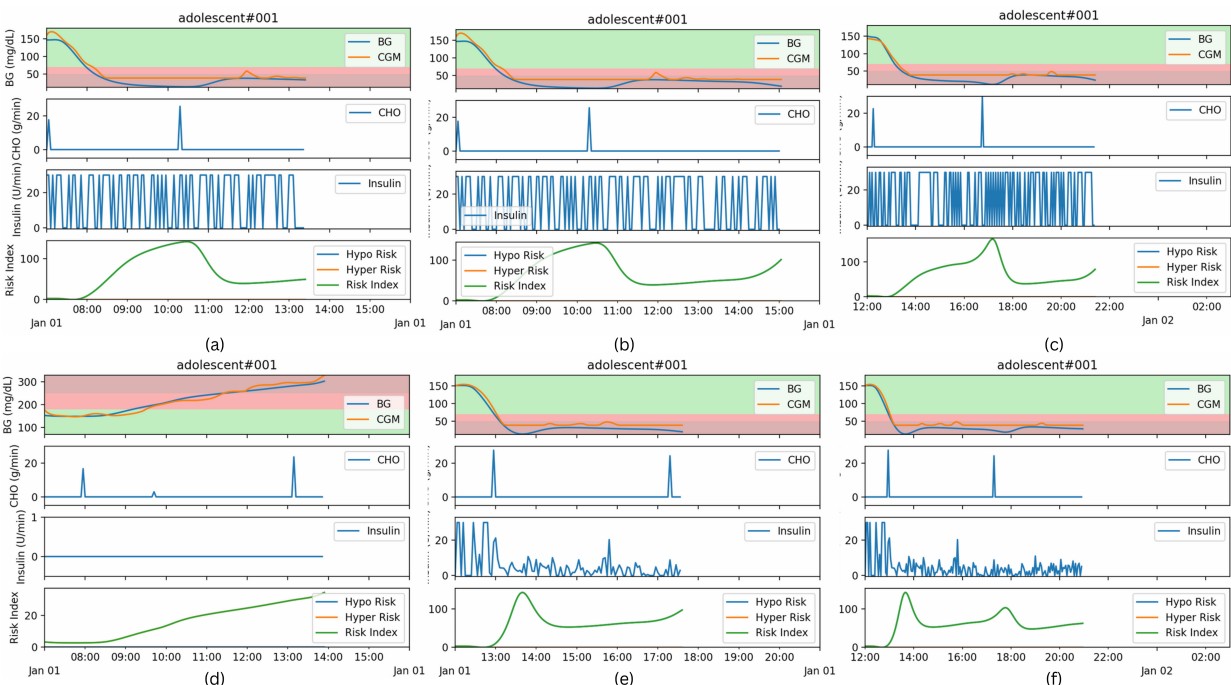

Figure 10: For patient profile adolescent#001 tests carried out over administration of insulin under constrained risk index which is estimated by our algorithm we observe the patient is stable for more than a day.

In our experiment we focus on the patient profiles given by Table 4 where the behavior of adolescent#001 is given by Figure 10 and the adults are given by Figure 11. From Figure 10, we see that the adolescent survives entire day which is depicted by the top row after which it dies despite movement of the glucose level to risky zones it adapts to it and prolongs the longevity. Similar behaviour is observed in the adults in Figure 11. The custom reward function for the controller is designed to incentivize maintaining blood glucose (BG) levels within a target range. Specifically, if the BG level in the last hour exceeds 180 mg/dL, the controller incurs a penalty of $-1$; if it drops below 70 mg/dL, the penalty is $-2$. Conversely, maintaining BG levels within the target range yields a reward of $+1$. This reward structure aligns with the clinical objective of avoiding hyperglycemia and hypoglycemia while promoting stable glucose control.

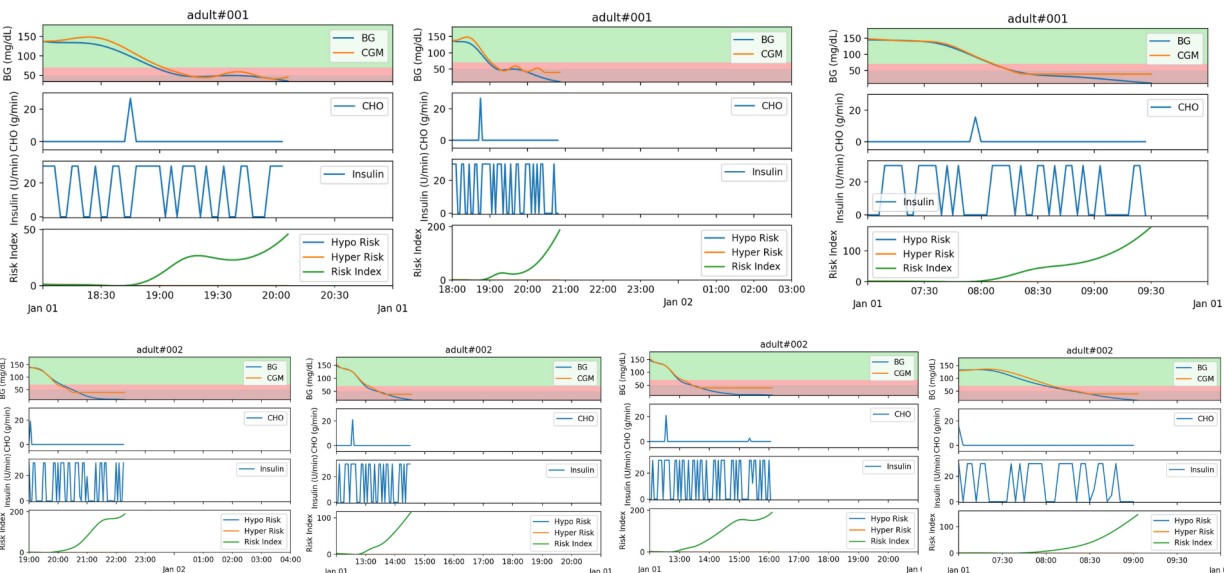

Figure 11: For patient profiles `adult#001` and `adult#002` similar tests carried out over administration of insulin.

**Meal Timing and Carbohydrate Intake:** To simulate realistic daily scenarios, meal timings and carbohydrate (CHO) intake are modeled using predefined scenarios. The meal schedule includes five instances: 7:00 AM (45 g CHO), 12:00 PM (70 g CHO), 4:00 PM (15 g CHO), 6:00 PM (80 g CHO), and 11:00 PM (10 g CHO). The specific timing and amount of CHO intake reflect typical meal patterns and are designed to challenge the controller's ability to manage varying glucose dynamics effectively. Each simulation runs for a full day of simulated time, with time steps representing 5-minute intervals. This fine granularity enables the accurate modeling of BG dynamics and controller responses. For meal scenarios, carbohydrate intake data is based on predefined patterns. The experimental setup evaluates the controller's ability to maintain BG levels within clinical targets under varying conditions, accounting for the unique physiological characteristics of each patient profile and the challenges posed by predefined meal scenarios.

The results of our experiments demonstrate the efficacy of our algorithm in managing the high-risk scenarios encountered by patients with DM1. The use of a controlled simulator environment, coupled with rigorous risk estimation techniques, allows us to develop and validate insulin administration policies that can be safely implemented in clinical settings.

## 7 Conclusion

This work introduces efficient algorithms for estimating extreme Value at Risk (VaR) and Conditional Value at Risk (CVaR). Our incremental and adaptive methods improve accuracy and computational efficiency in high-risk domains such as finance, healthcare, and robotics. The proposed single-pass variance reduction technique and the multi-time scale optimization approach offer efficient solutions to the challenges of extreme risk estimation. Theoretical and empirical analyses validate the effectiveness of these approaches. These advances contribute to more robust risk management practices in complex, dynamic systems, providing decision-makers with improved tools to assess and mitigate extreme risks.

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

**Appendix**

**Proof of Proposition 1:**

*Proof.* For $\lambda \in [0, 1]$ and $\gamma_1, \gamma_2 \in [\phi_l, \phi_u]$ with $\gamma_1 < \gamma_2$, we have

$$
\int \{\psi(\phi(\mathbf{X}), \lambda\gamma_1 + (1-\lambda)\gamma_2)\}d\mathbb{P}_x = \int \{(1-\rho)(\phi(\mathbf{X}) - \lambda\gamma_1 - (1-\lambda)\gamma_2)\mathbb{I}(\phi(\mathbf{X}), \lambda\gamma_1 + (1-\lambda)\gamma_2)^+ +
$$
$$
\rho(\lambda\gamma_1 + (1-\lambda)\gamma_2 - \phi(\mathbf{X}))\mathbb{I}(\phi(\mathbf{X}), \lambda\gamma_1 + (1-\lambda)\gamma_2)^-\}d\mathbb{P}_x
$$
$$
= \int \lambda(1-\rho)(\phi(\mathbf{X}) - \gamma_1)\mathbb{I}(\phi(\mathbf{X}), \gamma_1)^+ - \lambda(1-\rho)(\phi(\mathbf{X}) - \gamma_1)\mathbb{I}(\phi(\mathbf{X}), \gamma_1)^+\mathbb{I}(\phi(\mathbf{X}), \lambda\gamma_1 + (1-\lambda)\gamma_2)^- +
$$
$$
\lambda\rho(\gamma_1 - \phi(\mathbf{X}))\mathbb{I}(\phi(\mathbf{X}), \gamma_1)^- + \lambda\rho(\gamma_1 - \phi(\mathbf{X}))\mathbb{I}(\phi(\mathbf{X}), \gamma_1)^+\mathbb{I}(\phi(\mathbf{X}), \lambda\gamma_1 + (1-\lambda)\gamma_2)^- +
$$
$$
(1-\lambda)(1-\rho)(\phi(\mathbf{X}) - \gamma_2)\mathbb{I}(\phi(\mathbf{X}), \gamma_2)^+ + (1-\lambda)(1-\rho)(\phi(\mathbf{X}) - \gamma_2)\mathbb{I}(\phi(\mathbf{X}), \lambda\gamma_1 + (1-\lambda)\gamma_2)^+\mathbb{I}(\phi(\mathbf{X}), \gamma_2)^- 
$$
$$
+ (1-\lambda)\rho(\gamma_2 - \phi(\mathbf{X}))\mathbb{I}(\phi(\mathbf{X}), \gamma_2)^- - (1-\lambda)\rho(\gamma_2 - \phi(\mathbf{X}))\mathbb{I}(\phi(\mathbf{X}), \lambda\gamma_1 + (1-\lambda)\gamma_2)^+\mathbb{I}(\phi(\mathbf{X}), \gamma_2)^- d\mathbb{P}_x
$$
$$
= \lambda\int (1-\rho)(\phi(\mathbf{X}) - \gamma_1)\mathbb{I}(\phi(\mathbf{X}), \gamma_1)^+ + \rho(\gamma_1 - \phi(\mathbf{X}))\mathbb{I}(\phi(\mathbf{X}), \gamma_1)^- d\mathbb{P}_x +
$$
$$
(1-\lambda)\int (1-\rho)(\phi(\mathbf{X}) - \gamma_2)\mathbb{I}(\phi(\mathbf{X}), \gamma_2)^+ + \rho(\gamma_2 - \phi(\mathbf{X}))\mathbb{I}(\phi(\mathbf{X}), \gamma_2)^- d\mathbb{P}_x -
$$
$$
\int \lambda(\phi(\mathbf{X}) - \gamma_1)\mathbb{I}(\phi(\mathbf{X}), \gamma_1)^+\mathbb{I}(\phi(\mathbf{X}), \lambda\gamma_1 + (1-\lambda)\gamma_2)^- d\mathbb{P}_x +
$$
$$
\int (1-\lambda)(\phi(\mathbf{X}) - \gamma_2)\mathbb{I}(\phi(\mathbf{X}), \lambda\gamma_1 + (1-\lambda)\gamma_2)^+\mathbb{I}(\phi(\mathbf{X}), \gamma_2)^- d\mathbb{P}_x
$$
$$
= \lambda\int \psi(\phi(\mathbf{X}), \gamma_1)d\mathbb{P}_x + (1-\lambda)\int \psi(\phi(\mathbf{X}), \gamma_2)d\mathbb{P}_x - \int \lambda(\phi(\mathbf{X}) - \gamma_1)\mathbb{I}(\phi(\mathbf{X}), \gamma_1)^+\mathbb{I}(\phi(\mathbf{X}), \lambda\gamma_1 + (1-\lambda)\gamma_2)^+ +
$$
$$
(1-\lambda)(\gamma_2 - \phi(\mathbf{X}))\mathbb{I}(\phi(\mathbf{X}), \lambda\gamma_1 + (1-\lambda)\gamma_2)^+\mathbb{I}(\phi(\mathbf{X}), \gamma_2)^- d\mathbb{P}_x
$$
$$
\leqslant \lambda\int \psi(\phi(\mathbf{X}), \gamma_1)d\mathbb{P}_x + (1-\lambda)\int \psi(\phi(\mathbf{X}), \gamma_2)d\mathbb{P}_x.
$$

This completes the proof of Proposition 1. $\qquad\square$

**Proof of Proposition 2:**

*Proof.* From the definition of variance, we have,

$$
\mathbb{V}_{Q_t}\left[\widehat{\ell}_{t+1}\right] = \int \widehat{\ell}_{t+1}^2 dQ_t - \left(\int \widehat{\ell}_{t+1}dQ_t\right)^2
$$

Let $G := \{x \in \mathbb{R}^d : \frac{dQ_t}{d\nu}(x) \neq 0\}$. Now, from Eq. (15), we get

$$
\mathbb{V}_{Q_t}\left[\widehat{\ell}_{t+1}\right] = \int \frac{(d\mathbb{P}/d\nu)^2}{(dQ_t/d\nu)^2}\mathbb{I}^+(\phi(\mathbf{X}), \gamma_t)\frac{dQ_t}{d\nu}d\nu - \ell_{t+1}^2
$$
$$
= \int_G \frac{(d\mathbb{P}/d\nu)^2}{(dQ_t/d\nu)^2}\mathbb{I}^+(\phi(\mathbf{X}), \gamma_t)\frac{dQ_t}{d\nu}d\nu + \underbrace{\int_{G^c} \frac{(d\mathbb{P}/d\nu)^2}{(dQ_t/d\nu)^2}\mathbb{I}^+(\phi(\mathbf{X}), \gamma_t)dQ_t}_{= 0 \text{ since } Q_t(G^c) = 0} - \ell_{t+1}^2
$$
$$
= \int_G \frac{(d\mathbb{P}/d\nu)^2}{dQ_t/d\nu}\mathbb{I}^+(\phi(\mathbf{X}), \gamma_t)d\nu + \int_G \frac{\ell_{t+1}^2(dQ_t/d\nu)^2}{(dQ_t/d\nu)}d\nu - 2\int_G \frac{(d\mathbb{P}/d\nu)(dQ_t/d\nu)\ell_{t+1}}{(dQ_t/d\nu)}\mathbb{I}^+(\phi(\mathbf{X}), \gamma_t)d\nu
$$
$$
\tag{55}
$$

The last equality follows since

$$\int_G \frac{(d\mathbb{P}/d\nu)(dQ_t/d\nu)\ell_{t+1}}{dQ_t/d\nu}\mathbb{I}^+(\phi(\mathbf{X}),\gamma_t)d\nu = \ell_{t+1}^2 \text{ and } \int_G \frac{\ell_{t+1}^2(dQ_t/d\nu)^2}{(dQ_t/d\nu)}d\nu = \int_G \ell_{t+1}^2\frac{dQ_t}{d\nu}d\nu = \ell_{t+1}^2.$$

Therefore, from Eq. (55), we get

$$\mathbb{V}_{Q_t}\left[\hat{\ell}_{t+1}\right] = \int_G \frac{((d\mathbb{P}_x/d\nu)\mathbb{I}^+(\phi(\mathbf{X}),\gamma_t) - \ell_{t+1}(dQ_t/d\nu))^2}{(dQ_t/d\nu)}d\nu$$

$$= \int \frac{((d\mathbb{P}_x/d\nu)\mathbb{I}^+(\phi(\mathbf{X}),\gamma_t) - \ell_{t+1}(dQ_t/d\nu))^2}{(dQ_t/d\nu)}d\nu \text{ (Follows as integral is zero in } G^c)$$

$$\square$$

**Proof of Lemma 6:**

*Proof.* Here, we prove the asymptotic analysis of the sequence $\{\eta_t\}_{t\in\mathbb{N}}$ conditioned on $\gamma_t \equiv \gamma$ and $\theta_t \equiv \theta$. We recall here the stochastic recursion of $\eta_t$ from Step 6 of Algorithm 1 with $\gamma_t = \gamma$ and $\theta_t = \theta$:

$$\eta_{t+1} = \eta_t + \beta_t\zeta_\theta\left(\mathbf{Y}_{t+1}\right)(\mathbb{I}^+(\phi(\mathbf{Y}_{t+1}),\gamma)\Gamma(\mathbf{Y}_{t+1}) - \eta_t\mathbb{I}^+(\phi(\mathbf{Y}_{t+1}),\gamma)), \text{ where } \mathbf{Y}_{t+1} \sim Q_{\theta_t}$$

$$= \eta_t + \beta_t\zeta_\theta(\mathbf{Y}_{t+1})\left(h^1(\eta) + \mathbb{M}_{t+1}\right), \tag{56}$$

where,

$$\mathbb{M}_{t+1}^1 = \zeta_t\left(\mathbf{Y}_{t+1}\right)(\mathbb{I}^+(\phi(\mathbf{Y}_{t+1}),\gamma)\Gamma(\mathbf{Y}_{t+1}) - \eta_t\mathbb{I}^+(\phi(\mathbf{Y}_{t+1}),\gamma)) - $$
$$\mathbb{E}_\theta\left[\zeta_\theta\left(\mathbf{Y}_{t+1}\right)(\mathbb{I}^+(\phi(\mathbf{Y}_{t+1}),\gamma)\Gamma(\mathbf{Y}_{t+1}) - \eta_t\mathbb{I}^+(\phi(\mathbf{Y}_{t+1}),\gamma))|\mathcal{F}_t\right] \text{ and} \tag{57}$$

$$h^1(\eta) = \mathbb{E}_\theta\left[\zeta_\theta\left(\mathbf{Y}_{t+1}\right)(\mathbb{I}^+(\phi(\mathbf{Y}_{t+1}),\gamma)\Gamma(\mathbf{Y}_{t+1}) - \eta\mathbb{I}^+(\phi(\mathbf{Y}_{t+1}),\gamma))|\mathcal{F}_t\right]$$
$$= \mathbb{E}_\theta\left[\zeta_\theta\left(\mathbf{Y}_{t+1}\right)(\mathbb{I}^+(\phi(\mathbf{Y}_{t+1}),\gamma)\Gamma(\mathbf{Y}_{t+1}) - \eta\mathbb{I}^+(\phi(\mathbf{Y}_{t+1}),\gamma))\right] \tag{58}$$

The last equality follows since $\mathbf{Y}_{t+1}$ is independent of $\mathcal{F}_t$. Also, we have $\eta_t, t \geqslant 0$ is integrable. This can be shown by induction. The base case follows the assumption in the lemma. Now assume $\eta_t$ is integrable for $t > 0$. Now consider

$$\mathbb{E}\left[|\eta_{t+1}|\right] = \mathbb{E}\left[|\eta_t + \beta_t\zeta_t\left(\mathbf{Y}_{t+1}\right)(\mathbb{I}^+(\phi(\mathbf{Y}_{t+1}),\gamma)\Gamma(\mathbf{Y}_{t+1}) - \eta_t\mathbb{I}^+(\phi(\mathbf{Y}_{t+1}),\gamma))|\right]$$
$$\leqslant \mathbb{E}\left[|\eta_t|\right] + \mathbb{E}\left[|\beta_t\zeta_t(\mathbf{Y}_{t+1})\mathbb{I}^+(\phi(\mathbf{Y}_{t+1}),\gamma)\Gamma(\mathbf{Y}_{t+1})|\right] + \mathbb{E}\left[|\eta_t\mathbb{I}^+(\phi(\mathbf{Y}_{t+1}),\gamma)|\right]$$
$$\leqslant \mathbb{E}\left[|\eta_t|\right] + \beta_t\mathbb{E}_{\mathbb{P}_x}\left[|\Gamma(\mathbf{Y}_{t+1})|\right] + \mathbb{E}\left[|\eta_t|\right] < \infty \tag{59}$$
(by induction hypothesis and Assumption A2).

This establishes that $\mathbb{M}_t^1$ is well defined. Now, we employ the ODE based analysis of the stochastic approximation algorithms proposed in Chapter 2 of (Borkar, 2008) to study the limiting behavior of the stochastic sequence $\{\eta_t\}_{t\in\mathbb{N}}$, where we verify the necessary conditions (as prescribed by (Borkar, 2008)) required to establish the equivalence between the asymptotic behavior of the stochastic sequence $\{\eta_t\}_{t\in\mathbb{N}}$ to that of its deterministic flow induced by the associated ODE $\frac{d}{dt}\eta(t) = h^1(\eta(t))$. Then we study the qualitative behavior of the solutions of the associated ODE to identify the stable equilibrium points (which will also be the limit points of the sequence $\{\eta_t\}_{t\in\mathbb{N}}$ due to the settled equivalence).

**Part 1: To establish the equivalence between the stochastic recursion (56) and its associated ODE:**

To achieve this, one has to guarantee that the vector field $h^1$, the noise $\{\mathbb{M}_{t+1}^1\}_{t\in\mathbb{N}}$ and the stochastic sequence $\{\eta_t\}_{t\in\mathbb{N}}$ satisfy certain necessary conditions which are as follows:

- The vector field $h^1$ is Lipschitz continuous.

$$\|h^1(\eta_1) - h^1(\eta_2)\|_2 = \|\mathbb{E}_\theta \left[ \zeta_\theta \left( \mathbf{Y}_{t+1} \right) (\mathbb{I}^+(\phi(\mathbf{Y}_{t+1}), \gamma) \Gamma(\mathbf{Y}_{t+1}) - \eta_1 \mathbb{I}^+(\phi(\mathbf{Y}_{t+1}), \gamma)) \right] -$$
$$\mathbb{E}_\theta \left[ \zeta_\theta \left( \mathbf{Y}_{t+1} \right) (\mathbb{I}^+(\phi(\mathbf{Y}_{t+1}), \gamma) \Gamma(\mathbf{Y}_{t+1}) - \eta_2 \mathbb{I}^+(\phi(\mathbf{Y}_{t+1}), \gamma)) \right] \|_2$$
$$= \mathbb{P}_x(\phi(\mathbf{Y}_{t+1}) \geqslant \gamma) \|\eta_1 - \eta_2\|_2$$
$$\leqslant \|\eta_1 - \eta_2\|_2.$$

- $\{\mathbb{M}^1_{t+1}\}_{t\in\mathbb{N}}$ is a martingale difference noise sequence *w.r.t.* the filtration $\{\mathcal{F}_{t+1}\}_{t\in\mathbb{N}}$, *i.e.*, $\mathbb{M}^1_{t+1}$ is $\mathcal{F}_{t+1}-$measurable and integrable, $\forall t \in \mathbb{N}$ (follows from Eq. (59)). Also, for $t \geqslant 0$, we have

$$\mathbb{E}\left[\mathbb{M}^1_{t+1}|\mathcal{F}_t\right] = \mathbb{E}\Bigg[ \zeta_\theta \left( \mathbf{Y}_{t+1} \right) (\mathbb{I}^+(\phi(\mathbf{Y}_{t+1}), \gamma) \Gamma(\mathbf{Y}_{t+1}) - \eta_t \mathbb{I}^+(\phi(\mathbf{Y}_{t+1}), \gamma)) -$$
$$\mathbb{E}_\theta\left[ \zeta_\theta \left( \mathbf{Y}_{t+1} \right) (\mathbb{I}^+(\phi(\mathbf{Y}_{t+1}), \gamma) \Gamma(\mathbf{Y}_{t+1}) - \eta_t \mathbb{I}^+(\phi(\mathbf{Y}_{t+1}), \gamma)) | \mathcal{F}_t \right] \Bigg| \mathcal{F}_t \Bigg] = 0.$$

- From Assumption (A2) and the fact that $\widehat{Q}_t$ has finite first and second moments, $\forall t \in \mathbb{N}$, we get that $\exists K_2 \in (0, \infty)$, *s.t.* $\mathbb{E}\left[\|\mathbb{M}^0_{t+1}\|^2|\mathcal{F}_t\right] \leqslant K_2(1 + \|b_t\|^2), \forall t \in \mathbb{N}$.

$$\mathbb{E}\left[\|\mathbb{M}^1_{t+1}\|^2|\mathcal{F}_t\right] = \mathbb{E}\Bigg[ \Big\| \zeta_\theta \left( \mathbf{Y}_{t+1} \right) (\mathbb{I}^+(\phi(\mathbf{Y}_{t+1}), \gamma) \Gamma(\mathbf{Y}_{t+1}) - \eta_t \mathbb{I}^-(\phi(\mathbf{Y}_{t+1}), \gamma)) -$$
$$\mathbb{E}_\theta \left[ \zeta_\theta \left( \mathbf{Y}_{t+1} \right) (\mathbb{I}^+(\phi(\mathbf{Y}_{t+1}), \gamma) \Gamma(\mathbf{Y}_{t+1}) - \eta_t \mathbb{I}^-(\phi(\mathbf{Y}_{t+1}), \gamma)) | \mathcal{F}_t \right] \Big\|^2 \Bigg| \mathcal{F}_t \Bigg]$$
$$= \mathbb{E}_{\mathbb{P}_x}[\mathbb{I}^+(\phi(\mathbf{Y}_{t+1}), \gamma) \Gamma(\mathbf{Y}_{t+1})^\top \Gamma(\mathbf{Y}_{t+1})] -$$
$$\mathbb{E}_{\mathbb{P}_x}[\mathbb{I}^+(\phi(\mathbf{Y}_{t+1}), \gamma) \Gamma(\mathbf{Y}_{t+1})]^\top \mathbb{E}_{\mathbb{P}_x}[\mathbb{I}^+(\phi(\mathbf{Y}_{t+1}), \gamma) \Gamma(\mathbf{Y}_{t+1})] +$$
$$\mathbb{E}\left[\eta_t^\top \eta_t \mathbb{I}^+(\phi(\mathbf{Y}_{t+1}), \gamma)|\mathcal{F}_t\right] - \eta_t^\top \eta_t \mathbb{E}\left[\mathbb{I}^+(\phi(\mathbf{Y}_{t+1}), \gamma)|\mathcal{F}_t\right]$$
$$\leqslant 2\mathbb{E}_{\mathbb{P}_x}[\mathbb{I}^+(\phi(\mathbf{Y}_{t+1}), \gamma)\|\Gamma(\mathbf{Y}_{t+1})\|^2]$$
$$= 2\mathbb{E}_{\mathbb{P}_x}[\mathbb{I}^+(\phi(\mathbf{Y}), \gamma)\|\Gamma(\mathbf{Y})\|^2]$$
$$\text{Hence } \sup_t \mathbb{E}\left[\|\mathbb{M}^1_{t+1}\|^2|\mathcal{F}_t\right] \leqslant 2\mathbb{E}_{\mathbb{P}_x}[\|\Gamma(\mathbf{Y})\|^2] < \infty$$

- By appealing to the Borkar-Meyn thereom (Borkar & Meyn, 2000) (Theorem 7 of Chapter 3 in (Borkar, 2008)), one can show that the iterates $\eta_t$ are stable, *i.e.*, $\{\eta_t\}_{t\in\mathbb{N}}$ is bounded almost surely. The Borkar-Meyn stability theorem claims that iterates almost surely remain inside a bounded set when the dynamics of the flow induced by the dominant component of the vector field $h^0$ is globally asymptotically stable at the origin. Indeed, the flow of the dominating component is defined as the following limiting ODE:

$$\frac{d}{dt}\eta(t) = h^1_\infty(\eta(t)) := \lim_{r\to\infty} \frac{h^1(r\eta(t))}{r}, \quad t \geqslant 0. \tag{60}$$

In our case, the above limit exists and we have

$$h^1_\infty(\eta) = \lim_{r\to\infty} \frac{1}{r} \mathbb{E}_\theta \left[ \zeta_\theta \left( \mathbf{Y}_{t+1} \right) (\mathbb{I}^+(\phi(\mathbf{Y}_{t+1}), \gamma) \Gamma(\mathbf{Y}_{t+1}) - r\eta \mathbb{I}^+(\phi(\mathbf{Y}_{t+1}), \gamma)) \right]$$
$$= -\mathbb{P}_x(\phi(\mathbf{Y}) \geqslant \gamma) \mathbb{I}_{d\times d} \eta. \tag{61}$$

It is now easy to verify that the limiting ODE (61) is globally asymptotically stable at origin (since all the eigenvalues of the diagonal matrix $-\mathbb{P}_x(\phi(\mathbf{Y}) \geqslant \gamma)\mathbb{I}_{d\times d}$ are negative, real numbers) as required. Hence,

$$\sup_t \|\eta_t\| < \infty \ a.s.$$

Since the stochastic recursion (56) confirms the hypothesis of Corollary 4 in Chapter 2 of (Borkar, 2008), now, by appealing to the said corollary, we conclude that the limiting behavior of the stochastic recursion (56) is equivalent to the limiting behavior of the flow induced by the following ODE:

$$
\begin{aligned}
\frac{d}{dt}\eta(t) &= h^1(\eta(t)), \quad t \geq 0 \\
&= \mathbb{E}_\theta \left[ \zeta_\theta \left( \mathbf{Y} \right) (\mathbb{I}^+(\phi(\mathbf{Y}), \gamma)\Gamma(\mathbf{Y}) - \eta(t)\mathbb{I}^+(\phi(\mathbf{Y}), \gamma)) \right] \\
&= \mathbb{E}_{\mathbb{P}_x} \left[ \mathbb{I}^+(\phi(\mathbf{Y}), \gamma)\Gamma(\mathbf{Y}) \right] - \eta(t)\mathbb{E}_{\mathbb{P}_x} \left[ \mathbb{I}^+(\phi(\mathbf{Y}), \gamma) \right] \\
&= \mathbb{E}_{\mathbb{P}_x} \left[ \mathbb{I}^+(\phi(\mathbf{Y}), \gamma)\Gamma(\mathbf{Y}) \right] - \eta(t)\mathbb{P}_x \left( \phi(\mathbf{Y} \geq \gamma) \right).
\end{aligned}
\tag{62}
$$

**Part 1.2: Qualitative analysis of the limiting behavior of the associated ODE (62):**
For brevity, we rewrite the ODE (62) as follows:

$$
\frac{d}{dt}\eta(t) = D\eta(t) + \mathbb{E}_{\mathbb{P}_x} \left[ \mathbb{I}^+(\phi(\mathbf{Y}), \gamma)\Gamma(\mathbf{Y}) \right], \quad t \geq 0,
\tag{63}
$$

where $D$ is a diagonal matrix with $D_{ii} = -\mathbb{P}_x \left( \phi(\mathbf{Y} \geq \gamma) \right)$, $1 \leq \forall i \leq d$. Observe that the ODE (63) is a linear, first-order ODE and therefore, the stability of the stationary point $-D^{-1}\mathbb{E}_{\mathbb{P}_x} \left[ \mathbb{I}^+(\phi(\mathbf{Y}), \gamma)\Gamma(\mathbf{Y}) \right]$ (obtained by equating $\frac{d}{dt}\eta(t)$ to 0) is entirely characterized by the nature of the eigen-values of $D$. Now, since all the eigen-values of $D$ are negative real numbers (follows from the definition of the diagonal matrix $D$), we deduce that $-D^{-1}\mathbb{E}_{\mathbb{P}_x} \left[ \mathbb{I}^+(\phi(\mathbf{Y}), \gamma)\Gamma(\mathbf{Y}) \right]$ is a globally asymptotically stable equilibrium point of the ODE (63). Finally, by appealing to the previously established asymptotic equivalence from Part 1.1 between the stochastic recursion (56) and the ODE (63), we obtain the following result irrespective of the initial value $b(0)$ of the flow (63):

$$
\begin{aligned}
\lim_{t \to \infty} \eta_t = \lim_{t \to \infty} \eta(t) &= -D^{-1}\mathbb{E}_{\mathbb{P}_x} \left[ \mathbb{I}^+(\phi(\mathbf{Y}), \gamma)\Gamma(\mathbf{Y}) \right] \quad a.s. \\
&= \frac{\mathbb{E}_{\mathbb{P}_x} \left[ \mathbb{I}^+(\phi(\mathbf{Y}), \gamma)\Gamma(\mathbf{Y}) \right]}{\mathbb{P}_x \left( \phi(\mathbf{Y} \geq \gamma) \right)} \quad a.s.
\end{aligned}
$$

**Part 2: Proof of convergence of $\bar{\eta}_t$:**
We recall the stochastic recursion corresponding to $\bar{\eta}_t$ here:

$$
\bar{\eta}_{t+1} = \frac{1}{(t+1)^c} \left( t^c \bar{\eta}_t + \phi(\mathbf{Y}_{t+1}) \right), \quad \text{where } \bar{\eta}_0 = 0.
\tag{64}
$$

Using induction one can unfold the recursion to get,

$$
\bar{\eta}_t = \frac{1}{t^c} \sum_{k=1}^t \phi(\mathbf{Y}_k)
\tag{65}
$$

By Birnbaum–Raymond–Zuckerman inequality, we have, for $\epsilon > 0$,

$$
\mathbb{P}(\|\bar{\eta}_t - \mathbb{E}_\theta \left[ \phi(\mathbf{Y}) \right] \| \geq \epsilon) \leq \epsilon^{-1} t^{1-2c} \mathbb{E}_\theta \left[ \|\phi(\mathbf{Y}) - \mathbb{E}_\theta \left[ \phi(\mathbf{Y}) \right] \|^2 \right].
\tag{66}
$$

Since $c > 1$, we have $1 - 2c < -1$. Hence,

$$
\lim_{t \to \infty} \mathbb{P}(\|\bar{\eta}_t - \mathbb{E}_\theta \left[ \phi(\mathbf{Y}) \right] \| \geq \epsilon) = 0.
\tag{67}
$$

This implies that $\bar{\eta}_t$ converges to $\mathbb{E}_\theta \left[ \Gamma(\mathbf{X}) \right]$ in probability.

Consider the event $F_t = \{ \|\bar{\eta}_t - \mathbb{E}_\theta \left[ \phi(\mathbf{Y}) \right] \| \geq \epsilon \}$. Now, we have

$$
\begin{aligned}
\sum_{t=0}^\infty \mathbb{P}(F_t) &\leq \sum_{t=1}^\infty \frac{1}{t^{2c-1}} \epsilon^{-1} \mathbb{E}_\theta \left[ \|\phi(\mathbf{Y}) - \mathbb{E}_\theta \left[ \phi(\mathbf{Y}) \right] \|^2 \right] \\
&= \epsilon^{-1} \mathbb{E}_\theta \left[ \|\phi(\mathbf{Y}) - \mathbb{E}_\theta \left[ \phi(\mathbf{Y}) \right] \|^2 \right] \sum_{t=1}^\infty \frac{1}{t^{2c-1}} \\
&< \infty \text{ (Since } 2c - 1 > 1).
\end{aligned}
\tag{68}
$$

Also,

$$\mathbb{P}\left(\limsup_t F_t\right) = \mathbb{P}\left(\bigcap_{t=1}^{\infty}\bigcup_{k=t}^{\infty} F_k\right) = \lim_{t\to\infty}\mathbb{P}\left(\bigcup_{k=t}^{\infty} F_k\right) \leqslant \lim_{t\to\infty}\sum_{k=t}^{\infty}\mathbb{P}(F_k) = 0 \text{ (From Eq. (68)).} \tag{69}$$

This implies that

$$\mathbb{P}\left(\{\omega \mid \exists N_\omega \ s.t., \ \forall t \geqslant N_\omega, \|\bar{\eta}_t(\omega) - \mathbb{E}_\theta\left[\phi(\mathbf{Y})\right]\| \geqslant \epsilon\}\right) = 0$$
$$\Rightarrow \mathbb{P}\left(\{\omega \mid \exists N_\omega \ s.t., \ \forall t \geqslant N_\omega, \|\bar{\eta}_t(\omega) - \mathbb{E}_\theta\left[\phi(\mathbf{Y})\right]\| < \epsilon\}\right) = 1. \tag{70}$$

Since $\epsilon > 0$ is arbitrary, we have,

$$\mathbb{P}\left(\{\omega \mid \lim_{t\to\infty}\bar{\eta}_t(\omega) = \mathbb{E}_\theta\left[\phi(\mathbf{Y})\right]\}\right) = 1. \tag{71}$$

Hence,

$$\lim_{t\to\infty}\bar{\eta}_t = \mathbb{E}_\theta\left[\phi(\mathbf{Y})\right] \ \text{almost surely.} \tag{72}$$

$$\square$$

**Proof of Theorem 2:**

*Proof.* Here, for easy reference, we recall Step 6 of Algorithm 1, *viz.*,

$$\gamma_{t+1} = \gamma_t - \alpha_t\zeta_t(\widehat{\mathbf{X}}_{t+1})\Delta_t^\gamma(\widehat{\mathbf{X}}_{t+1}), \ \text{where } \widehat{\mathbf{X}}_{t+1} \sim \widehat{Q}_t.$$

The above equation can be further viewed as

$$\gamma_{t+1} = \gamma_t + \alpha_t\left(\mathbb{M}_{t+1} - \mathbb{E}\left[\zeta_t(\widehat{\mathbf{X}}_{t+1})\Delta_t^\gamma(\widehat{\mathbf{X}}_{t+1})|\mathcal{F}_t\right]\right), \tag{73}$$

$$\text{where } \mathbb{M}_{t+1} := \mathbb{E}\left[\zeta_t(\widehat{\mathbf{X}}_{t+1})\Delta_t^\gamma(\widehat{\mathbf{X}}_{t+1})|\mathcal{F}_t\right] - \zeta_t(\widehat{\mathbf{X}}_{t+1})\Delta_t^\gamma(\widehat{\mathbf{X}}_{t+1}). \tag{74}$$

Since $\{\widehat{\mathbf{X}}_{t+1}\}_{t\in\mathbb{N}}$ is independent, we get

$$\mathbb{E}\left[\zeta_t(\widehat{\mathbf{X}}_{t+1})\Delta_t^\gamma(\widehat{\mathbf{X}}_{t+1})|\mathcal{F}_t\right] = \mathbb{E}_{\widehat{Q}_t}\left[\zeta_t(\widehat{\mathbf{X}}_{t+1})\Delta_t^\gamma(\widehat{\mathbf{X}}_{t+1})\right]. \tag{75}$$

Note that $\widehat{Q}_t$ is obtained using a Bernoulli trial (See Step 4 of Algorithm 1). Hence, $\widehat{Q}_t$ has two choices: $Q_{\theta_t}$ or $\mathbb{P}_x$. Now for $\widehat{Q}_t = Q_{\theta_t}$, we get,

$$\mathbb{E}_{\widehat{Q}_t}\left[\zeta_t(\widehat{\mathbf{X}}_{t+1})\Delta_t^\gamma(\widehat{\mathbf{X}}_{t+1})\right] = \mathbb{E}_{Q_{\theta_t}}\left[\frac{d\mathbb{P}_x}{dQ_{\theta_t}}(\widehat{\mathbf{X}}_{t+1})\Delta_t^\gamma(\widehat{\mathbf{X}}_{t+1})\right] = \mathbb{E}_{\mathbb{P}_x}\left[\Delta_t^\gamma(\widehat{\mathbf{X}}_{t+1})\right].$$

Also, the above equality directly holds in the case of $\widehat{Q}_t = \mathbb{P}_x$.

A fortiori, $\forall t \in \mathbb{N}$, we have,

$$\mathbb{E}_{\widehat{Q}_t}\left[\zeta_t(\widehat{\mathbf{X}}_{t+1})\Delta_t^\gamma(\widehat{\mathbf{X}}_{t+1})\right] = \mathbb{E}_{\mathbb{P}_x}\left[\Delta_t^\gamma(\widehat{\mathbf{X}}_{t+1})\right]. \tag{76}$$

Therefore, the noise term $\mathbb{M}_{t+1}$ can be rewritten as:

$$\mathbb{M}_{t+1} = \mathbb{E}_{\mathbb{P}_x}\left[\Delta_t^\gamma(\widehat{\mathbf{X}}_{t+1})\right] - \zeta_t(\widehat{\mathbf{X}}_{t+1})\Delta_t^\gamma(\widehat{\mathbf{X}}_{t+1}). \tag{77}$$

Now, here we will investigate further the nature of the non-noise term $\mathbb{E}_{\mathbb{P}_x}\left[\Delta_t^\gamma(\widehat{\mathbf{X}}_{t+1})\right]$. We have

$$\mathbb{E}_{\mathbb{P}_x}\left[\Delta_t^\gamma(\widehat{\mathbf{X}}_{t+1})\right] \in \mathbb{E}_{\mathbb{P}_x}\left[\partial_\gamma\psi(\phi(\widehat{\mathbf{X}}_{t+1}), \gamma_t)\right], \text{ where} \tag{78}$$

$$\psi(\phi(x), \gamma) = (1-\rho)(\phi(x) - \gamma)\mathbb{I}(\phi(x), \gamma)^+ + \rho(\gamma - \phi(x))\mathbb{I}(\phi(x), \gamma)^- \tag{79}$$

and $\partial_\gamma\psi$ (the sub-differential of $\psi(\cdot, \gamma)$ w.r.t. $\gamma$) is a set-valued map and is defined as follows:

$$\partial_\gamma\psi(\phi(x), \gamma) = \begin{cases} -(1-\rho)\mathbb{I}(\phi(x), \gamma)^+ + \rho\mathbb{I}(\phi(x), \gamma)^-, \text{ for } \gamma \neq \phi(x), \\[2mm] [-(1-\rho), \rho], \text{ for } \gamma = \phi(x). \end{cases} \tag{80}$$

For brevity, let $h(\gamma) := -\mathbb{E}_{\mathbb{P}_x}\left[\partial_\gamma\psi(\phi(\mathbf{X}), \gamma)\right]$. (We consider here the r.v. $\mathbf{X}$ instead of $\widehat{\mathbf{X}}_{t+1}$ for notational convenience.)

Our primary objective in this proof is to analyze the limiting behaviour of the stochastic recursion (73). The analysis involves two parts:

1. To establish the equivalence between the stochastic recursion (73) and the associated differential inclusion (DI) given by $\frac{d}{dt}\gamma(t) \in h(\gamma(t)); t \geqslant 0$ and

2. To perform a qualitative analysis on the associated DI to identify the stable equilibrium points.

**Part 1:** To establish the equivalence between the stochastic recursion (73) and the associated differential inclusion (DI) given by $\frac{d}{dt}\gamma(t) \in h(\gamma(t)); t \geqslant 0$, we follow the framework provided in (Benaïm et al., 2005) and Chapter 5 of (Borkar, 2008). According to the framework, one has to guarantee that the set-valued map $h$ (which identifies the DI), the noise sequence $\{\mathbb{M}_{t+1}\}$ and the iterates $\{\gamma_t\}$ satisfy certain necessary conditions.

**(B1):** The preliminary step is to warrant that the associated DI is well-posed. To that end, we have to attest certain conditions on $h$. The set-valued map $h : [\phi_l, \phi_u] \to \{\text{subsets of } \mathbb{R}\}$ satisfies the following properties:

- For each $\gamma \in [\phi_l, \phi_u]$, $h(\gamma)$ is convex and compact: Indeed, it follows directly from Eq. (80). Note that for each $\gamma \in [\phi_l, \phi_u]$, $-h(\gamma)$ is either a singleton or the closed interval $[-(1-\rho), \rho]$.

- For each $\gamma \in [\phi_l, \phi_u]$, we have $\sup_{y \in h(\gamma)} |y| < K_1(1 + |\gamma|)$, for some $0 < K_1 < \infty$.

  Indeed, for each $\gamma \in [\phi_l, \phi_u]$, $-h(\gamma)$ is either the scalar $\mathbb{E}_{\mathbb{P}_x}\left[-(1-\rho)\mathbb{I}(\phi(\mathbf{X}), \gamma)^+ + \rho\mathbb{I}(\phi(\mathbf{X}), \gamma)^-\right]$ or the bounded closed interval $[-(1-\rho), \rho]$. Hence the above bound exists.

- $h$ is upper semi-continuous.

  To prove this, one has to show the following: if the sequence $\{\gamma_n\}$ converges to $\bar{\gamma}$ and any sequence $\{y_n\}$ converges to $\bar{y}$ where $y_n \in h(\gamma_n), \forall n$, then $\bar{y} \in h(\bar{\gamma})$. Note that for each $\gamma \in [\phi_l, \phi_u]$, there are two possibilities for $-h(\gamma)$. It is either $\mathbb{E}_{\mathbb{P}_x}\left[-(1-\rho)\mathbb{I}(\phi(\mathbf{X}), \gamma)^+ + \rho\mathbb{I}(\phi(\mathbf{X}), \gamma)^-\right]$ or the closed interval $[-(1-\rho), \rho]$. Also,

$$\int -(1-\rho)\mathbb{I}(\phi(\mathbf{X}), \gamma)^+ + \rho\mathbb{I}(\phi(\mathbf{X}), \gamma)^- d\mathbb{P}_x$$
$$= -(1-\rho)\mathbb{P}_x(\phi(\mathbf{X}) \geqslant \gamma) + \rho\mathbb{P}_x(\phi(\mathbf{X}) \leqslant \gamma)$$
$$\in [-(1-\rho), \rho]. \tag{81}$$

Now consider the case when $-y_n = -h(\gamma_n) = \int -(1-\rho)\mathbb{I}(\phi(\mathbf{X}), \gamma_n)^+ + \rho\mathbb{I}(\phi(\mathbf{X}), \gamma_n)^- d\mathbb{P}_x$, then $-y_n = -(1-\rho)\mathbb{P}_x(\phi(\mathbf{X}) \geqslant \gamma_n) + \rho\mathbb{P}_x(\phi(\mathbf{X}) \leqslant \gamma_n)$ converges to $-\bar{y} = -(1-\rho)\mathbb{P}_x(\phi(\mathbf{X}) \geqslant \bar{\gamma}) +$

$\rho \mathbb{P}_x(\phi(\mathbf{X}) \leqslant \bar{\gamma})$. This follows from the continuity of probability measures (Billingsley, 2013). Now from Eq. (81), we have $-\bar{y} \in -h(\bar{\gamma})$, *i.e.*, $\bar{y} \in h(\bar{\gamma})$.

Now consider the case when $-y_n \in [-(1-\rho), \rho]$ and $-\bar{y} = \mathbb{E}_{\mathbb{P}_x} [-(1-\rho)\mathbb{I}(\phi(\mathbf{X}), \gamma)^+ + \rho \mathbb{I}(\phi(\mathbf{X}), \gamma)^-]$. This implies that $\psi(\cdot, \gamma)$ is differentiable at $\gamma = \bar{\gamma}$, while only sub-differentials exist at $\gamma = \gamma_n, \forall n \in \mathbb{N}$. This particular scenario is not possible. The reason being $\psi$ is piece-wise linear in $\gamma$ and $\psi(\cdot, \gamma)$ is differentiable at $\gamma = \bar{\gamma}$. Therefore, there exists a neighbourhood around $\bar{\gamma}$ such that $\psi(\cdot, \gamma)$ is linear. However, by hypothesis $\{\gamma_n\} \to \bar{\gamma}$ which is impossible due to the linear behaviour of $\psi$ around $\bar{\gamma}$ and the non-differentiability of $\psi$ at each $\gamma_n$.

**(B2):** Further, we have to attest certain conditions on the noise term $\mathbb{M}_{t+1}$ (defined in Eq. (74)). The noise term $\mathbb{M}_{t+1}$ satisfies the following properties:

- $\{\mathbb{M}_{t+1}, t \in \mathbb{N}\}$ is a martingale difference noise sequence, *i.e.*, $\mathbb{M}_t$ is $\mathcal{F}_t$-measurable $\forall t \in \mathbb{N}\setminus\{0\}$ and is integrable. Also, $\mathbb{E}[\mathbb{M}_{t+1}|\mathcal{F}_t] = 0$ *a.s.*, $\forall t \in \mathbb{N}$.

- Since $\Delta_t^\gamma(\widehat{\mathbf{X}}_{t+1})$ is bounded *a.s.*, we find that $\Delta_t^\gamma(\widehat{\mathbf{X}}_{t+1})$ has finite first and second order moments. Hence,

$$\mathbb{E}\left[|\mathbb{M}_{t+1}|^2 | \mathcal{F}_t\right] \leqslant K_2(1 + |\gamma_t|^2), \;\; 0 < K_2 < \infty.$$

**(B3):** Finally, we establish the stability (almost sure boundedness) of the sequence $\{\gamma_t\}$, *i.e.*, $\sup_{t \in \mathbb{N}} |\gamma_t| < \infty$ *a.s.*. Note that $\phi(x) \in [\phi_l, \phi_u], \forall x$. At first, we consider the case when $\gamma_{t_0} > \phi_u$, for some $t_0 \in \mathbb{N}$. Hence, from Step 6 of Algorithm 1, we have

$$\gamma_{t_0+1} = \gamma_{t_0} - \alpha_{t_0} \zeta_{t_0}(\widehat{\mathbf{X}}_{t_0+1})\left(\rho\mathbb{I}(\phi(\widehat{\mathbf{X}}_{t_0+1}), \gamma_{t_0})^- - (1-\rho)\mathbb{I}(\phi(\widehat{\mathbf{X}}_{t_0+1}), \gamma_{t_0})\right)$$

$$= \gamma_{t_0} - \alpha_{t_0} \zeta_{t_0}(\widehat{\mathbf{X}}_{t_0+1})(\rho + 0)$$

$$= \gamma_{t_0} - \alpha_{t_0} \zeta_{t_0}(\widehat{\mathbf{X}}_{t_0+1})\rho. \tag{82}$$

The second equality follows since $\phi(\widehat{\mathbf{X}}_{t_0+1}) \in [\phi_l, \phi_u]$ and $\gamma_{t_0} > \phi_u$. Now, from Assumption (A1) we have $\sum_t \alpha_t = \infty$. By reason of this statement and the fact that $\zeta_t(\cdot, \cdot) > 0$, we conclude from Eq. (82) that there exists a $t_0' > t_0$ such that $\gamma_{t_0'} \leqslant \phi_u$. One can argue similarly to prove that when $\gamma_t < \phi_l$, for some $t \in \mathbb{N}$, then there exists a $t' > t$ such that $\gamma_{t'} \geqslant \phi_l$. This implies that whenever the iterates $\{\gamma_t\}$ leave the closed interval $[\phi_l, \phi_u]$, they eventually drift back towards the vicinity of the closed interval $[\phi_l, \phi_u]$ in finite time. Also, it is easy to verify that the upper bound on the leap the iterates $\{\gamma_t\}$ can generate outside of the closed interval $[\phi_l, \phi_u]$ is given by $Q \sup_t \alpha_t$, where $Q$ is defined in Assumption (A2). Hence,

$$\sup_{t \in \mathbb{N}} |\gamma_t| \leqslant \max\left\{|\phi_l - Q \sup_t \alpha_t|, |\phi_u + Q \sup_t \alpha_t|, |\gamma_0|\right\}$$

$$< \infty. \quad \text{(Follows from Assumption (A1))} \tag{83}$$

Now, by appealing to Theorem 2 in Chapter 5 of (Borkar, 2008) along with the results from (B1-B3), we deduce that the stochastic sequence $\{\gamma_t\}$ asymptotically tracks the following differential inclusion (DI)

$$\frac{d}{dt}\gamma(t) \in h(\gamma(t)) = -\mathbb{E}_{\mathbb{P}_x}\left[\partial_\gamma \psi(\phi(\mathbf{X}), \gamma(t))\right]$$

$$= -\partial_\gamma \mathbb{E}_{\mathbb{P}_x}\left[\psi(\phi(\mathbf{X}), \gamma(t))\right]. \tag{84}$$

Note that the interchange of $\mathbb{E}_{\mathbb{P}_x}[\cdot]$ and $\partial_\gamma$ in the above DI follows by appealing to the Dominated Convergence Theorem (Rubinstein & Shapiro, 1993).

**Part 2:** Now we perform a qualitative analysis of the above DI to identify its stable equilibrium points. For brevity, let $\gamma^* \triangleq \texttt{VaR}_\rho(\mathbb{P}_x)$.

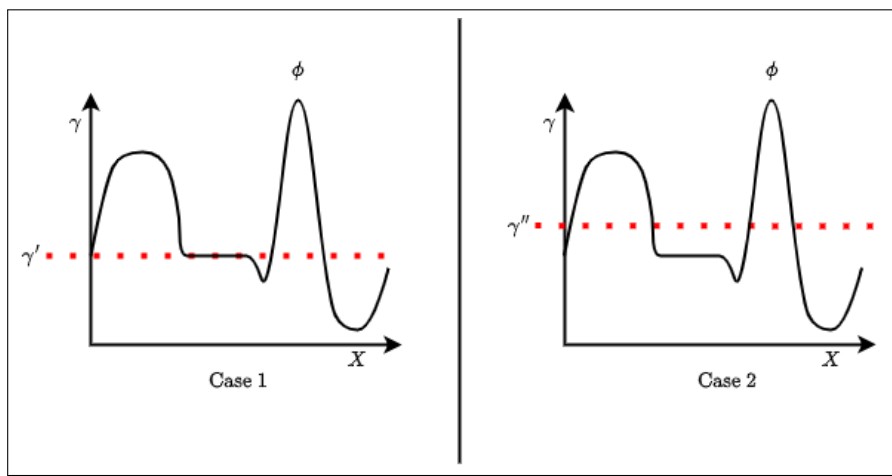

Figure 12: Two possible scenarios considered for the qualitative analysis of the DI (84).

Assume $\phi$ is not constant (If it is constant, then that constant value is the the unique solution of the DI (84)). There are two cases to consider here:

**Case 1:** Consider $\gamma' \in \{\gamma \in \mathbb{R} \mid \mathbb{P}_x(\phi(\mathbf{X}) = \gamma) > 0\}$. This case is illustrated in Case 1 of Figure 12. Note that $h(\gamma') = -[-(1 - \rho), \rho]$ (follows from Eq. (80)). Since $0 \in h(\gamma')$, we find that $\gamma'$ is an equilibrium point of the DI (84). We now conduct a phase space analysis in the neighbourhood of $\gamma'$ to understand the nature of the stability of the equilibrium point $\gamma'$. To do this, choose ${}^u\gamma' > \gamma'$. At ${}^u\gamma'$, we have

$$
\begin{aligned}
h({}^u\gamma') &= -\mathbb{E}_{\mathbb{P}_x}\left[-(1-\rho)\mathbb{I}(\phi(\mathbf{X}), {}^u\gamma')^+ + \rho\mathbb{I}(\phi(\mathbf{X}), {}^u\gamma')^-\right] \\
&= (1-\rho)\mathbb{P}_x\left(\phi(\mathbf{X}) \geqslant {}^u\gamma'\right) - \rho\mathbb{P}_x\left(\phi(\mathbf{X}) \leqslant {}^u\gamma'\right) \\
&= \mathbb{P}_x\left(\phi(\mathbf{X}) \geqslant {}^u\gamma'\right) - \rho.
\end{aligned}
$$

The sign of $\mathbb{P}_x\left(\phi(\mathbf{X}) \geqslant {}^u\gamma'\right) - \rho$ decides the direction of the drift of the DI at ${}^u\gamma'$. Again, there are three scenarios to consider here:

**(1a):** If $\gamma' > \gamma^*$, then $\mathbb{P}_x\left(\phi(\mathbf{X}) \geqslant {}^u\gamma'\right) < \rho$ (directly follows from the definition (1)) and hence $\mathbb{P}_x\left(\phi(\mathbf{X}) \geqslant {}^u\gamma'\right) - \rho < 0$. So at ${}^u\gamma'$, the direction of the drift of the DI is towards $\gamma'$. Now we analyze the left neighbourhood of $\gamma'$. Choose ${}^l\gamma' \in (\gamma^*, \gamma')$. In this case, $\mathbb{P}_x\left(\phi(\mathbf{X}) \geqslant {}^l\gamma'\right) - \rho < 0$ (directly follows from the definition (1)). Hence the direction of the drift of the DI at ${}^l\gamma'$ is away from $\gamma'$. Hence $\gamma'$ is a saddle point and hence unstable. This scenario is illustrated in Figure 13(a).

**(1b):** If $\gamma' < \gamma^*$, then choose ${}^u\gamma' \in (\gamma', \gamma^*)$. Observe that $\mathbb{P}_x\left(\phi(\mathbf{X}) \geqslant {}^u\gamma'\right) \geqslant \rho$ (follows from the definition (1)) and hence $\mathbb{P}_x\left(\phi(\mathbf{X}) \geqslant {}^u\gamma'\right) - \rho \geqslant 0$. So at ${}^u\gamma'$, the direction of the drift of the DI is away from $\gamma'$. Now for the analysis of the left neighbourhood of $\gamma'$, we choose ${}^l\gamma' < \gamma'$. In this case, $\mathbb{P}_x\left(\phi(\mathbf{X}) \geqslant {}^l\gamma'\right) - \rho \geqslant 0$ (directly follows from the definition (1)). Hence the direction of the drift of the DI at ${}^l\gamma'$ is towards $\gamma'$. Hence $\gamma'$ is a saddle point and hence unstable. This scenario is illustrated in Figure 13(b).

**(1c):** If $\gamma' = \gamma^*$, then for any ${}^u\gamma' > \gamma'$, we have $\mathbb{P}_x\left(\phi(\mathbf{X}) \geqslant {}^u\gamma'\right) < \rho$ (again follows from the definition (1)) and hence $\mathbb{P}_x\left(\phi(\mathbf{X}) \geqslant {}^u\gamma'\right) - \rho < 0$. So at ${}^u\gamma'$, the direction of the drift of the DI is towards $\gamma'$. Now, choose ${}^l\gamma'$ from the left neighborhood of $\gamma'$, i.e., ${}^l\gamma' < \gamma'$. In this case, $\mathbb{P}_x\left(\phi(\mathbf{X}) \geqslant {}^l\gamma'\right) - \rho > 0$ (directly follows from the definition (1)). Hence, the direction of the DI drift at ${}^l\gamma'$ is also toward $\gamma'$. Hence, $\gamma' = \gamma^*$ is a stable equilibrium point of the DI. This scenario is illustrated in Figure 13(c).

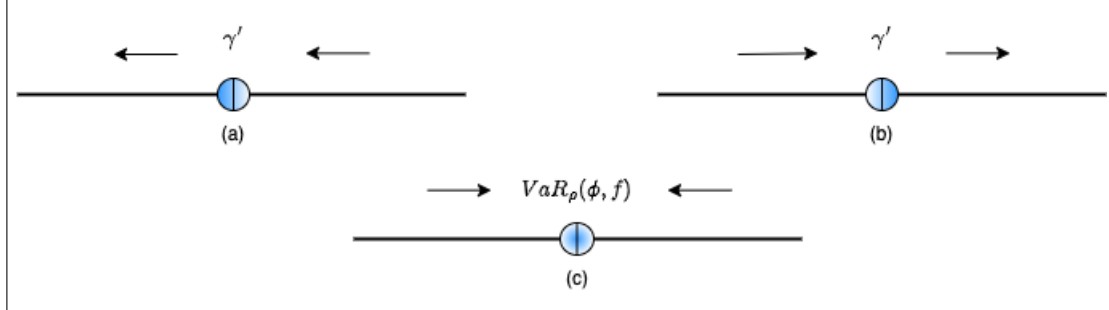

Figure 13: Nature of the stable points of the DI (84)

**Case 2:** Consider the set $Z = \{\gamma \in \mathbb{R} \mid \mathbb{P}_x(\phi(\mathbf{X}) = \gamma) = 0\}$. The set $Z$ has a non-zero probability since the performance function $\phi$ is non-constant. This case is illustrated in Case 2 of Figure 12. The only root of $h$ that belongs to the set $Z$ is $\gamma^*$. In fact, for $\gamma \in Z$, we have

$$h(\gamma) = 0$$
$$\Rightarrow -\mathbb{E}_{\mathbb{P}_x}\left[-(1-\rho)\mathbb{I}(\phi(\mathbf{X})^+, \gamma) + \rho\mathbb{I}(\phi(\mathbf{X}), \gamma)^-\right] = 0$$
$$\Rightarrow (1-\rho)\mathbb{P}_x\left(\phi(\mathbf{X}) \geqslant \gamma\right) - \rho\mathbb{P}_x\left(\phi(\mathbf{X}) \leqslant \gamma\right) = 0$$
$$\Rightarrow \mathbb{P}_x\left(\phi(\mathbf{X}) \geqslant \gamma\right) - \rho = 0$$
$$\Rightarrow \mathbb{P}_x\left(\phi(\mathbf{X}) \geqslant \gamma\right) = \rho$$
$$\Rightarrow \gamma = \gamma^*.$$

In this case also, one can perform a similar analysis as given in (1c) to show that $\gamma^*$ is indeed a stable equilibrium point of the DI (84). In addition, we define a Lyapunov function $V(\gamma) \triangleq \mathbb{E}_{\mathbb{P}_x}\left[\psi(\phi(\mathbf{X}), \gamma)\right] - \mathbb{E}_{\mathbb{P}_x}\left[\psi(\phi(\mathbf{X}), \gamma^*)\right]$. One can verify $V(\gamma) > 0$, $\forall \gamma \in \mathbb{R}\backslash\{\gamma^*\}$. (It follows since $\mathbb{E}_{\mathbb{P}_x}\left[\psi(\phi(\mathbf{X}), \gamma)\right]$ is a convex function and $\gamma^*$ is its global minimum). In addition, $V(\gamma^*) = 0$. Also, note that for $\gamma \in \mathbb{R}$, we have $yh(\gamma) \leqslant 0$, $\forall y \in \partial_\gamma V(\gamma)$. Therefore, $\gamma^*$ is the global attractor of the flow induced by the DI (84) (Benaïm et al., 2005).

A fortiori, by appealing to Corollary 4 in Chapter 5 of (Borkar, 2008), we obtain that the iterates $\{\gamma_t\}$ converge almost surely to $\gamma^* = \mathtt{VaR}_\rho(\mathbb{P}_x)$. This completes the proof of Theorem 2.

The convergence of $\eta_t$ and $\bar{\eta}_t$ is analogous to that provided in Lemma 6. $\qquad\square$

**Proof of Lemma 7:**

*Proof.* By the continuity of $C_\rho^{(3)}$ and $\Delta_t$ being Bernoulli random variable, we have by Taylor's theorem,

$$C_\rho(\omega_t + c_t\Delta_t) \approx C_\rho(\omega_t) + c_t\Delta_t^\top\nabla C_\rho(\omega_t) + \frac{c_t^2}{2!}\Delta_t^\top\nabla^2 C_\rho(\omega_t)\Delta_t + \frac{c_t^3}{3!}\nabla^3 C_\rho(\omega_t)\Delta_t \otimes \Delta_t \otimes \Delta_t,$$

where $\bar{\omega}_t$ lies on the line segment between $\omega_t$ and $\omega_t + c_t\Delta_t$. Hence,

$$\frac{C_\rho(\omega_t + c_t\Delta_t) - C_\rho(\omega_t - c_t\Delta_t)}{2c_t\Delta_t} = \frac{\Delta_t^\top}{\Delta_t}\nabla C_\rho(\omega_t) + \frac{c_t^2}{12\Delta_t}\nabla^3\left(C_\rho(\bar{\omega}_t) + C_\rho(\bar{\omega}_t')\right)\Delta_t \otimes \Delta_t \otimes \Delta_t$$

Let $b_{t_l}$ denotes the $l^{th}$ term of the bias vector $b_t$. Then

$$b_{t_l} = \mathbb{E}\left[\widehat{\nabla C}_\rho(\omega_t) - \nabla C_\rho(\omega_t) \mid \mathcal{F}_t\right]$$
$$= \mathbb{E}\left[\frac{\Delta_{t_l}}{\Delta_{t_l}}\nabla C_\rho(\omega_t) + \frac{c_t^2}{12\Delta_{t_l}}\nabla^3\left(C_\rho(\bar{\omega}_t) + C_\rho(\bar{\omega}_t')\right)\Delta_t \otimes \Delta_t \otimes \Delta_t - \nabla_t C_\rho(\omega_t)\Big|\mathcal{F}_t\right] \qquad (85)$$

where $\bar{\omega}'_t$ lies on the line segment between $\omega_t$ and $\omega_t - c_t \Delta_t$. Now note that,

$$\mathbb{E}\left[\Delta_t^{-1}\Delta_t^\top \nabla C_\rho(\omega_t) \mid \mathcal{F}_t\right] = (\nabla C_\rho(\omega_t))_1 \mathbb{E}\left[\Delta_t^{-1}\Delta_{t_1} \mid \mathcal{F}_t\right] + \cdots + (\nabla C_\rho(\omega_t))_p \mathbb{E}\left[\Delta_t^{-1}\Delta_{t_p} \mid \mathcal{F}_t\right]$$
$$\text{(Since, } \nabla C_\rho(\omega_t) \text{ is measurable w.r.t } \mathcal{F}_t)$$

$$= (\nabla C_\rho(\omega_t))_1 \mathbb{E}\left[\begin{pmatrix} 1 \\ \Delta_{t_2}^{-1}\Delta_{t_1} \\ \vdots \\ \Delta_{t_p}^{-1}\Delta_{t_1} \end{pmatrix} \mid \mathcal{F}_t\right] + \cdots + (\nabla C_\rho(\omega_t))_p \mathbb{E}\left[\begin{pmatrix} \Delta_{t_1}^{-1}\Delta_{t_p} \\ \Delta_{t_2}^{-1}\Delta_{t_p} \\ \vdots \\ 1 \end{pmatrix} \mid \mathcal{F}_t\right]$$

$$= (\nabla C_\rho(\omega_t))_1 \begin{pmatrix} 1 \\ \mathbb{E}\Delta_{t_2}^{-1}\mathbb{E}\Delta_{t_1} \\ \vdots \\ \mathbb{E}\Delta_{t_p}^{-1}\mathbb{E}\Delta_{t_1} \end{pmatrix} + \cdots + (\nabla C_\rho(\omega_t))_p \begin{pmatrix} \mathbb{E}\Delta_{t_1}^{-1}\mathbb{E}\Delta_{t_p} \\ \mathbb{E}\Delta_{t_2}^{-1}\mathbb{E}\Delta_{t_p} \\ \vdots \\ 1 \end{pmatrix}$$

$$\text{(Since, } \mathbb{E}\Delta_{t_i} = 0, \forall i \in [1\ldots p] \text{ and } \Delta_{t_i} \text{ is independent of } \Delta_{t_j} \forall i \neq j)$$

$$= (\nabla C_\rho(\omega_t))_1 \begin{pmatrix} 1 \\ 0 \\ \vdots \\ 0 \end{pmatrix} + \cdots + (\nabla C_\rho(\omega_t))_p \begin{pmatrix} 0 \\ 0 \\ \vdots \\ 1 \end{pmatrix} = \nabla C_\rho(\omega_t) \tag{86}$$

Therefore, from eq.85 and eq.86 we get,

$$b_{t_l} = \frac{1}{12}\mathbb{E}\left[\frac{1}{\Delta_{t_l}}\left(\nabla^3 C_\rho(\bar{\omega}_t) + \nabla^3 C_\rho(\bar{\omega}_t')\right)\bar{\Delta}_t \otimes \bar{\Delta}_t \otimes \bar{\Delta}_t \mid \mathcal{F}_t\right] \tag{87}$$

We can bound the term on the right-hand side of eq. 87 in magnitude as follows

$$b_t = \frac{1}{12}\mathbb{E}\left[\frac{1}{\Delta_{t_l}}\left(\nabla^3 C_\rho(\bar{\omega}_t) + \nabla^3 C_\rho(\bar{\omega}_t')\right)\bar{\Delta}_t \otimes \bar{\Delta}_t \otimes \bar{\Delta}_t \mid \mathcal{F}_t\right] \leqslant \frac{\alpha c_t^2}{6}\sum_{i_1}\sum_{i_2}\sum_{i_3}\mathbb{E}\left[\frac{\Delta_{t_{i_1}}\Delta_{t_{i_2}}\Delta_{t_{i_3}}}{\Delta_{k_l}}\right]$$
$$\leqslant \frac{p^3 \alpha c_t^2}{6} = \mathcal{O}(c_t^2),$$

The first inequality follows as $\nabla^3 C_\rho(\bar{\omega}) \leqslant \alpha, \forall \omega$ and the latter inequality follows since $\frac{\Delta_{t_{i_1}}\Delta_{t_{i_2}}\Delta_{t_{i_3}}}{\Delta_{t_l}} \leqslant 1$. $\quad\square$

**Proof of Theorem 3:**

*Proof.* Consider the recursion from Step 12 of the algorithm:

$$\omega_{t+1} = \omega_t + a_t\left(\underbrace{\frac{1}{2c_t\Delta_t}\left(\bar{\eta}^+_{t,N_t} - \bar{\eta}^-_{t,N_t}\right) - \widehat{\nabla C}_\rho(\omega_t)}_{p_t} + \underbrace{\mathbb{E}\left[\widehat{\nabla C}_\rho(\omega_t) - \nabla C_\rho(\omega_t) \mid \mathcal{F}_t\right]}_{b_t} - \right.$$
$$\left. \mathbb{E}\left[\widehat{\nabla C}_\rho(\omega_t) - \nabla C_\rho(\omega_t) \mid \mathcal{F}_t\right] + \widehat{\nabla C}_\rho(\omega_t)\right)$$
$$= \omega_t + a_t\left(p_t + b_t + \nabla C_\rho(\omega_t) + \widehat{\nabla C}_\rho(\omega_t) - \mathbb{E}\left[\widehat{\nabla C}_\rho(\omega_t) \mid \mathcal{F}_t\right]\right)$$
$$= \omega_t + a_t\left(p_t + b_t + e_t + \nabla C_\rho(\omega_t)\right), \text{ where } e_t = \widehat{\nabla C}_\rho(\omega_t) - \mathbb{E}\left[\widehat{\nabla C}_\rho(\omega_t) \mid \mathcal{F}_t\right]. \tag{88}$$

Define

$$\xi_{t+1} = \sum_{i=0}^{t} a_t e_t, t \geqslant 0. \tag{89}$$

Then

$$\mathbb{E}\left[\xi_{t+1}|\mathcal{F}_t\right] = \mathbb{E}\left[\sum_{i=0}^{t-1} a_t e_t \Big| \mathcal{F}_t\right] = \sum_{i=0}^{t} a_t \mathbb{E}\left[e_t|\mathcal{F}_t\right] + a_t \left(\mathbb{E}\left[\widehat{\nabla C}_\rho(\omega_t)|\mathcal{F}_t\right] - \mathbb{E}\left[\widehat{\nabla C}_\rho(\omega_t) \mid \mathcal{F}_t\right]\right) = \xi_t. \quad (90)$$

This implies that $\{\xi_t\}$ is a martingale with respect to filtration $\{\mathcal{F}_t\}$. Also, since $C_\rho$ is continously differentiable, we have $\xi_t$ is square-integrable, $\forall t$, $i.e$, $\mathbb{E}\left[\|\xi_t\|^2\right] < \infty$, $\forall t$. Again, by the continuous differentiability of $C_\rho$, we obtain

$$\sum_t \mathbb{E}\left[\|\xi_{t+1} - \xi_t\|^2|\mathcal{F}_t\right] = \sum_t a_t^2 \mathbb{E}\left[\|e_t\|^2\right] < \infty \text{ on the set } \{\sup_t \|\omega_t\| < \infty\}. \quad (91)$$

Therefore, by Martingale convergence theorem (Yeh, 1995), we get

$$\lim_{t\to\infty} \xi_t \text{ exists on the event } \{\sup_t \|\omega_t\| < \infty\}. \quad (92)$$

Hence, by Corollary 3, Chapter 2 and Theorem 7, Chapter 8 of (Borkar, 2008) along with the facts that $N_t = O(\lceil (t+1)^{c/2} \rceil)$, $p_t \to 0$ (from Lemma 7) and $b_t \to 0$ (from Theorem 2) as $t \to \infty$, the asymptotic behavior of the sample paths belonging to the event $\{\sup_t \|\omega_t\| < \infty\}$ is equivalent to the long-term behavior of the dynamical system induced by the ODE

$$\frac{d\omega(t)}{dt} = \nabla C_\rho(\omega(t)), t \geqslant 0. \quad (93)$$

This further implies that the iterates $\omega_t$ corresponding to the sample paths belonging to the event $\{\sup_t \|\omega_t\| < \infty\}$ converge to any of the compact transitive invariant sets connected internally in chains of (93). Invariant sets are subsets of the state space that remain unchanged under the flow of the dynamical system. The dynamical system (93) driven by the gradient of the CVaR is a gradient flow where the only possible invariant sets are the subsets of $H = \{\omega|\nabla\texttt{CVaR}_\rho(\omega) = 0\}$ (Lemma 1, Section 10.2 of (Borkar, 2008)). Further, by invoking the LaSalle invariance principle and the Lyapunov theorem, one can obtain that the asymptotically stable points inside $H$ are given by $\{\omega \in H|\nabla^2\texttt{CVaR}_\rho(\omega) \preccurlyeq 0\}$. $\qquad\square$

**Proof of Theorem 4:**

*Proof.* One can follow the multi-timescale approach as shown in proofs of Lemma 6 and Theorem 2. Here, the stochastic recursion governing $\kappa_t$ introduces an additional layer of complexity. However, this recursion does not complicate the analysis significantly because $\kappa_t$ evolves at the same timescale as $\gamma_t$, as they share a unilateral dependency. Specifically, the evolution of $\gamma_t$ and $\eta_t$ depends on the values of $\kappa_t$, which determines the samples and the sampling ratio at each time step. This unilateral dependence means that while $\gamma_t$ and $\eta_t$ influence the behavior of $\kappa_t$, the reverse is not true—$\kappa_t$ is not directly influenced by $\gamma_t$ or $\eta_t$. As a result, $\kappa_t$ can be analyzed independently of the other variables. One can easily show that the recursion 49 asymptotically behaves like the gradient flow. The analysis of the remaining stochastic recursions of the theorem follow similarly to that of Lemma 6 and Theorem 2. $\qquad\square$

