# OpenReview forum: "Extreme Risk Measures: Estimation and Optimization via Stochastic Approximation"
_TMLR — Rejected by TMLR_

### Review · Reviewer_mNAP · 2024-10-09

**Summary Of Contributions:**

This work presents a stochastic approximation algorithm for estimating and optimizing the quantile/superquantile with convergence analysis and applies it to some practical examples.

**Audience:**

Yes

**Claims And Evidence:**

No

**Requested Changes:**

Some of these are questions.

Major:
1. Sometimes the set {$...$} only contains condition, please be crystal clear about the definition. (e.g. {$x\in X: ...$})
2. Figure 1 and the explanation are a bit misleading; the issue is its large variance as mentioned at the bottom of page 6.  But the way it explains would mislead the readers in the way that the estimate cannot be close to the true quantile; can you change the explanation so that it is clearer that the variance is the issue?
3. The definition of Q_t is very unclear; is it just a truncated measure?  Or is it a probability measure?  Can you define it explicitly?
4. Relatedly, whenever you define Ranon-Nikodym derivative, please be sure that it can be well-defined on an explicitly specified region.  In some part, it is explicitly mentioned, but it is unclear in other parts.
5. Also equation 9 and the sentence above may not be equivalent requirement; (9) would not be sufficient for the Radon-Nikodym derivative to exist.
6. Section 3.1 logical flow is unclear; please be crystal clear what you want to show at the end, and how you show it before delving into each mathematical step.  It seems there are some assumptions involved.  Please be crystal clear what you assume as well before delving into the details.
7. equation 32; why can you apply Lemma 3?  continuity assumption holds here?
8. eq 36; please say it is from equation 17.  There are some parts that do not say why the equality, inequality can be derived while there are some explanations for very easy parts.  Please have a balanced explanation for all the transformation.
9. page 14, from “Further, we assume” there are many typos and wrong expressions. Please modify them.  Also, for all x.
10. Eq 39 is this correct?  Q_{t+1}(…) = 1?  it is because of the vague definition of Q_t
11. 3.2 Assumptions and Theorem are mentioned before they appear; it is hard to follow.




Minor:
1. The work of Sony AI Gran Turismo is cited as a work of risk sensitive sequential decision making; it is to me just a distributional RL which is not used for “risk sensitive” application but for performance improvement at least in that work.
2. 1.3 absolute continuous definition is strange $P(A)=0...$ A should be P/Q.
3. mean → the mean
4. The sentence at the bottom of page 5 is unclear; why $P_x$ is important for talking about the differentiability of $\psi$?
5. page 6   note that … is bounded → over which interval?
6. Proposition 2 proof first equation → typo
7. Proof of Proposition 2; why can you remove G at the last equality?
8. page 10 rule 21 → 22?
9. Proof of Lemma4 there are some typos.
10. Lemma 5 : need to make Q mathecal
11. What is the definition of CONV?
12. page 15 a recursions ?
13. page 16 one can estimate … can be estimated
14. page 17 Step 8??
15. in reference   hjb, gaussian etc. must be capital letter

**Strengths And Weaknesses:**

Strength:
If some parts get clarified, this paper seems to be a solid contribution to quantile estimation with sufficient application examples.

Weakness:
1. Some of the math contain possible errors or are unclear, which makes it hard to completely verify the claim.
2. There seems to be a logical leap or insufficient explanation in some part of the paper.

The current form is not sufficient to judge if this is entirely correct; so please carefully modify the manuscript to be crystal clear about some steps and flows.
Overall attempts and analysis/experiments should be very solid once those get clarified.

---

> ### Author Response · Authors · 2024-11-21
> **Technical Clarifications, Analytical Refinements, and Methodological Enhancements**
>
> We thank the reviewers for their valuable feedback, significantly improving our manuscript. We clarified key definitions, refined the logical flow, and balanced derivation explanations. All typographical, grammatical, and notational inconsistencies were corrected. References were also updated ensuring enhanced clarity.
>
> ### Major
> 1. *Sometimes the set `{...}` only contains a condition; please be crystal clear about the definition (e.g., `{x \in X: ...}`).*
> **Response:** We are unable to figure out the comment. Kindly provide more details.
>
> 2. *Figure 1 and  ... clearer that the variance is the issue?*
>    **Response:** Fixed in the updated manuscript.
>
> 3. *The definition of \(Q_t\) is very unclear ... define it explicitly?*
>    **Response:** \(Q_t\) represents the zero-variance distribution. As you correctly noted, it corresponds to the truncated distribution. However, sampling from this distribution becomes increasingly difficult as \(t\) grows, since the event \(\{\phi(X) \geq \gamma_t\}\) becomes less likely. To address this, we maintain a surrogate distribution \(Q_t\) (which is further approximated by \(Q_{\theta_t}\)) that gradually evolves to capture the rare region \(\{\phi(X) \geq \gamma_t\}\).
>
> 4. *Relatedly, whenever you  ... unclear in other parts.*
>    **Response:** For the Radon-Nikodym derivative to exist, one needs to satisfy absolute continuity as mentioned in Equation (11). Since we are considering NEF (natural exponential family), which has its support over the entire Euclidean space, the condition is trivially satisfied.
>
> 5. *Also, Equation 9 ... to exist.*
>    **Response:** Equation (9) is the absolute continuity definition. This implies that the original distribution \(\mathbb{P}_x\) is absolutely continuous with respect to the surrogate \(Q_t\). This is sufficient for the Radon-Nikodym derivative to exist.
>
> 6. *Section 3.1 logical flow is unclear ... into the details.*
>    **Response:** Updated the manuscript. Please refer to paragraph 1 of Section 3.1.
>
> 7. *Equation 32; why can you ... hold here?*
>    **Response:** Added the required assumptions in the statements of Lemma 5 and Theorem 1. Please refer to the updated manuscript.
>
> 8. *Equation 36; please say ... transformations.*
>    **Response:** Resolved in the updated manuscript.
>
> 9. *Page 14: from "Further, we assume," there are ... for all \(x\).*
>    **Response:** Resolved in the updated manuscript. Please refer to page 14.
>
> 10. *Equation 39: Is this correct? ... of \(Q_t\).*
>     **Response:** Yes, correct.
>
> 11. *Section 3.2: Assumptions ... hard to follow.*
>     **Response:** Fixed. Please see the updated Section 3.2.
>
> ### Minor Comments
>
> 1. *The work of Sony AI Gran Turismo ... in that work.*
>    **Response:** You are indeed right. The reference has been replaced with a more fundamental reference that explores the risk-sensitive sequential decision-making paradigm.
>
> 2. *1.3 absolute continuous ... \(\mathrm{P} / \mathrm{Q}\).*
>    **Response:** The definition provided is the standard definition of absolute continuity.
>
> 3. *mean \(\rightarrow\) the mean*
>    **Response:** Fixed in the updated manuscript.
>
> 4. *The sentence at the bottom of page 5 ... differentiability of \(\psi\)?*
>    **Response:** You are correct. We have now removed it. We thank the reviewer for the comment.
>
> 5. *Page 6 note that ... is bounded \(\rightarrow\) over which interval?*
>    **Response:** From the definition of \(\partial_\gamma \psi(\phi(x), \gamma)\) given by Eq. (6), it follows that \(|\partial_\gamma \psi(\phi(x), \gamma)|\) is upper bounded by 1.
>
> 6. *Proposition 2 proof first equation \(\rightarrow\) typo*
>    **Response:** Fixed in the updated manuscript.
>
> 7. *Proof of Proposition 2: Why ... equality?*
>    **Response:** There was an error in Proposition 2 which contained \(G\) on the integral, which it should not. As in the region \(G^C\), the integral is zero; it follows that the integral is only over \(G\), which is dropped.
>
> 8. *Page 10 rule \(21 \rightarrow 22\)?*
>    **Response:** The update rule given is correct as we are referring to the update of the surrogate distribution parameters \(\theta_{t+1}\) and not the update of the \(\gamma_{t+1}\) estimate.
>
> 9. *Proof of Lemma 4: There are some typos.*
>    **Response:** Fixed in the updated manuscript.
>
> 10. *Lemma 5: Need to make \(Q\) mathematical.*
>     **Response:** Fixed in the updated manuscript.
>
> 11. *What is the definition of CONV?*
>     **Response:** We have defined in the updated manuscript. Please refer to Section 1.4, *Summary of Notation*.
>
> 12. *Page 15: "a recursions?"*
>     **Response:** Fixed in the updated manuscript.
>
> 13. *Page 16: "one can estimate ... can be estimated."*
>     **Response:** Fixed in the updated manuscript.
>
> 14. *Page 17: "Step 8??"*
>     **Response:** Fixed in the updated manuscript.
>
> 15. *In references: HJB, Gaussian, etc., must be in capital letters.*
>     **Response:** Fixed in the updated manuscript.

---

> ### Comment · Reviewer_mNAP · 2024-11-25
> **Thank you for the response**
>
> Thank you for the response; I acknowledge the receipt of the response.
> I will post any additional question etc. if there is any, but for now I have the followings:
>
> 1. I understand that sometimes people write {${\rm condition~of} ~x$} in statistics etc. but please clarify where x belongs to for the cases where it is not completely obvious ({$x\in\mathbb{R}^n: {\rm condition ~of } ~x$}).
>
> 2. $P\ll Q$ should be the condition "$\forall A: Q(A)=0\Rightarrow P(A)=0$", but not "$P(A)=0\Rightarrow Q(A)=0$".  This relates to some of my major and minor concerns as well.

---

> > ### Author Response · Authors · 2024-11-26
> >
> > 1. *I understand that sometimes people write `{condition of x}` in statistics, etc., but please clarify where x belongs to for the cases where it is not completely obvious (`{x ∈ ℝⁿ : condition of x}`).*
> >    **Response:** Thank you for the clarification. We have updated our notation to ensure clarity. Please refer to Section 1.4 for the revised details.
> >
> > 2. *`P ≪ Q` should be the condition "`∀A: Q(A) = 0 ⇒ P(A) = 0`", but not "`P(A) = 0 ⇒ Q(A) = 0`". This relates to some of my major and minor concerns as well.*
> >    **Response:** Thank you for pointing this out. We have corrected this in the updated manuscript. Please refer to Section 1.4.

---

> > > ### Comment · Reviewer_mNAP · 2024-11-26
> > > **Thank you**
> > >
> > > Thank you for the response;
> > > For 2. is Radon-Nikodym derivative of $P_x$ w.r.t. the surrogate distribution $Q_t$ well defined (page 8 top)?  Is it from (10)?   I remember there are also other places where it is unclear how the Radon-Nikodym derivative is guaranteed to exist.

---

> > > > ### Author Response · Authors · 2024-11-29
> > > >
> > > > Thank you for your comment.
> > > >
> > > > *For 2. is Radon-Nikodym derivative of \\(P_x\\) w.r.t. the surrogate distribution \\(Q_t\\) well defined (page 8 top)? Is it from (10)? I remember there are also other places where it is unclear how the Radon-Nikodym derivative is guaranteed to exist.*
> > > >
> > > > **Response:** You are correct; it does come from Eq. 10. We have updated the equations in our revised manuscript as
> > > > (the indexing from \\(t+1\\) to \\(t\\))
> > > >
> > > >
> > > > \\(
> > > > \mathbb{P}_x\left(\{\phi(\mathbf{X}) \geq \gamma_t\} \cap B\right) = 0 \implies Q_t\left(\{\phi(\mathbf{X}) \geq \gamma_t\} \cap B\right) = 0, \, \forall B \text{ (Borel set)}.
> > > > \\)
> > > >
> > > > \\(
> > > > \frac{d\mathbb{P}_x}{d\nu}(x) = 0 \implies \frac{dQ_t}{d\nu}(x) = 0, \forall x \in \{\phi(x) \geq \gamma_t\} \, \text{almost everywhere}
> > > > \\)
> > > >
> > > > Additionally, in the introduction, we have updated the \\(P_x\\) definition as
> > > >
> > > > "Let \\(P_{x}\\) be the probability measure of \\(\mathbf{X}\\) which is absolutely continuous w.r.t Lebesgue measure \\(\nu\\)."
> > > >
> > > > Because of the above assumptions, all the Radon-Nikodym derivates in the paper are valid. This update mentioned above is not provided in the present revision as it is under review by the other reviewers, and we shall update it as soon as we receive the other reviews.

---

> ### Comment · Reviewer_mNAP · 2024-11-29
> **Thank you for the response**
>
> Thank you for the response;
> again, $P(A)=0\Rightarrow Q(A)=0$ should be $Q(A)=0\Rightarrow P(A)=0$ isn't it?

---

> ### Author Response · Authors · 2024-11-29
> **Thank you for the clarification again**
>
> ***Response:*** We have made the correction in the updated manuscript, the changes are given below:
>
> <Page 7>
> In our approach, at each iteration $t$, the sample  $\mathbf{X}_{t+1}$ is drawn using a surrogate measure  $Q_t$, which is absolutely continuous with respect to the Lebesgue measure $\nu$ and may differ from the given measure $\mathbb{P}_x$.
>
> To identify an appropriate surrogate measure, it suffices to restrict the search to the subspace of measures for which $\mathbb{P}_x$, when truncated to the region $\\{\phi(\mathbf{X}) \geq \gamma_t\\}$, is absolutely continuous with respect to the surrogate measure. In other words, any Borel set within the region $\\{\phi(\mathbf{X}) \geq \gamma_t\\}$ that has zero measure under $Q_t$ must also have zero measure under $\mathbb{P}_x$.
>
> $
> Q_{t}(\{\phi(\mathbf{X}) \geq \gamma_{t}\} \cap B) = 0 \Rightarrow P_{x}(\{\phi(\mathbf{X}) \geq \gamma_{t}\} \cap B) = 0, \forall B: \texttt{Borel set}.
> $
> This implies that
> $
> \frac{dQ_{t}}{d\nu}(x) = 0  \Rightarrow  \frac{dP_{x}}{d\nu}(x) = 0,  \forall x \in \{\phi(x) \geq \gamma_{t}\} \texttt{ almost everywhere}
> \Rightarrow  \\{\frac{dQ_{t}}{d\nu}(x)\mathbb{I}(\mathcal{\phi}(x),\gamma_{t})^{+} \neq 0\\} \supseteq \\{\frac{dP}{d\nu}(x)\mathbb{I}(\mathcal{\phi}(x),\gamma_{t})^{+} \neq 0\\}  \texttt{ almost everywhere}.
> $

---

> > ### Comment · Reviewer_dLZW · 2024-12-05
> >
> > This is from the authors' response above:
> >
> > > To identify an appropriate surrogate measure, it suffices to restrict the search to the subspace of measures for which $P_x$, when truncated to the region ${\phi(X) \geq \gamma\}$, is **absolutely continuous with respect to the surrogate measure**. In other words, any Borel set within the region  ${\phi(X) \geq \gamma\}$ that has zero measure under $Q_t$ must also have zero measure under $P_x$.
> >
> > And this is from the latest version of the paper
> >
> > > To find such a suitable surrogate measure, it is sufficient to seek within the subspace of measures that are **absolutely continuous with respect to the original measure** $P_x$ truncated to the region ${\phi(X) \geq \gamma\}$, i.e., for every Borel set inside the region  ${\phi(X) \geq \gamma\}$ which have a zero measure w.r.t. $P_x$ must have a zero measure w.r.t. $Q_{t+1}$ as well.
> >
> > I share Reviewer mNAP's confusion as to what is absolutely continuous with respect to what.

---

> > > ### Author Response · Authors · 2024-12-14
> > > **Response - 2**
> > >
> > > There was an error in the earlier version of the paper on Page 7. Now we have corrected it. Please see the highlighted (red) text in Page 7 of the latest version of the manuscript

---

> > > > ### Comment · Reviewer_mNAP · 2024-12-17
> > > > **Thank you**
> > > >
> > > > Thank you for your update; while I can see that you have modified, updated, and clarified some parts, due to the certain amount of errors and update which somehow require re-verifying the contents, I wouldn't recommend acceptance for now.  It makes sense to ask authors to revise and check the entire draft thoroughly again before possible resubmission in the future.

---

> ### Author Response · Authors · 2024-12-17
> **Response**
>
> Dear Reviewer
>
> Kindly let us know the error as we have combed through it multiple times. The absolute continuity error was a small error which we had missed. We have now fixed it as well.

---

### Review · Reviewer_dLZW · 2024-10-14

**Summary Of Contributions:**

Given a random variable $X \sim \mathbb{P}$, a return function $\phi$ and a threshold $\rho \in [0, 1]$, this paper is concerned with estimating two central risk measures:
- the Value-at-Risk (or quantile) $\mathrm{VaR} = \sup \\{\gamma : \mathbb{P}(\phi(X) \geq \gamma) \geq \rho \\}$
- the Conditional Value-at-Risk (or superquantile) $\mathrm{CVaR} = \mathbb{E} [\phi(X) \vert \phi(X) \geq \mathrm{VaR} ]$

The authors introduce an online algorithm for estimating $\mathrm{VaR}$ and $\mathrm{CVaR}$ and deduce a method to optimize $\mathrm{CVaR}$. Their algorithm can be executed incrementally on large datasets, since it is based on stochastic approximation. It leverages adaptive importance sampling (IS) to reduce variance for extreme values of the threshold $\rho$. The use of multiple timescales is presented as an answer to intractable integrals and latent probability distributions $\mathbb{P}$.

A theoretical convergence analysis is provided for the estimation and optimization procedures. Numerical experiments demonstrate how the new algorithm behaves on various practical problems where accurate risk estimation is paramount.

**Audience:**

Yes

**Broader Impact Concerns:**

The experiments section presents high-stakes applications in the realm of finance or health. So far, the intricate presentation of the algorithm does not make it transparent and trustworthy enough to claim that such applications can be realized with confidence.

**Claims And Evidence:**

No

**Requested Changes:**

## Critical changes

**Form**

- Keep the body of the paper to a dozen pages or so, and move the bulk of the mathematical details to an appendix.
- Give a high-level overview of the procedure early on, before explaining each individual step.
- Add labels to figure axes and more detailed descriptions of what they represent.
- Complement the code with a brief README to explain how your experiments can be reproduced.

**Content**

- Answer the questions listed in the "Weaknesses" part of the review.
- Explain which parts of your algorithm are new and which parts are already known. In particular, compare to He et al. (2023), which seems to be the work closest to your own, and discuss your use of stochastic approximation.
- In Theorems 2, 3, 4, give more details as to:
  - Why $\gamma_t$ converges to the right value but $\eta_t$ and $\bar{\eta}_t$ do not
  - How the choice of exponential family impacts the variational inference error

## Optional changes

**Presentation**

- Put citations in parentheses / square brackets with `\parencite` or `\citep`
- Avoid referring to theorems or propositions that have not been given yet, or add a comment like "as we will see in ...".
- Perhaps notations would be lighter if every measure was characterized using its density wrt Lebesgue? Would that cause any theoretical issues with e.g. Radon-Nikodym derivatives?

**Clarifications**

> P2: The superquantile is also more tractable in optimization contexts owing to its desirable properties such as coherency with respect to return distribution

What does coherency mean here?

> P3: Variance reduction techniques, particularly importance sampling, have been utilized to manage the large variance in extreme risk estimation.

Define "IS" acronym here.

> P5: Proposition 1

The proof can be shortened by just using the convexity of ReLU.

> P7: Equation 11

Is the set inclusion sign in the wrong direction?

> P7: before Equation 12

Where does the derivative $d \mathbb{P} / d Q_t$ come from? Also, I think that from this point on there is a time index confusion between $Q_t$ and $Q_{t+1}$ (isn't $X_{t+1}$ drawn from $Q_{t+1}$?).

> P8: Equation 14

$\ell_{t+1}$ has a simpler expression / interpretation: the true probability to exceed $\gamma_t$

> P10: $K(\theta)$ is the cumulant function, which ensures that the distribution is normalized

Is $m(\theta)$ equal to the gradient of the cumulant? I think you use this fact in a later proof.

> P13: Theorem 1

What do $\mathcal{Q}$ and $\mathcal{C}$ correspond to intuitively, for someone who might have skipped the proofs?

> P15: one might fall prey to over-compliance i.e., scenarios where a substantial fraction of the samples belong to the region $\\{\phi(X) \geq \gamma_t\\}$.

Why is that bad? Isn't that precisely what we want?

> P15: we obtain the sampling measure $\hat{Q}_t$

I would denote it as $\hat{Q}_{\theta_t}$.

> P15: with parameter $λ \in [0, 1]$ (fixed a priori)

How does one choose it?

> P25: CVaR optimization involves adjusting parameters to minimize the expected losses in extreme scenarios.

Then why is Eq 76 a maximization?

> P29: One of the challenges is that the true underlying probability distribution $\mathbb{P}_x$ that governs the rewards is not explicitly known. Instead, one has access to an oracle which can provide samples or realizations from this distribution $\mathbb{P}_x$.

Does that limit the number of time steps $t$ to the number of drawn samples?

> P30: Note that we follow the same timescale as that of $\gamma_t$ since there exists a unilateral coupling between $\kappa_t$ and other iterates.

Why is this justified?

> P31: we compare the values of VaR and CVaR generated through Monte-Carlo (MC) estimates against the values generated by our procedure

What is your stopping criterion and hyperparameter setup?

> P33: We choose a period of about a decade from 2015 - 2023 and solve the optimization problem in eq. 90 using CVXPY Diamond & Boyd (2016) and MOSEK ApS (2024) solver.

Can you adapt your CVaR minimization algorithm to a constrained setting? If I understand correctly, here you estimate first and then optimize second?

> P33: we also verify by the efficient frontier analysis

What does this entail?

> P36: The results demonstrate that the agent maintains stability despite increasing levels of uncertainty associated with decreasing $\rho$.

How do we deduce this from the curves?

> P37: What is the difference between the values and the rewards?

**Typos**

> P1: This paper introduces an incremental, single-pass, and adaptive variance reduction ~techniqueS~ technique to estimate extreme VaR and CVaR

> P2: The utility value of the extreme risk measures is ~considerably~ significant

> P4: without ~incuring~ incurring heavy computational cost.

> P4: $\mathbb{P}(A) = 0 \implies \mathbb{P}(A) = 0$

Should be $\mathbb{Q}(A)$

> P7: by sampling $X_{t+1}$ using the surrogate PDF $w_{t+1}$

Should be $Q_{t+1}$

> P9: $\theta_t = ...$

Should be $\theta_{t+1}$

> P10: The crucial question to address is whether the application of the aforementioned update rule ~(21)~ (22)

> P12: We define mollifier

> P12: continuous and ~Reimann~ Riemann integrable

> P16: Therefore, one can estimate $\mathrm{CVaR}$ ~can be estimated~ as follows:

> P26: which is estimated using the ~randomnly~ randomly perturbed

> P31: primarily we see how our algorithm ~fairs~ fares with actual returns

> P29: moment projection into the ~parameterize~ parameterized probability measure

**Strengths And Weaknesses:**

Please note that I am not an expert on risk measures, nor am I familiar with the related literature. This paper was very interesting for me to read and I reviewed it to the best of my ability, but I skipped most of the proofs so I cannot speak to their correctness.

## Strengths

This paper combines a lot of different ideas into a single algorithmic procedure. Here is a summary of the main ones, as I understood them:

1. Reformulate the $\mathrm{VaR}$ definition as a convex optimization problem with variable $\gamma$ (Eq 5). That reformulation was known.
2. Use stochastic approximation to iteratively improve an estimate $\gamma_t$ of the optimum, based on samples $X_t \sim \mathbb{P}$ (Eq 7, 8).
3. To drive $X_{t+1}$ towards the region of interest $\\{ \phi(X) \geq \gamma_t \\}$, describe an ideal surrogate measure $Q_{t+1}$ that can be used for IS (Eq 13).
4. Exploit variational inference to project this ideal surrogate $Q_{t+1}$ onto a parametric measure $Q_{\theta_{t+1}}$ (Eq 18).
5. Pick $Q_\theta$ among the Natural Exponential Family (NEF) to simplify the relation between moments and parameters (Eq 20, 22).
6. Add a second layer of stochastic approximation based on samples $Y_t \sim Q_{\theta_t}$ to avoid computing intractable moment integrals (Eq 40).
7. Let the two stochastic approximation recursions evolve on different time scales to guarantee convergence.
8. For $\mathrm{CVaR}$ optimization, use all previous steps to obtain randomly perturbed finite difference gradients (Eq 77).
9. When the distribution $\mathbb{P}$ is latent, replace it with a variational estimate thanks to a third layer of stochastic approximation.

In addition, the authors study the convergence behavior of their algorithm with great care. Theorems 2 and 3 are especially impressive, because they precisely characterize the limits of the estimates and relate those to the variational approximation of the surrogate measure.

## Weaknesses

**Clarity**

The main weakness of the paper is its obscure exposition. In some cases, interlacing proofs with the main text is a good idea, but here I found it to be very distracting. I struggled to follow the flow of ideas, and the many technical details made it harder to see the big picture.

**Algorithm**

Perhaps as a result of this convoluted presentations, I also have doubts about the validity / relevance of a few mathematical aspects. I list my main questions below.

> P1: a single-pass, incremental, adaptive, stable algorithm that could efficiently process large datasets in the order of quadratic complexity per iteration with respect to the dimension of the input space.

How do you obtain the iteration complexity?

> P7: it is sufficient to seek within the subspace of measures that are absolutely continuous with respect to the original measure $\mathbb{P}_x$ truncated to the region $\\{\phi(X) \geq \gamma_t\\}$

In the rest of the article it remains unclear to me whether the whole support of $Q_{t+1}$ is included in $\\{\phi(X) \geq \gamma_t\\}$ or not. I would assume that it is required, but Equation 9 does not imply it (one would have to remove the intersection on the right-hand side).

> P7: As mentioned earlier, our goal is to find the surrogate measure $Q_t$ such that the event $\\{\phi(X_{t+1}) \geq \gamma_t\\}$ is more likely. Hence, we seek the optimum surrogate measure that minimizes the variance of the random variable $\zeta_t(X_{t+1}) I^+(\phi(X_{t+1}, \gamma_t)$

The conclusion that the surrogate measure needs to minimize the variance is not obvious to me. If the surrogate measure needs to be proportional to $\mathbb{P}$ on the extreme region $\\{\phi(X_{t+1}) \geq \gamma_t\\}$ and be zero elsewhere, can we just define it as "$\mathbb{P}$ restricted to that region"? That would probably make a lot of things clearer.

> P10: From the inverse function theorem, it follows that m is also invertible.

How do you compute that inverse in the non-Gaussian case?

> P11: The above inequality implies that the update rule (19) will monotonically improve the concentration of the probability measure of the surrogate measure on the event $\\{\phi(X) \geq \gamma_t\\}$.

This is one of my most confusing questions, sorry in advance. You prove that $Q_{\theta_{t+1}}$ puts more weight on the extreme event $\\{\phi(X) \geq \gamma_t\\}$ than $\mathbb{P}$, but when you say "monotonically" I'm more curious as to what happens with respect to the previous time step. Do we have monotonicity in that sense too, e.g. compared to the previous measure $Q_{\theta_{t}}$ on $\\{\phi(X) \geq \gamma_{t-1}\\}$, and would it even make sense to ask this?

> P14: Proposition 3

How reasonable is the hypothesis $1-\delta \leq ... \leq 1+\delta$? After all it involves $Q_t$, which we do not know.

> P15: Equation 40

How does a single sample from $Q_{\theta_t}$ allow approximation of the quotient of two integrals?

> P15: We prove in Lemma 6 that iterates $\eta_t$ indeed track the ideal $m(\theta)$ and $m^{-1}$ in the most cases are $O(1)$ computable.

The text of Lemma 6 doesn't make it obvious what you mean by $O(1)$ or by "most cases". Can you elaborate?

> P16: The Gaussian (or normal) distribution is in fact one of the most common among the NEF family.

Why would one choose a Gaussian (which is unbounded and symmetric) to approximate a truncated distribution?

> P26: the CVaR which is estimated using the randomnly perturbed finite difference method

Is it possible to use automatic differentiation instead, possibly with a stochastic computational graph due to your use of random samples? See the foundational paper <http://arxiv.org/abs/1506.05254> and the ones that cite it.

> As the number of perturbations increases, the average gradient estimate approaches the true gradient asymptotically, meaning that with enough samples, the variance in the gradient estimate becomes negligible and the estimate becomes increasingly accurate.

For finite differences, the number of function evaluations needed to obtain the exact gradient scales with the input dimension. Does that make accurate estimation intractable with your finite difference scheme when the dimension grows large?

> P28: Theorem 3. Assume that $a_t = o(\alpha_t)$.

Does that turn your 2-timescale algorithm into a 3-timescale algorithm? If so, will it ever converge?

**Experiments**

The experiments seem controversial to me, especially those in section 6.1. If I understand correctly, the baseline is standard Monte-Carlo estimation of the (super)quantile at threshold $\rho = 10^{-6}$, based on $10^6$ samples. This yields a large variance and makes MC estimates unreliable. Thus, another baseline is needed to determine the "true" quantiles (maybe numerical quadrature or exact formulas based on the PDF?). Otherwise you cannot make claims such as

> P31: We observe that EX outperforms MC in scenarios where the tail is particularly heavy, as evidenced by the more accurate CVaR estimates for the Log-normal, GEV and Student’s t distributions.

Furthermore, the estimation procedures (both MC and the new one) need to be run several times and variances need to be reported.
Finally, there should be a comparison to another method that is not plain MC, for example a less sophisticated form of importance sampling. Without a nontrivial alternative to benchmark against, the following claim is not substantiated:

> P39: Theoretical and empirical analyses validate the effectiveness of these approaches.

As for the experiments on actual data, they need more detailed explanations if they are to be understood by readers unfamiliar with finance-related problems.

---

> ### Author Response · Authors · 2024-11-22
> **Weakness addressal and Technical Enhancements**
>
> We thank the reviewers for their valuable feedback and detailed review. Due to the length of responses, they are split over multiple comments.
>
> ## Fixes for Weaknesses
>
> ### Algorithm
>
> 1. **P1**: *a single-pass, incremental, adaptive, stable algorithm that could efficiently process large datasets in the order of quadratic complexity per iteration with respect to the dimension of the input space.*
>
>    **Response**: The complexity of the CVaR estimation/optimization algorithm per iteration depends on the computational complexity in computing \\(\Gamma(\cdot)\\). In the case of Gaussian where \\(\Gamma(x) = (x, xx^{\top})^{\top}\\), the complexity is quadratic in the dimension of the space. This holds for most distributions from the NEF space.
>
> 2. **P7**: *It is sufficient to seek within the subspace of measures that are absolutely continuous with respect to the original measure.*
>
>    **Response**: You are correct. The support of \\(dP_{x}/dv\\) will not be inside that of \\(dQ_{t+1}/dv\\) in the full sense but rather in the “almost everywhere” sense. We have updated the equations accordingly.
>
> 3. **Surrogate Measure Minimizing Variance**: *The conclusion that the surrogate measure needs to minimize the variance is not obvious...*
>
>    **Response**: The region \\(\{\phi(X) \geq \gamma_t\}\\) becomes rarer as \\(t \to \infty\\), and the goal of the sampling distribution is to address this issue. We select the sampling distribution with the minimum possible variance, which is the zero-variance distribution.
>
> 4. **P10**: *From the inverse function theorem, it follows that \\(m\\) is invertible. How do you compute that inverse in the non-Gaussian case?*
>
>    **Response**: In the Gaussian case, \\(m(\theta) = (\mu, \Sigma + \mu\mu^{\top})\\). Similarly, for most NEF members, it is possible to derive a closed-form expression for \\(m(\cdot)\\), and its inverse can be computed in time proportional to \\(d\\). We have corrected the previous error stating that \\(m^{-1}\\) requires \\(O(1)\\) time.
>
> 5. **P11**: *The update rule (19) will monotonically improve the concentration of the probability measure of the surrogate...*
>
>    **Response**: We aim to find a surrogate measure with higher probability concentration in the region \\(\{\phi(X) \geq \gamma_t\}\\). We clarified the statement to reflect the comparison to the original measure \\(\mathbb{P}_x\\).
>
> 6. **P14**: *Proposition 3: How reasonable is the hypothesis?*
>
>    **Response**: If the sampling ratio \\(\zeta_t\\) is constrained as \\(1 - \delta \leq \frac{\zeta_t(x)\mathbb{I}(\phi(x), \gamma_t)^{+}}{\mathbb{P}_x(\phi(X) \geq \gamma_t)} \leq 1 + \delta\\), we can lower bound the probability of the rare region \\(\{\phi(X) \geq \gamma_t\}\\).
>
> 7. **P15**: *How does a single sample approximate the quotient of two integrals?*
>
>    **Response**: Since both expectations are taken with respect to the same probability measure, a single sample can evaluate both expectations via the stochastic recursion in Equation (40). This technique is well-known in stochastic approximation literature. The proof is provided in Lemma 6.
>
> 8. **P15**: *Iterates track the ideal, but in most cases are \\(O(1)\\) computable.*
>
>    **Response**: As mentioned earlier, in the Gaussian case, computing \\(\Sigma\\) from \\(m\\) requires \\(\Theta(d^2)\\) time. This was corrected in the manuscript.
>
> 9. **P16**: *Why choose a Gaussian to approximate a truncated distribution?*
>
>    **Response**: While the values of \\(X \in \mathbb{R}^d\\) are not bounded, the NEF, which we use to approximate \\(\mathbb{P}_x\\), has an exponential tail, making the probability of regions outside \\(\{\phi(x) \geq \gamma_t\}\\) negligible. In cases where this tail needs to be clipped, truncated Gaussian distributions or other NEF variants can be used.
>
> 10. **P26**: *Can automatic differentiation be used instead of random perturbations?*
>
>     **Response**: Automatic differentiation is possible and can be used with the loss function in Equation (19).
>
> 11. **P27**: *Does finite difference estimation become intractable as dimension grows?*
>
>     **Response**: The finite differences converge to the true gradient at a rate bounded by the square of the step size. The dimension does not directly affect the estimation, and asymptotically, the recursion behaves like Gradient Descent (Theorem 3).
>
> 12. **P28**: *Does this turn the 2-timescale algorithm into a 3-timescale one?*
>
>     **Response**: \\(\texttt{VaR}\\) and \\(\texttt{CVaR}\\) estimation remain a two-timescale algorithm (Algorithm 1), but CVaR optimization (Algorithm 2) is a three-timescale algorithm due to the additional recursion for \\(\omega_t\\).
>
> --continued over to next comment--

---

> > ### Author Response · Authors · 2024-11-22
> > **Continued**
> >
> > ### Experiments
> >
> > 1. **P31**: *The experiments seem controversial, especially in section 6.1. What baseline is used for comparison?*
> >
> >    **Response**: We have dropped this experiment as exact \\(\texttt{VaR}\\) and \\(\texttt{CVaR}\\) values for some distributions are hard to compute.
> >
> > 2. **P39**: *The MC estimation needs variances and comparisons to another method.*
> >
> >    **Response**: As mentioned above, the experiment has been removed.
> >
> > 3. **P40**: *Experiments on actual data need more explanation for non-experts.*
> >
> >    **Response**: Additional details for the finance-related experiments are provided in the updated manuscript, specifically in sections 6.1.1 and 6.1.2.
> >
> > ### Critical Changes
> >
> > 1. **Comparison to He et al. (2023)**: *Explain which parts of your algorithm are new and compare to He et al. (2023).*
> >
> >    **Response**: He et al. (2023) introduced a generalized quantile estimation algorithm, but it does not address extreme cases. Our algorithm improves quantile estimation in scenarios with small upside risk thresholds.
> >
> > 2. **Theorems 2, 3, 4**: *Give more details as to why \\(\gamma_t\\) converges but \\(\eta_t\\) does not...*
> >
> >    **Response**: The proof follows the dynamical systems approach from *Stochastic Approximation: A Dynamical Systems Approach* by Vivek S. Borkar. We show that \\(\gamma_t\\) converges to the true quantile, \\(\eta_t\\) to the surrogate distribution, and \\(\bar{\eta}_t\\) to the \\(\texttt{CVaR}\\).
> >
> > 3. **Choice of Exponential Family**: *How does the choice of exponential family impact the variational inference error?*
> >
> >    **Response**: The NEF allows tractable algorithms with closed-form expressions, ensuring stable results. The approximation error depends on the chosen distribution space. Stability may fail if less smooth spaces are chosen.
> >
> > ### Optional Changes
> >
> > 1. **Citation Format**: *Put citations in parentheses/square brackets with \\(\backslash \texttt{parencite}\\) or \\(\backslash \texttt{citep}\\).*
> >
> >    **Response**: Done.
> >
> > 2. **Referencing Theorems/Propositions**: *Avoid referring to theorems or propositions not yet given, or add a comment like "as we will see in...".*
> >
> >    **Response**: Done.
> >
> > 3. **Notations for Measures**: *Would using the PDFs instead of Radon-Nikodym derivatives cause any theoretical issues?*
> >
> >    **Response**: Replacing the Radon-Nikodym derivative with PDFs does not cause theoretical issues. The results will still follow.
> > ### Clarification
> >
> > 1. **P2**: The superquantile ... coherency with respect to the return distribution.
> >    *What does *coherency* mean here?*
> >    **Response**: Coherency in this context refers to the properties of a coherent risk measure, as the superquantile or **CVaR** is a coherent risk measure. The properties are mentioned in the following statement, and the reference has been fixed to the correct position.
> >
> > 2. **P3**: Variance reduction techniques, particularly importance sampling, have been utilized to manage the large variance in extreme risk estimation.
> >    *Define "IS" acronym here.*
> >    **Response**: Fixed in the updated manuscript.
> >
> > 3. **P5**: Proposition 1. The proof can be shortened by just using the convexity of ReLU.
> >    **Response**: ReLU can be used to prove this, and we thank the reviewer for providing that solution. However, our current proof is based on fundamental principles. We kindly request the reviewer to allow the current proof to be retained.
> >
> > 4. **P7**: Equation 11. *Is the set inclusion sign in the wrong direction?*
> >    **Response**: The direction of the inclusion is correct. Note that we are considering \\( \neq \\). The statement above the inclusion is for \\( = \\).
> >
> > 5. **P7**: Before Equation 12. *Where does the derivative \\( \frac{d\mathbb{P}}{dQ_t} \\) come from? Also, I think that from this point on there is a time index confusion between \\( Q_t \\) and \\( Q_{t+1} \\) (isn't \\( X_{t+1} \\) drawn from \\( Q_{t+1} \\)?)*
> >    **Response**: In the time indexing considered in this paper, the sample \\( X_{t+1} \\) is generated from the distribution \\( Q_t \\). This notation indicates that the distribution \\( Q_t \\) at the current time \\( t \\) generates the sample \\( X_{t+1} \\), which is used to obtain the subsequent distribution \\( Q_{t+1} \\).
> >
> > 6. **P8**: Equation 14. *\\( \ell_{t+1} \\) has a simpler expression/interpretation: the true probability to exceed \\( \gamma_t \\).*
> >    **Response**: Correct.
> >
> > 7. **P10**: \\( K(\theta) \\) is the cumulant function, which ensures that the distribution is normalized. *Is \\( m(\theta) \\) equal to the gradient of the cumulant? I think you use this fact in a later proof.*
> >    **Response**: Yes, correct. The details with references are provided on the same page in the paragraph following Equation (21).
> >
> > -- continued over to next comment --

---

> > ### Comment · Reviewer_dLZW · 2024-12-05
> >
> > I thank the authors for their responses, and will post my follow up questions and remarks below.
> >
> > > > P7: It is sufficient to seek within the subspace of measures that are absolutely continuous with respect to the original measure.
> > >
> > > Response: You are correct. The support of $dQ_{t+1}/d\nu$ will not be inside that of $dP_x/d\nu$ in the full sense but rather in the “almost everywhere” sense. We have updated the equations accordingly.
> >
> > My issue was not with the distinction between "everywhere" and "almost everywhere". I wanted to know whether $Q_{t+1}$ has any mass at all outside of the extremal region $\{\phi(X) \geq \gamma_t\}$. Is $Q_{t+1}$ just $P_x$ restricted to $\{\phi(X) \geq \gamma_t\}$?
> >
> > > >Surrogate Measure Minimizing Variance: The conclusion that the surrogate measure needs to minimize the variance is not obvious...
> > >
> > > Response: The region $\{\phi(X) \geq \gamma_t\}$ becomes rarer as $t \to \infty$, and the goal of the sampling distribution is to address this issue. We select the sampling distribution with the minimum possible variance, which is the zero-variance distribution.
> >
> > If $Q_t$ is just the restriction of $P_x$ to the region $\{\phi(X) \geq \gamma_t\}$, why do you need to introduce a new random variable and minimize its variance? I am probably missing something, but at the moment it does feel like needless notational complexity to me.
> >
> > >> P10: From the inverse function theorem, it follows that $m$ is invertible. How do you compute that inverse in the non-Gaussian case?
> > >
> > > Response: [...] Similarly, for most NEF members, it is possible to derive a closed-form expression for $m$, and its inverse can be computed in time proportional to $d$.
> >
> > Isn't it $d^2$?
> >
> > > > P14: Proposition 3: How reasonable is the hypothesis?
> > >
> > > Response: If the sampling ratio is constrained, we can lower bound the probability of the rare region.
> >
> > I understand that. My question is whether it is reasonable to assume this kind of constraint on the sampling ratio in the first place.
> >
> > >> P27: Does finite difference estimation become intractable as dimension grows?
> > >
> > >Response: The finite differences converge to the true gradient at a rate bounded by the square of the step size. The dimension does not directly affect the estimation, and asymptotically, the recursion behaves like Gradient Descent (Theorem 3).
> >
> > Let me rephrase. How does the precision of the gradient estimate scale with the dimension? My intuition is that, if it behaves like deterministic finite differences, you need to scale the number of random perturbation samples as the dimension grows (unlike reverse-mode automatic differentiation, whose precision is dimension-agnostic).
> >
> > > > P28: Does this turn the 2-timescale algorithm into a 3-timescale one? _If so, will it ever converge?_
> > >
> > >Response: VaR and CVaR and estimation remain a two-timescale algorithm (Algorithm 1), but CVaR optimization (Algorithm 2) is a three-timescale algorithm due to the additional recursion for $\omega_t$.
> >
> > You left out the second part of the question, which interrogates practical tractability of a 3-timescale algorithm.

---

> ### Author Response · Authors · 2024-11-22
> **Continued**
>
> 8. **P13**: Theorem 1. *What do \\( \mathcal{Q} \\) and \\( \mathcal{C} \\) correspond to intuitively, for someone who might have skipped the proofs?*
>    **Response**: Added on Page 14 following the results. Please refer to Page 14.
>
> 9. **P15**: *One might fall prey to over-compliance, i.e., scenarios where a substantial fraction of the samples belong to the region \\( \{ \phi(X) \geq \gamma_t \} \\). Why is that bad? Isn't that precisely what we want?*
>    **Response**: It can be problematic, as we may observe a disproportionately large number of samples from the region \\( \{ \phi(X) \geq \gamma_t \} \\). To maintain balance, it is essential to occasionally sample from the original measure as well. This process is regulated by the factor \\( \lambda \in [0, 1] \\).
>
> 10. **P15**: *We obtain the sampling measure \\( \hat{Q}_t \\). I would denote it as \\( \hat{Q}_{\theta_t} \\).*
>     **Response**: We request the reviewer to retain the notation. \\( \hat{Q}_{t} \\) signifies either \\( Q_{\theta_t} \\) or \\( \mathbb{P}_x \\).
>
> 11. **P15**: With parameter \\( \lambda \in [0,1] \\) (fixed a priori). *How does one choose it?*
>     **Response**: \\( \lambda \\) is chosen a priori. One can use a schedule to adjust \\( \lambda \\) over time. In the early stages, a larger \\( \lambda \\) can be chosen to give more weight to \\( \mathbb{P}_x \\), making it more likely. As \\( t \\) increases, \\( \lambda \\) is gradually reduced to shift the focus towards the surrogate distribution.
>
> 12. **P25**: CVaR optimization involves adjusting parameters to minimize the expected losses in extreme scenarios. *Then why is Eq. 76 a maximization?*
>     **Response**: It was an error. Now fixed in the updated manuscript.
>
> 13. **P29**: One of the challenges is that the true ... known. Instead, one has access to an oracle which can provide samples or realizations from this distribution \\( \mathbb{P}_x \\).
>     *Does that limit the number of time steps \\( t \\) to the number of drawn samples?*
>     **Response**: This suggests that the original probability measure \\( \mathbb{P}_x \\) is unknown, but samples from it are available. This is a common scenario, particularly in reinforcement learning and other estimation problems. The process will be time-dependent, and the relationship between time and the estimates is difficult to quantify, especially in the context of stochastic approximation algorithms.
>
> 14. **P30**: *Note that we follow the same timescale as that of \\( \gamma_t \\) since there exists a unilateral coupling between \\( \kappa_t \\) and other iterates.*
>     *Why is this justified?*
>     **Response**: \\( \kappa_t \\) approximates the latent probability measure \\( \mathbb{P}_x \\). The samples drawn from \\( \mathbb{P}_x \\) are used to estimate \\( \mathbb{P}_x \\), and this approximation is then employed in the update recursion for \\( \gamma_t \\) in \\( \zeta_t \\). This indicates a unilateral coupling, where \\( \gamma_t \\) depends on \\( \kappa_t \\), but not vice versa.
>
> 15. **P31**: *We compare the values of VaR and CVaR generated through Monte-Carlo (MC) estimates against the values generated by our procedure.*
>     *What is your stopping criterion and hyperparameter setup?*
>     **Response**: Please refer to Response 1 of Section 2.1.2 above.
>
> 16. **P33**: *We choose a period of about a ... Eq. 90 using CVXPY (Diamond & Boyd, 2016) and MOSEK ApS (2024) solver.*
>     *Can you adapt your CVaR minimization algorithm to a constrained setting? If I understand correctly, here you estimate first and then optimize second?*
>     **Response**: Yes, it can be adapted, but in our setting, we focus on the CVaR optimization of the negative returns distribution of the portfolio.
>
> 17. **P33**: *We also verify by the efficient frontier analysis.*
>     *What does this entail?*
>     **Response**: The detailed explanation for efficient frontier analysis and it's significance in portfolio optimization is added in the updated manuscript. Please refer to section 6.1.2.
>
> 18. **P36**: *The results demonstrate that the agent maintains stability despite increasing levels of uncertainty associated with decreasing \\( \rho \\).*
>     *How do we deduce this from the curves?*
>     **Response**: The stability is evident from the curves as the rewards exhibit no abrupt deviations even under varying levels of uncertainty, characterized by decreasing $\rho$. Additionally, the variance bands in rewards remains consistent across these levels.
>
> 19. **P37**: *What is the difference between the values and the rewards?*
>     **Response**: Rewards are received from following the CVaR policy, whereas the value is the CVaR-augmented objective function. A detailed explanation is provided in the updated manuscript.
>
> ---
>
> ### Typos
> All typos have been fixed, and the manuscript has been updated accordingly.

---

> > ### Comment · Reviewer_dLZW · 2024-12-05
> >
> > ## Clarification
> >
> > > > P7: Equation 11. Is the set inclusion sign in the wrong direction?
> > >
> > > Response: The direction of the inclusion is correct. Note that we are considering $\neq$. The statement above the inclusion is for $=$.
> >
> > Indeed, the two statements refer to $\neq$ and $=$ respectively. The first one is $p = 0 \implies q = 0$, so shouldn't the second one be the contraposition $q \neq 0 \implies p \neq 0$, or equivalently $\\{q \neq 0\\} \subseteq \\{p \neq 0\\}$? Sorry if I'm missing something obvious.
> >
> > > > P7: Also, I think that from this point on there is a time index confusion between $Q_t$ and $Q_{t+1}$
> > >
> > >Response: In the time indexing considered in this paper, the sample $X_{t+1}$ is generated from the distribution $Q_t$.
> >
> > A few lines earlier on P7, you state "In our approach, at each iteration t, the sample $X_{t+1}$ is chosen using a surrogate measure $Q_{t+1}$ which is possibly different from the given measure $P_x$."
> > Please fix whichever statement is incorrect and review temporal indices in the rest of the paper accordingly.
> >
> > > >P15: "One might fall prey to over-compliance, i.e., scenarios where a substantial fraction of the samples belong to the region $\phi(X) \geq \gamma_t$." Why is that bad? Isn't that precisely what we want?
> > >
> > >Response: It can be problematic, as we may observe a disproportionately large number of samples from the region $\phi(X) \geq \gamma_t$. To maintain balance, it is essential to occasionally sample from the original measure as well. This process is regulated by the factor $\lambda$.
> >
> > I still don't understand why we need to "maintain balance". Wouldn't it be ideal to only sample extremal values?

---

> ### Comment · Reviewer_dLZW · 2024-12-05
>
> ## Experiments
>
> > > P31: The experiments seem controversial, especially in section 6.1. What baseline is used for comparison?
> >
> > Response: We have dropped this experiment as exact VaR and CVaR values for some distributions are hard to compute.
>
> The purpose of the algorithms we discuss is not to compute exact values, it is to improve upon Monte-Carlo estimation. As pointed out by Reviewer Y6cf, the right way to perform this comparison (assuming that MC and your method are both unbiased) would be to plot and contrast standard deviations of the estimates for both methods. **Giving up on this experiment is an unwise choice if you want to prove that your algorithm outperforms naive MC.**
>
> ## Critical changes
>
> The authors seem to have ignored the main weakness I pointed out (related to clarity of the exposition), as well as all the critical changes I requested to improve it (keep the core reasoning and arguments in the paper's body, move proofs and technical details to the appendix, add a clear outline of the main intuitions similar to the one I tried to give in my review). **The lack of clarity was pointed out by all reviewers and has not been addressed. In the current state I will not recommend accepting the paper for publication.**
>
> >> Comparison to He et al. (2023): Explain which parts of your algorithm are new and compare to He et al. (2023).
> >
> >Response: He et al. (2023) introduced a generalized quantile estimation algorithm, but it does not address extreme cases. Our algorithm improves quantile estimation in scenarios with small upside risk thresholds.
>
> That does not seem to be true. For instance, on page 2 of He et al. (2023), I read _"In this paper, we are interested in situations where the root-finding problem involves extremal or rare-event considerations. A primary example is extreme quantile estimation, in which the target probability level can be very close to 0 or 1 (e.g., $10^{-6}$)."_ Please clarify the novelties that your approach brings.
>
> > > Theorems 2, 3, 4: Give more details as to why $\gamma_t$ converges but $\eta_t$ does not...
> >
> > Response: The proof follows the dynamical systems approach from Stochastic Approximation: A Dynamical Systems Approach by Vivek S. Borkar. We show that $\gamma_t$ converges to the true quantile, $\eta_t$ to the surrogate distribution, and $\bar{\eta}_t$ to the CVaR.
>
> My question was more about getting an intuitive understanding of why this is the case. In particular, why the choice of variational family influences $\eta_t$ and $\bar{\eta}_t$ but not $\gamma_t$.

---

> > ### Author Response · Authors · 2024-12-14
> > **Response - 2**
> >
> > 'The purpose of the algorithms we discuss is not to compute exact values, it is to improve upon Monte-Carlo estimation. As pointed out by Reviewer Y6cf, the right way to perform this comparison (assuming that MC and your method are both unbiased) would be to plot and contrast the standard deviations of the estimates for both methods. Giving up on this experiment is an unwise choice if you want to prove that your algorithm outperforms naive MC.
> >
> > **Response:**  We will add those as suggested within this weekend
> >
> > The authors seem to have ignored the main weakness I pointed out (related to clarity of the exposition), as well as all the critical changes I requested to improve it (keep the core reasoning and arguments in the paper's body, move proofs and technical details to the appendix, add a clear outline of the main intuitions similar to the one I tried to give in my review). The lack of clarity was pointed out by all reviewers and has not been addressed. In the current state, I will not recommend accepting the paper for publication.
> >
> > **Response:** We have now restructured the manuscript. Kindly have a look.
> >
> > That does not seem to be true. For instance, on page 2 of He et al. (2023), I read, "In this paper, we are interested in situations where the root-finding problem involves extremal or rare-event considerations. A primary example is extreme quantile estimation...
> >
> > **Response:** He et. al (2023) is a generalized root finding and quantile estimation paper that can be applied to extreme cases.
> > But the algorithm proposed there is very generic. Specifically, their algorithm treats variance reduction as a black-box function, denoted as I(), without providing explicit details on how the variance reduction is achieved or optimized within the algorithm. This abstraction means that the method does not delve into the specific mechanisms or techniques for handling variance reduction, leaving it as an assumed external function. However, in our case, we propose a more practical version with details about how to overcome the unavailability of samples through an approximate surrogate distribution, which serves as an approximation of the true underlying zero variance distribution. This surrogate model allows us to generate reliable estimates and make predictions even when access to direct samples is limited or infeasible through an adaptive mechanism which is fine tuned as the iterates evolve, making the predictions even better as time goes to infinity.
> >
> > My question was more about getting an intuitive understanding of why this is the case. In particular, why the choice of variational family influences .....
> >
> > **Response:** Variational family does not influence the convergence. As explained in the claims, $\gamma_t$ converges to the VaR as it tracks the solution to the convex optimization problem given in (5). However, $\eta_t$ tracks the approximate zero variance distribution and $\bar{\eta}_t$ tracks the mean of the zero variance distribution. Hence $\bar{\eta}_t$ converges to the CVaR as CVaR is the mean of the zero-variance distribution.

---

### Review · Reviewer_Y6cf · 2024-11-07

**Summary Of Contributions:**

The paper addresses estimating the (conditional) value at risk for extremely rare events. An estimation algorithm based on a surrogate distribution for rare events is derived and made tractable through approximations. The algorithm is extended for streaming data and latent data distribution settings.

**Audience:**

Yes

**Broader Impact Concerns:**

I have no concern about a negative broader impact.
The paper introduces algorithms to incorporate risk into decision-making problems - a desirable property, as it adds to the technical repertoire of dealing with risk in a principled manner.

**Claims And Evidence:**

Yes

**Requested Changes:**

### General
[G1] From my understanding, Algorithm 1 and its analysis are the main contributions of this paper. 4 and 5 are extensions needed to apply the algorithm to specific problems. The section structure and titles do not fit this. I would suggest restructuring:

[G2] Move the related work section from section 4 to section 3.
In order to clarify the structure of the paper, include an outline subsection following *1.2 Our contribution*.

[G3] Many figures do not have proper axes labels or text references.
### Analysis Section
[A1]: For some statements, the proof precedes the statement; these should be switched, for example, in Proposition 3.

[A2]: On page 10, When the natural exponential family is introduced, a rationale is missing. Is the problem intractable without analytic solutions?

[A3]: On page 12, Lemma 3 appears to be cited from prior work. Please denote these consistently by directly citing them in the result, similar to what is done in Lemma 1.

[A4]: On page 15: The derivation at the bottom of the page, leading up to (44), could be transformed into a statement (e.g., a corollary).

[A5]: On page 31: In Theorem 4, please mention the specific proofs or provide a proof sketch.
### Experimental Section

The experimental section is unconvincing. There are no comparisons to true values or baselines. It is unclear whether the VaR/CVaR optimization achieves the desired effect, as traditional metrics for return/reward are reported without context.

[E1]: Table 1 is incorrectly labeled as 6.1 when referenced.

[E] 6.1: The comparison to Monte Carlo (MC) estimation lacks any baseline or ground truth values. Is MC or EX more effective? Additionally, including standard deviations from repeated experiments would help address the concerns raised in the remark on page 31.

[E] 6.3: The authors state that the variance of the CVaR performance is well-behaved. However, the results do not demonstrate how the CVaR policy is superior to policies trained using traditional reinforcement learning (RL) algorithms, such as those from stable baselines.

[E] 6.5: The experimental results are not well described. The figures are somewhat confusing, as they run for different amounts of steps and appear to stop prematurely, sometimes at high-risk indices.

### Minor Comments
- p2: The image should be treated as a figure and referenced.
- Figures 2, 5, 7, and 9 are not referenced in the text.

**Strengths And Weaknesses:**

### Strengths
- [S1] The analytical derivations are detailed; the algorithm is thus transparent and well-motivated.
- [S2] Example applications from different domains are provided.

### Weaknesses
- [W1] The structure of the paper is not entirely clear to me. A variety of ideas is presented, and the transition is sometimes not very smooth, for example, 3->4 has an entire related work section. The paper presents a variety of ideas, but as they are submitted as a single manuscript, they should be tied together more coherently.
- [W2] The experimental section is not convincing nor polished.

---

> ### Author Response · Authors · 2024-11-21
> **Clarifications and Revisions on Algorithm Contributions, Analysis, and Experimental Results**
>
> We thank the reviewers for their valuable feedback. We clarified the core contributions, refined the structure, addressed analysis and experimental concerns, and corrected figure references. These improvements enhanced the manuscript's clarity, rigour, and flow.
>
> ### General Comments
> - **[G1]** *From my understanding, Algorithm 1 and its analysis are the main contributions of this paper. 4 and 5 are extensions needed to apply the algorithm to specific problems. The section structure and titles do not fit this. I would suggest restructuring.*
>
>     **Response:** Algorithms 1 and 2 represent the core contributions, focusing on superquantile estimation and optimization, respectively. Algorithm 3 extends Algorithm 1 to cases with latent probability measures, while Algorithm 4 adapts Algorithm 2 for reinforcement learning applications. We believe the section titles and structure align well with the manuscript’s content and flow. If there are specific concerns, we request further details to address them effectively.
>     **P.S.:** Algorithm 5 does not exist.
>
> - **[G2]** *Move the related work section from section 4 to section 3. In order to clarify the structure of the paper, include an outline subsection following 1.2 Our contribution.*
>
>     **Response:** Section 3 focuses on superquantile estimation and its analysis. Moving related work from Section 4 to Section 3 would disrupt the manuscript’s flow, as related work in Section 4 pertains specifically to CVaR optimization. A complete outline has been added after the contribution section in 1.3.
>
> - **[G3]** *Many figures do not have proper axes labels or text references.*
>
>     **Response:** Fixed in the updated manuscript.
>
> ### Analysis Section
> - **[A1]** *For some statements, the proof precedes the statement; these should be switched, for example, in Proposition 3.*
>
>     **Response:** Resolved in the revised manuscript.
>
> - **[A2]** *On page 10, when the natural exponential family is introduced, a rationale is missing. Is the problem intractable without analytic solutions?*
>
>     **Response:** You are correct. NEF is a standard parameterized family of distributions with closed-form expressions, enabling tractable algorithms by allowing moment projection of the zero-variance measure onto this space. The approximation error depends on the distribution space being projected, but the use of NEF ensures stable and tractable algorithms.
>
> - **[A3]** *On page 12, Lemma 3 appears to be cited from prior work. Please denote these consistently by directly citing them in the result, similar to what is done in Lemma 1.*
>
>     **Response:** Resolved in the revised manuscript.
>
> - **[A4]** *On page 15: The derivation at the bottom of the page, leading up to (44), could be transformed into a statement (e.g., a corollary).*
>
>     **Response:** This section provides intuition for updating \(\bar{\eta}_t\) and cannot be reframed as a corollary. However, additional clarifications have been added.
>
> - **[A5]** *On page 31: In Theorem 4, please mention the specific proofs or provide a proof sketch.*
>
>     **Response:** A proof sketch has been added in the updated manuscript. Please refer to the proof section of Theorem 4 in Section 5.
>
> ### Experimental Section
> - **[E1]** *Table 1 is incorrectly labelled as 6.1 when referenced.*
>
>     **Response:** Fixed in the updated manuscript.
>
> - **[E2]** *6.1: The comparison to Monte Carlo (MC) estimation lacks any baseline or ground truth values. Is MC or EX more effective? Additionally, including standard deviations from repeated experiments would help address the concerns raised in the remark on page 31.*
>
>     **Response:** We have dropped this experiment as the exact \(\texttt{VaR}\) and \(\texttt{CVaR}\) values for some distributions are challenging to compute.
>
> - **[E3]** *6.3: The authors state that the variance of the CVaR performance is well-behaved. However, the results do not demonstrate how the CVaR policy is superior to policies trained using traditional reinforcement learning (RL) algorithms, such as those from stable baselines.*
>
>     **Response:** The goal of the experiment is to maximize the CVaR of rewards, not to benchmark against traditional RL algorithms like those in Stable Baselines. The focus is on evaluating the effectiveness of our method for CVaR optimization.
>
> - **[E4]** *6.5: The experimental results are not well described. The figures are somewhat confusing, as they run for different amounts of steps and appear to stop prematurely, sometimes at high-risk indices.*
>
>     **Response:** We have updated the experimental descriptions in Section 6.5. Please refer to the updated manuscript.
>
> ### Minor Comments
> - **[M1]** *p2: The image should be treated as a figure and referenced.*
>
>     **Response:** Fixed in the updated manuscript.
>
> - **[M2]** *Figures 2, 5, 7, and 9 are not referenced in the text.*
>
>     **Response:** Fixed in the updated manuscript.

---

### Decision · Action_Editor_kAz6 · 2025-01-09

**Recommendation:** Reject

**Comment:**

To begin, I apologize for the fact that the review process for this paper took much longer than is typical by TMLR standards. That said, the paper itself is much longer than a typical TMLR submission, and included a lot of technical details which ended up being key points for discussion between the reviewers and authors.

Overall, two of the three reviewers recommended against acceptance, and even the reviewer which leaned towards acceptance had significant remaining concerns about how the actual utility of the proposed algorithm. Here are some final comments from the reviewers:

> My main remaining concern is that it is still unclear how the proposed algorithm performs compared to existing algorithms. If the paper should be accepted, I would strongly recommend the authors add a comparison to MC estimation methods and ground truth or baseline (C)VaR in terms of needed samples, computation times, and accuracy.

> Regarding the content, I was not able to verify every derivation. However, even my modestly technical questions regarding the beginning of the proof remained unanswered. Among others: why is $Q\_{t}$ defined in such a convoluted way by minimizing a variance? do randomized finite differences scale well with dimension? how tractable is a three-timescale algorithm? Additionally, the experiment comparing the new VaR estimation method with a simple Monte-Carlo scheme was not meaningful due to the lack of confidence intervals, but instead of fixing it, the authors removed it entirely. This strongly hinders practical evaluation of the variance reduction benefits.

> Regarding the form, the main intuitions of the algorithm proposed here are completely obscured by the technicalities and formalism. Most readers would benefit from a cleaner separation between ideas and mathematical boilerplate. It would also help to outline which parts of the method are new, and how it compares to the previous state of the art (which is apparently cited but not analyzed).

> The overall story and the contents seem interesting; but partially due to relatively many wrong mathematical connections (that could be fixed easily) it's been hard to completely verify the correctness. The response has been partially satisfactory but did not resolve my aforementioned concerns.

I am in agreement with the reviewers, and I don't think the current paper is suitable to be accepted to TMLR.

**Audience:**

In principle, estimation of typical risk metrics such as VaR and CVaR under extreme settings with very little data available is a natural problem of interest in machine learning, and so there naturally should be some interest in this paper. That said, with the aforementioned clarity issues, and the fact that this paper does not read like a TMLR paper, but rather a paper written for experts in mathematical financial risk, the effective audience for this paper as it stands is quite limited.

**Claims And Evidence:**

My understanding of the main claim of the paper is simply that the authors have introduced a new procedure for estimating VaR and CVaR in highly "extreme" settings, i.e., for high-level quantiles (here, VaR of the returns distribution) or CVaR based on such high-level quantiles.

The derivation of the authors' new procedure is technically detailed, and includes analysis by which they establish sufficient conditions for their procedure to enjoy desirable properties such as almost-sure convergence to the desired risk value, even when very "deep" into the tails. They also provide a more general-use procedure which can be applied when the distribution is unknown, and conduct empirical experiments to evaluate the practical utility of their proposed method.

While at face value there is a lot of "evidence" provided for the straightforward main claims, the problem here is that the evidence is not clear. All of the reviewers had significant difficulties parsing the details of the theoretical analysis, and many points raised by the reviewers remain unanswered.